# Multimodal Scaling Laws for Task & Data-Optimized Models of Visual Cortex

**Abdulkadir Gokce** [1]   **Yingtian Tang** [1]   **Martin Schrimpf** [1]

## Abstract

Task-optimized neural networks are the leading in-silico models of sensory cortex, yet the field lacks a unified understanding of which modeling choices drive improved brain alignment. Prior NeuroAI work is fragmented across datasets and modalities, making it difficult to determine robust scaling trends. Here, we systematically investigate the scaling laws of model-to-brain alignment across 8 neural datasets (spanning electrophysiology, fMRI, EEG, and MEG) and over 600 models with diverse architectures and pretraining configurations. We report three scaling trends: (1) *Pretraining saturation*: Alignment improves with pretraining compute and data scale but saturates across all recording modalities. (2) *Complementary fine-tuning*: Hybrid task & neural data optimization yields consistent improvements in alignment that generalize across datasets and modalities. (3) *Mapping scaling*: Increasing the number of neural samples to fit model-to-brain mappings yields log-linear gains with the largest impact on alignment. Finally, we propose a novel subject-shared cross-attention mapping which drastically reduces parameter count and improves alignment. Taken together, these results establish multimodal scaling laws that guide resource allocation for next-generation brain models.

## 1. Introduction

Task-optimized deep networks have become a central model for studying sensory systems, acting as computational hypotheses that can be tested by comparing in-silico representations to brain recordings (Yamins & DiCarlo, 2016; Richards et al., 2019; Schrimpf et al., 2020). Both sides of this comparison are advancing rapidly: machine learning models are trained in a variety of ways to achieve ever-better ground-truth performance; and datasets in brain science cover multiple species and recording modalities at unprecedented scale and signal quality (Allen et al., 2021; Hebart et al., 2023; Papale et al., 2025).

However, since current models remain far from perfect replicas of brain function, the field lacks a clear path forward: *what is the most effective way to improve model-to-brain alignment?* One view for addressing this question is to incorporate more and more details about the brain into the models, such as neuroanatomy, biophysical details, learning processes, and so forth. While perfectly valid, we believe there is a more direct approach that is beginning to show promise: identifying scaling trends, along which alignment improves in a predictable manner. Recent advances in machine learning have been largely driven by such increased scale, yet there is no established analogue in NeuroAI.

Multiple factors underlie the building of a brain model: If alignment is primarily limited by task-optimized pretraining scale, then the best path forward is larger models and more data; if it is limited by the neural supervision signal, then curated neural objectives and training recipes should dominate; and if it is limited by the mapping and neural sample size, then methodological choices may overshadow representational differences. Without disentangling these factors, it is difficult to establish a resource allocation plan for developing improved brain-aligned models and interpret why one model outperforms another.

To tackle this open question, we turn to a well-developed concept from deep learning known as *scaling laws*: task performance often follows predictable power-law trends as a function of training data, compute, and model capacity (Kaplan et al., 2020; Hoffmann et al., 2022; Zhai et al., 2022). Initial studies in NeuroAI report improved correspondence to neural measurements with increased model scale and/or training data (Antonello et al., 2023), while others suggest that different notions of alignment can saturate and may be driven more by data than by parameters (Gokce & Schrimpf, 2025). A major obstacle is the lack of unified evaluations that span multiple datasets and modalities under a consistent protocol, making it hard to separate robust trends from benchmark-specific artifacts.

Here we provide a systematic, modality-spanning analysis of how brain–model alignment scales with (i) *pretraining*

[1]EPFL. Correspondence to: Abdulkadir Gokce <abdulkadir.gokce@epfl.ch>, Martin Schrimpf <martin.schrimpf@epfl.ch>.

*Proceedings of the $43^{rd}$ International Conference on Machine Learning*, Seoul, South Korea. PMLR 306, 2026. Copyright 2026 by the author(s).

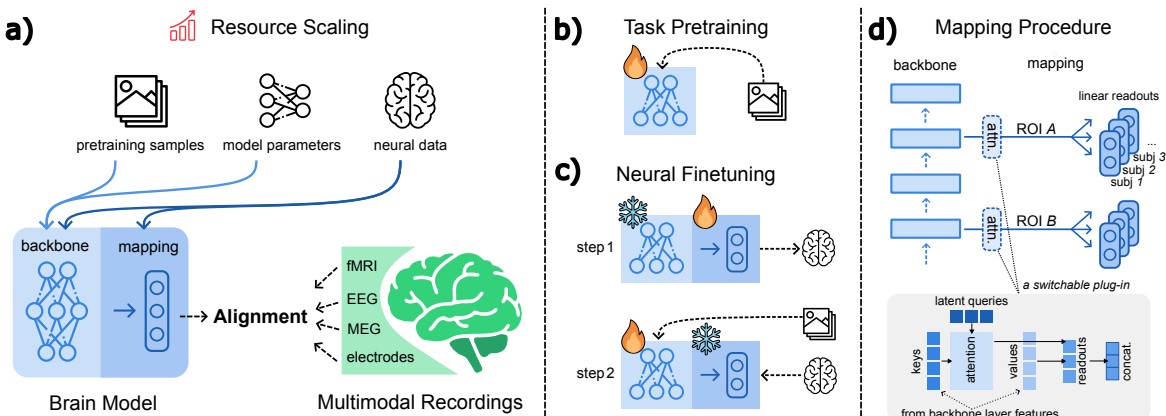

*Figure 1.* **Overview of study design and alignment pipeline. a)** We evaluate *brain–model alignment* by pairing a vision backbone with a learned mapping from model features to neural responses, and comparing predictions to multimodal recordings (electrophysiology, fMRI, EEG, MEG). We study how alignment scales with three ingredients: pretraining resources (data/compute and model size), the amount of neural supervision used to adapt the backbone, and the amount of neural data available for fitting mappings. **b)** Models are first pretrained on large-scale vision data, optionally followed by **c)** neural fine-tuning using targets from specific brain regions, enabling controlled comparisons of task-only vs. neurally supervised representations. **d)** For each dataset and ROI, we select a backbone layer and fit an ROI-specific mapping, including linear readouts and a novel light-weight attention-based probe that can share parameters across subjects while retaining subject-specific outputs.

*resources* used to train models, (ii) *neural fine-tuning* for guiding whole-model representations towards being more brain-like, and (iii) *neural mapping* as a final step for linking frozen representations to brain recordings. We evaluate over 600 pretrained vision models on eight widely used neural datasets spanning macaque electrophysiology and human fMRI/EEG/MEG. We use a consistent pipeline to enable controlled comparisons across datasets, modalities, and model families. Fig. 1 shows an overview of the evaluation pipeline and the three primary scaling axes.

**Contributions.**

- **Unified scaling laws across modalities.** We show that pretraining scale increases alignment but saturates at large scale, across electrophysiology, fMRI, EEG, and MEG.

- **Neural supervision provides complementary gains.** We find that hybrid task+neural fine-tuning consistently but moderately improves alignment, and that these gains transfer across datasets and modalities.

- **Neural data drives mapping improvements.** We show that dataset-specific mapping performance strongly improves log-linearly with increased paired stimulus–response data.

- **Multi-subject readouts improve data efficiency.** We introduce a multi-subject attention-based mapping that shares parameters across subjects and can match or exceed per-subject linear decoding in some regimes while using an order of magnitude fewer parameters.

## 2. Related Work

**Neural benchmarks and large-scale datasets.** NeuroAI increasingly utilizes standardized benchmarks and shared protocols, enabling direct comparisons across model classes (Schrimpf et al., 2018; 2020; 2021; Cichy et al., 2019; 2021; Turishcheva et al., 2024). Recent large-scale datasets further support systematic evaluation across modalities, including NSD (Allen et al., 2021; Kay et al., 2013) and (Boyle et al., 2020) for natural-scene fMRI, the THINGS ecosystem spanning fMRI/MEG/EEG over shared object-image stimuli (Hebart et al., 2023; 2019; Grootswagers et al., 2022; Gifford et al., 2022), and electrophysiology benchmarks that precisely measure neuron predictivity in macaque visual cortex (Freeman et al., 2013; Majaj et al., 2015; Papale et al., 2025). Our work builds on these resources by evaluating many modern vision models across multiple datasets and modalities under a unified, reliability-aware pipeline.

**Model–brain alignment and neural supervision.** Most alignment studies quantify correspondence via linear predictivity or representational similarity on frozen features, often using regularized linear readouts (Yamins et al., 2014; Schrimpf et al., 2018; Conwell et al., 2024). Recent work explores richer mappings (e.g., improved readouts and attention-based decoders; Ivanova et al., 2022; Saha et al., 2025; Adeli et al., 2025; Hwang et al., 2025; Yu et al., 2026; Chen et al., 2026) and end-to-end neural prediction objectives (Khosla et al., 2022; St-Yves et al., 2023). Complementarily, "brain-tuning" task-optimized models on neural data can improve predictivity and transfer to other benchmarks or properties such as robustness (Dapello et al., 2023;

*Table 1.* Overview of datasets used in this paper; all visual stimuli. Names from now on abbreviated to concise identifier handles.

| Dataset | Modality | Dataset Handle | # Stimuli |
|---|---|---|---|
| Freeman et al. (2013) | Ephys | FZ-EP | 135 |
| Majaj et al. (2015) | Ephys | MH-EP | 3,200 |
| Papale et al. (2025) | Ephys | TVSD-EP | 22,248 |
| Hebart et al. (2023) | fMRI | T-fMRI | 8,640 |
| Allen et al. (2021) | fMRI | NSD-fMRI | 70,000 |
| Hebart et al. (2023) | MEG | T-MEG | 22,248 |
| Grootswagers et al. (2022) | EEG | T-EEG1 | 22,248 |
| Gifford et al. (2022) | EEG | T-EEG2 | 16,740 |

Moussa et al., 2025; Lu et al., 2024). We connect these lines of investigation by separating the roles of pretraining, neural fine-tuning, and mapping design within a coherent evaluation framework.

**Scaling laws for alignment.** Deep learning performance often follows predictable scaling trends with data, compute, and parameters (Kaplan et al., 2020; Hoffmann et al., 2022; Zhai et al., 2022). For brain alignment, evidence suggests improvements with model and data scale but with benchmark-dependent saturation (Antonello et al., 2023; Shen et al., 2025; Gokce & Schrimpf, 2025; Willeke et al., 2026), while behavioral alignment appears especially sensitive to training sample count (Muttenthaler et al., 2023; Lonnqvist et al., 2025). In decoding, performance can scale roughly log-linearly with additional recording time, driven primarily by within-subject data (Banville et al., 2025). Our contribution advances the state of the art with a scaling analysis spanning multiple recording modalities that jointly characterizes pretraining scale, neural fine-tuning, and model-to-brain mappings.

## 3. Methods

**Datasets.** We evaluate model–brain alignment on eight public neural datasets spanning macaque electrophysiology/spiking and human fMRI/MEG/EEG, with controlled and naturalistic image stimuli. Macaque data include *FreemanZiemba2013* (V1/V2; controlled stimuli; Freeman et al., 2013), *MajajHong2015* (V4/IT; natural images; Majaj et al., 2015), and *TVSD* (V1/V4/IT multi-unit activity; natural images; Papale et al., 2025). Human data include *THINGS-fMRI* and *THINGS-MEG* (Hebart et al., 2023), *THINGS-EEG1* and *THINGS-EEG2* (Grootswagers et al., 2022; Gifford et al., 2022, we exclude low-SNR subjects for a final count of 10 subjects from EEG1), and *NSD* (Allen et al., 2021). We preprocess each dataset with modality-appropriate standardized pipelines and estimate dataset-specific noise ceilings for noise-normalized predictivity scores. Table 1 summarizes datasets and abbreviations; further details are in Appendix A.

**Models.** We evaluate brain alignment across a diverse set of pretrained vision backbones spanning convolutional (e.g., ResNet/ConvNeXt), recurrent (CORnet-style), and transformer (ViT-style) architectures, trained with supervised classification, self-supervised learning (DINOv2/v3; Oquab et al., 2024; Siméoni et al., 2025), and image–text contrastive objectives (CLIP-style; Radford et al., 2021). Models are drawn from two complementary sources: (i) spvvs (Gokce & Schrimpf, 2025), a controlled scaling suite of 600+ task-optimized models with systematically varied pretraining scale (data, parameters, and compute) used for the primary scaling analyses; and (ii) selected timm models (Wightman, 2019), which extend coverage to larger scales and alternative pretraining regimes and serve as an external validation set. The full model inventory and pretraining details are provided in Appendix B (Table S2).

**Brain alignment and encoding models.** We measure model–brain alignment by how well features from a candidate layer $\ell$ predict neural responses under a standardized encoding-model pipeline (Fig. 1). For stimulus $\mathbf{x}$, the model yields features $\mathbf{Z}_\ell(\mathbf{x})$, and for each subject $s$ and ROI $r$ we fit a linear readout

$$\widehat{\mathbf{y}}_{r,s}(\mathbf{x}) = W_{r,s}\,\mathrm{vec}(\mathbf{Z}_\ell(\mathbf{x})) + \mathbf{b}_{r,s}, \qquad (1)$$

estimating parameters $W_{r,s}$ with ridge regression on the training split,

$$\min_{W_{r,s},\,\mathbf{b}_{r,s}} \sum_{\mathbf{x}\in\mathcal{D}_{\mathrm{train}}} \|\mathbf{y}_{r,s}(\mathbf{x}) - \widehat{\mathbf{y}}_{r,s}(\mathbf{x})\|_2^2 + \alpha\,\|W_{r,s}\|_F^2\,. \quad (2)$$

We select $\alpha$ by cross-validation on the training split and evaluate on held-out data. Predictivity is Pearson correlation between predicted and observed responses, averaged within ROI, and is noise-normalized using dataset-specific noise ceilings (Appendix A).

For the primary pretraining-scaling analyses, this simple readout intentionally measures linearly accessible neural structure rather than the predictive upper bound of a highly expressive decoder. Pearson correlation provides a common predictivity metric, and noise-ceiling normalization accounts for dataset-specific response reliability. We evaluate richer mappings separately to quantify additional gains from readout design.

**ROI–layer assignments.** For each architecture and benchmark ROI, we commit a single representative layer shared among subjects for downstream analyses. To avoid selection bias across candidate layers, we keep the test split held out and perform nested cross-validation on the training split: for each layer $\ell$, we fit ridge-regularized encoding models (Eq. (1)–(2)) using five-fold outer CV, selecting $\alpha$ within each training fold via an inner LOOCV loop.

To scale layer sweeps, we apply a fixed random projection (30,000 dims) to layer activations prior to fitting; projection robustness is analyzed in Appendix C. For each ROI and dataset, we compute noise-normalized predictivity per subject, average across subjects, and select the layer maximizing this cross-subject mean as the committed layer. Thus, the committed layer for a given ROI is shared across subjects within a dataset and used throughout the remainder of the paper. Additional analyses of cross-dataset/modal consistency and layerwise progression are in Appendix D.

**Scaling power-law curves.** We study the relation between alignment scores and computational resources spent on pretraining, and fit empirical scaling curves following Gokce & Schrimpf (2025), adapting standard scaling-law methodology (Kaplan et al., 2020; Hoffmann et al., 2022) to brain-alignment scores. Fits use linear predictivity from the `spvvs` suite, while `timm` models are held out for validation; we use 30,000-d random projections for tractable large-scale evaluation. We fit

$$S = E - A\left(X + 10^\lambda\right)^{-\alpha}, \qquad (3)$$

where $S$ is alignment, $X \in \{D, N, C\}$ denotes pretraining samples, parameters, or compute, and $(E, A, \alpha, \lambda)$ are fitted constants (we drop the offset when the reduced form fits better). We define pretraining FLOPs (a compute proxy) as the number of pretraining samples multiplied by the model's FLOPs per forward pass. Uncertainty is estimated via 1,000 bootstrap resamples, reporting 95% confidence intervals; further optimization details follow Gokce & Schrimpf (2025).

**Neural fine-tuning.** Next, we turn to resources spent on optimization with neural data. We fine-tune a visual backbone $f_\theta$ on paired stimulus–neural recordings while freezing subject- and ROI-specific mappings. Concretely, we first fit linear ridge readouts (Eq. (1)–(2); $\alpha$ via LOOCV per $(r, s)$), then optimize only $\theta$. Given a minibatch $B$ of neural–stimulus pairs, we minimize

$$\mathcal{L}_{\text{neural}} = \frac{1}{|B|} \sum_{\mathbf{x} \in B} \sum_{r,s} \frac{1}{p_{r,s}} \|\mathbf{y}_{r,s}(\mathbf{x}) - \widehat{\mathbf{y}}_{r,s}(\mathbf{x})\|_2^2, \quad (4)$$

with gradients propagated only through $\theta$. Training alternates neural minibatches with ImageNet minibatches, optimizing

$$\mathcal{L}_{\text{total}} = \lambda_{\text{img}} \mathcal{L}_{\text{img}} + \lambda_{\text{neural}} \mathcal{L}_{\text{neural}} \qquad (5)$$

where $\mathcal{L}_{\text{img}}$ is the ImageNet cross-entropy loss and $\lambda_{\text{img}}, \lambda_{\text{neural}} \geq 0$ are scalar weights. We evaluate without random projections: both the fine-tuned backbone and its corresponding initial backbone are evaluated using the original (unprojected) features, so alignment differences reflect representational changes rather than projection artifacts. We study data scaling by fine-tuning on subsets of neural samples (3 seeds per configuration; Appendix G.1).

**Scaling linear and attention probes.** To characterize how mapping performance scales with neural data, we subsample the available training set and fit one readout per subject and ROI. In addition to standard linear probes, we evaluate an attention-based probe that inserts a cross-attention module between backbone tokens and the readout (Fig. 1d). Given token features $\mathbf{Z}_\ell(\mathbf{x})$ from layer $\ell$, a cross-attention block $g_\phi$ with $M$ learned queries $\mathbf{Q}$ produces

$$\mathbf{H}_\phi(\mathbf{x}) = g_\phi(\mathbf{Q}, \mathbf{Z}_\ell(\mathbf{x})), \qquad (6)$$

followed by a subject- and ROI-specific linear head

$$\widehat{\mathbf{y}}_{r,s}(\mathbf{x}) = W_{r,s}\,\text{vec}(\mathbf{H}_\phi(\mathbf{x})) + \mathbf{b}_{r,s}. \qquad (7)$$

We consider single-subject probes and a multi-subject variant that shares $g_\phi$ across subjects while keeping subject/ROI-specific heads. We evaluate without random projections and average results over three seeds (Appendix K).

## 4. Results

### 4.1. Pretraining scaling improves alignment but exhibits clear saturation.

Our results indicate that scaling up pretraining compute, model size, and pretraining sample count leads to saturation in alignment across benchmarks and recording modalities (Fig. 2). We quantify these trends by fitting parametric scaling curves on `spvvs` models (dots) and validating the same functional trends on `timm` models (crosses), which follow the predicted trajectories despite being out-of-family. Across benchmarks, alignment increases systematically with pretraining FLOPs and sample count, but with pronounced diminishing returns at larger scales (Fig. 2a–b). Scaling parameters also improves alignment, but yields smaller marginal gains and saturates earlier than data/compute scaling (Fig. 2c), consistent with training data being the stronger driver of alignment in this regime. Saturation levels also vary across modalities: fMRI and electrophysiology benchmarks (TVSD-EP, NSD-fMRI) reach higher plateaus, whereas M/EEG benchmarks (e.g., T-EEG1) show more modest improvements and saturate at lower absolute alignment. The saturation effect is consistent across brain-alignment metrics beyond linear readouts; additional pretraining results using alternative metrics are reported in Appendix E.3.

### 4.2. Architecture effects are strongest at low pretraining scale, while trajectories converge with scaling

To examine how architecture interacts with scale, we trace alignment along model series trained with increasing pretraining compute (Fig. 3a) and compare architecture families across pretraining sample counts (Fig. 3b). Figure 3a shows that many architectures converge to similar alignment levels at high scale, despite being trained with substantially

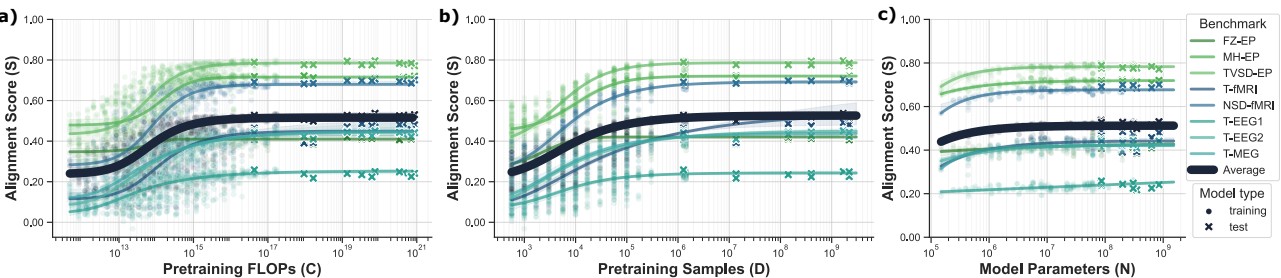

*Figure 2.* **Scaling pretraining improves brain alignment but saturates.** Brain alignment $S$ (noise-normalized Pearson-$r$; *Methods*) increases with **a)** pretraining compute FLOPs ($C$), **b)** pretraining samples $D$, and **c)** model parameters $N$, but ultimately plateaus across all benchmarks (colors). Scaling dataset samples yields larger gains than scaling parameters. We fit parametric scaling curves using the spvvs model set and treat timm models as held-out, showing that the trends generalize to larger scales and alternative pretraining objectives. Points show individual models; solid curves indicate fitted trends with uncertainty bands; markers denote model source.

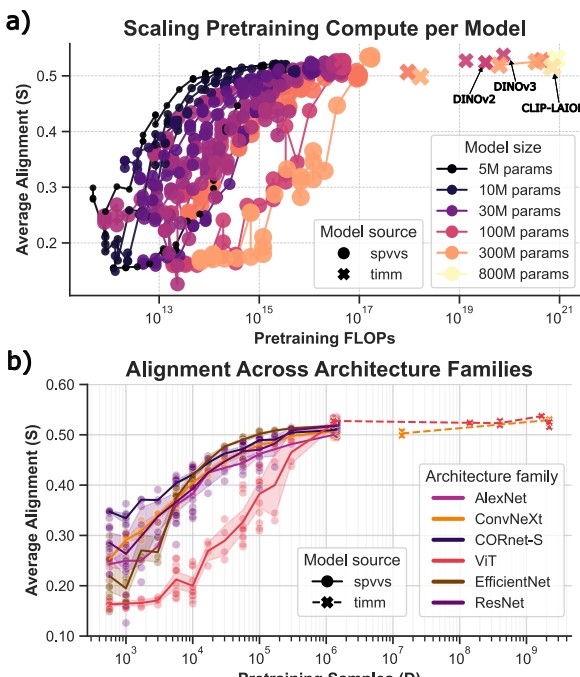

*Figure 3.* **Pretraining scale reduces architectural relevance.** **a)** Training trajectories across different model sizes converge to similar saturation levels. Markers are connected for the same architecture trained with larger data/compute budgets. **b)** Alignment differs markedly across architecture families at small scale, but these gaps shrink with more data and compute.

different compute budgets. Notably, several widely used pretrained models (e.g., DINOv2/v3 and CLIP; Oquab et al., 2024; Siméoni et al., 2025; Radford et al., 2021) attain comparable alignment despite large differences in model size and training data. In the low-scale regime, however, architecture choice strongly modulates alignment: families separate clearly at small sample counts, highlighting the importance of inductive biases when data is limited (Fig. 3b). For example, CORnet-S—a biologically inspired recurrent convolutional architecture (Kubilius et al., 2019)—achieves

high average alignment with minimal pretraining, whereas Vision Transformers lag in this regime. With increased pretraining, these gaps narrow, and ViT models can surpass CORnet-S at larger scales. Overall, increasing pretraining scale reduces architecture-dependent disparities and leads to convergence toward similar high-scale performance bands (Fig. 3a–b), suggesting that pretraining can partially compensate for architectural differences even as gains diminish near saturation.

### 4.3. Neural fine-tuning scales with data

We examine whether neural fine-tuning can overcome the saturation observed under pretraining scaling. To this end, we fine-tune task-optimized backbones using a hybrid objective that combines task and neural supervision (Sec. 3) while systematically varying the number of available neural training samples. As a base setup, we fine-tune a ViT-S backbone using IT responses for TVSD-EP and THINGS-fMRI, ventral ROIs for NSD-fMRI, and occipital–parietal channel groups for T-EEG1, T-EEG2, and T-MEG, pooling all available subjects. We exclude FZ-EP and MH-EP from fine-tuning due to their smaller scale. We assess the impact of these design choices (e.g., ROI selection and backbone architecture) in subsequent sections, and report additional analyses (including stimulus–response permutation controls and changes in task accuracy) in Appendix G.3.

Figure 4a shows that fine-tuning on paired stimulus–neural recordings yields consistent gains in brain alignment as the number of neural training samples increases. Here, we summarize alignment as the average noise-ceiled Pearson $r$ across benchmarks, subjects, and ROIs. Across datasets, alignment improves monotonically over the explored range with no clear evidence of saturation, suggesting that performance remains data-limited in these regimes. The main exception is T-EEG1, where fine-tuning slightly decreases alignment, consistent with its lower signal-to-noise ratio relative to the other datasets (Appendix A).

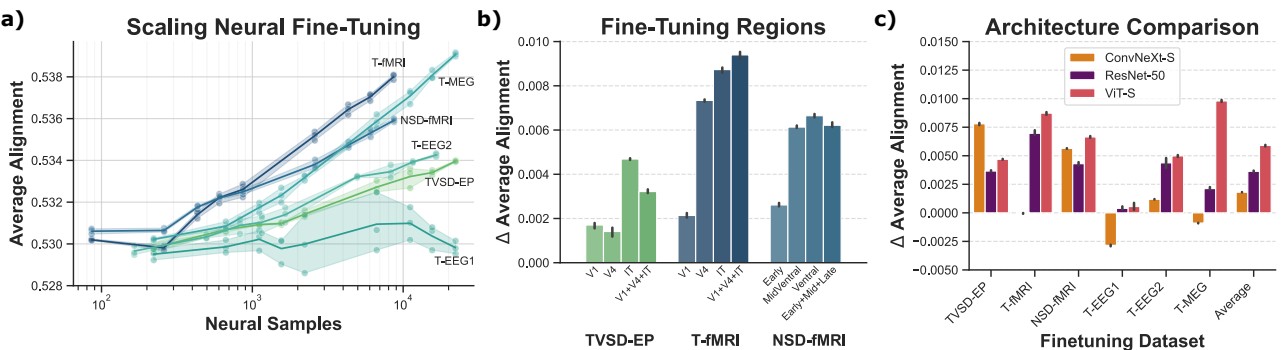

*Figure 4.* **Fine-tuning on neural data yields consistent gains in brain alignment. (a)** Brain alignment improves with the number of neural training samples, with no clear saturation at the size of current datasets. y-axis shows the average Pearson $r$ across all benchmarks and each line represents ViT-S models fine-tuned on one specific dataset. **(b)** Fine-tuning on progressively later visual ROIs produces larger alignment improvements. **(c)** Across datasets, ViT-S typically benefits more from neural fine-tuning than ResNet-50 and ConvNeXt-S.

Gains are largest for the fMRI datasets (NSD-fMRI and T-fMRI), followed by T-MEG. Despite its relatively high noise ceiling, TVSD-EP shows more moderate improvements, which may reflect differences in target coverage and the extent to which our static, temporally aggregated objective matches each modality. Overall, the strongest gains emerge at larger sample sizes, indicating that additional neural data can be effectively converted into improved predictivity via supervised fine-tuning.

### 4.4. Larger gains in higher-level visual ROIs

To localize where neural supervision is most effective, we fine-tune using targets from different ROIs along the visual hierarchy (Fig. 4b). Across both electrophysiology and fMRI benchmarks, supervising later ventral-stream regions yields larger alignment gains than supervising early visual areas.

In our setup, the fine-tuning loss for each ROI is applied at that ROI's committed layer (i.e., the layer mapped to that region). As a result, early-ROI supervision primarily constrains shallow representations, whereas higher-ROI supervision directly shapes deeper features. For the different alignment gains across hierarchy, we hypothesize two complementary factors underlying this pattern. First, losses applied at early layers may have a diluted impact on downstream representations because later layers can partially re-map or compensate for changes in early features; in contrast, constraining deeper layers can more directly influence the abstract representations that drive predictivity in higher ROIs. Second, later ventral ROIs may provide a more general and transferable supervision signal: compared to early visual areas, higher-level regions capture more abstract structure that is shared across stimuli and experimental paradigms, and thus may transfer more consistently across datasets and modalities when used as fine-tuning targets.

### 4.5. Fine-tuning benefits are stratified by architecture

Fine-tuning gains depend strongly on the backbone architecture (Fig. 4c). Across benchmarks, ViT-S shows the largest improvements in brain alignment in nearly all cases (with TVSD-EP as an exception), making it the most responsive backbone to neural supervision. In contrast, ConvNeXt-S and ResNet-50 benefit little from fine-tuning and often exhibit negligible or even negative changes relative to the pretrained baseline. Together, these results suggest that architectural choice materially shapes how effectively neural data can refine a representation. In practice, Vision Transformers are a strong default when the goal is to maximize the return on limited neural supervision. One plausible explanation is that their more flexible, data-driven representations adapt more readily to neural fine-tuning than the stronger inductive biases of convolutional architectures.

### 4.6. Neural fine-tuning transfers across datasets

Figure 5 shows the change in brain-alignment score after fine-tuning the visual backbone on paired stimulus–neural recordings from a single dataset (panel title), then evaluating the resulting model across all benchmarks (x-axis). The y-axis reports the change relative to the pretrained baseline.

Across most fine-tuning datasets, we observe predominantly positive transfer: optimizing the representation to better predict neural responses on one benchmark typically increases predictivity on other benchmarks as well, including across recording modalities. Although the gains are modest, they are consistent in magnitude (up to ∼0.01–0.02 absolute improvement), suggesting that neural fine-tuning nudges representations toward features that generalize across experimental settings rather than overfitting to a single dataset. Interestingly, cross-modal transfer can be stronger than within-modality transfer in some cases; for example, fine-tuning on NSD-fMRI improves T-EEG2 more than T-fMRI.

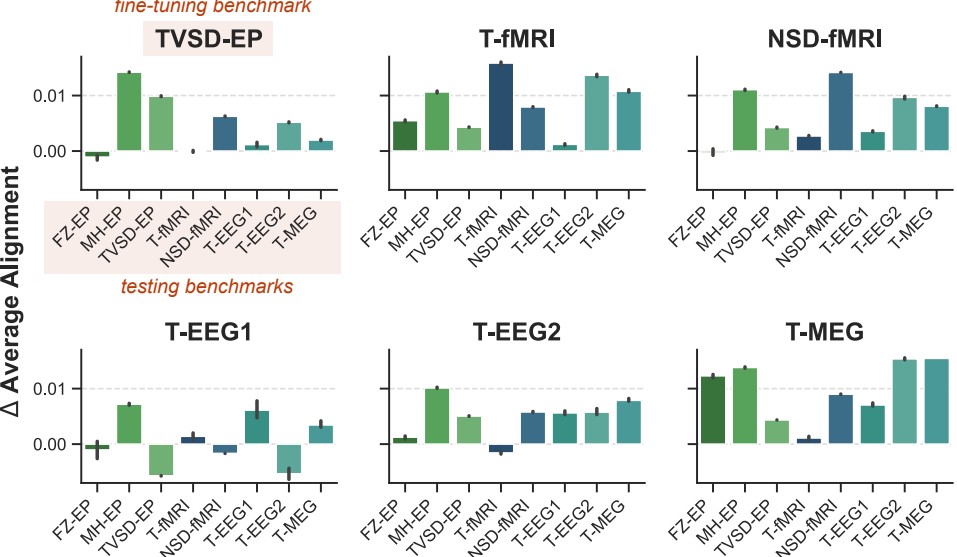

Figure 5. **Fine-tuning transfers across multimodal neural benchmarks.** For each panel, we fine-tune ViT-S on one dataset (panel title) and evaluate the change in brain alignment (y-axis) relative to the pretrained baseline across all benchmarks (x-axis). Fine-tuning often improves alignment on all benchmarks, across recording modalities (EP, fMRI, EEG/MEG).

In Appendix G.3, we confirm that these improvements are meaningful using a stimulus–recording permutation control, and we also document a trade-off between neural alignment gains and downstream task performance.

### 4.7. Neural-sample scaling is reliable across modalities

Figure 6 plots the alignment score ($S$) as a function of the number of stimulus–neural pairs used to fit the mapping ($F$, log scale) for each dataset. Across benchmarks, increasing $F$ yields consistent improvements, closely following an approximately log-linear trend over the explored range. Figure S62 reports the fitted parametric forms and their corresponding equations in detail.

For T-fMRI, T-EEG1, T-EEG2, and T-MEG, the fitted curves predict perfect saturation levels at $1.0$ (not shown in the figure), suggesting that alignment remains mapping-data-limited and could continue to improve with additional fitting stimuli. In contrast, TVSD-EP saturates above $\sim 0.9$ and NSD-fMRI around $\sim 0.8$, indicating earlier plateaus that may be better addressed by increasing mapping expressivity than by expanding the linear mapping dataset.

### 4.8. Attention-based readouts are competitive with linear mappings at full data

Table 2 compares three readout families trained on all available stimulus–neural pairs: a per-subject linear mapping (LINEAR–SS), a per-subject cross-attention mapping (ATTENTION–SS), and a multi-subject variant that shares a cross-attention block across subjects

with lightweight subject-specific heads (ATTENTION–MS). Overall, attention-based readouts are competitive with linear decoding at full data and often achieve the strongest alignment. In particular, ATTENTION–MS matches or exceeds LINEAR–SS on TVSD-EP, T-fMRI, NSD-fMRI, and T-EEG1, suggesting that a small amount of cross-attention can better capture non-linear structure in the mapping from model features to neural responses. At the same time, linear readouts remain strong on some benchmarks, indicating that the benefits of attention are not uniform across modalities and datasets.

### 4.9. Multi-subject sharing improves attention-based mapping with fewer parameters

Sharing the cross-attention module across subjects yields consistent gains: ATTENTION–MS outperforms ATTENTION–SS on every benchmark and achieves the best average alignment overall (Table 2). This pattern is consistent with the hypothesis that a substantial portion of the stimulus-to-brain mapping is shared across individuals and can be captured by a common attention block, while subject-/ROI-specific heads account for residual anatomical and measurement differences. Importantly, these gains come in a compact regime: both attention-based readouts use substantially fewer parameters than LINEAR–SS, and ATTENTION–MS is the most parameter-efficient of the three. We further characterize this parameter–performance trade-off in Appendix Fig. S63. Additional readout baselines (factorized low-rank and shallow MLP) match the parameter count of the attention probes but underperform both linear and attention readouts (Appendix J.1).

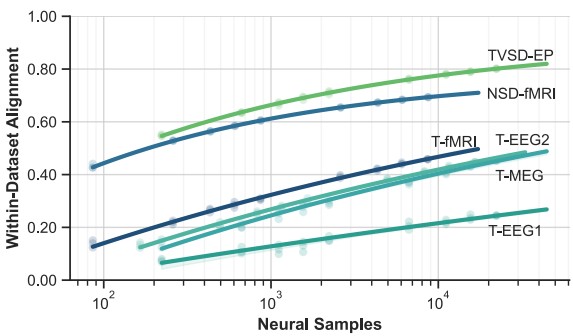

*Figure 6.* **Scaling neural samples for fitting a mapping.** For each dataset, increasing the number of stimulus–neural pairs used to fit a linear mapping yields consistent gains in alignment score for held-out benchmark samples. Points show empirical results, and solid curves show scaling-law fits $S(F) = E - AF^{-\alpha}$.

*Table 2.* Noise-ceiled Pearson $r$ across benchmarks for different mapping readouts. SS = single-subject; MS = multi-subject. Scores are averaged across subjects, ROIs, and three random seeds. Parameter counts are averaged across all instantiations of each probe type. Best per row is shown in **bold**.

| Benchmark | Linear (SS) | Attention (SS) | Attention (MS) |
|---|---|---|---|
| TVSD-EP | 0.801 | 0.844 | **0.846** |
| NSD-fMRI | 0.693 | 0.678 | **0.724** |
| T-fMRI | 0.458 | 0.461 | **0.487** |
| T-EEG1 | 0.245 | 0.234 | **0.257** |
| T-EEG2 | **0.447** | 0.390 | 0.434 |
| T-MEG | **0.453** | 0.410 | 0.414 |
| Average | 0.516 | 0.503 | **0.527** |
| Avg. #Parameters | $2.30 \times 10^8$ | $2.53 \times 10^7$ | $\mathbf{2.06 \times 10^7}$ |

related improvements in neural alignment accompany a reduction in task evaluation accuracy in this model set.

### 4.10. Pretraining task accuracy tracks neural alignment with diminishing returns

Figure 7a relates the task competence acquired during supervised pretraining to a model's ability to predict neural responses. Averaging $r_{\text{NC}}$ over ROIs and benchmarks gives a conservative model-level summary that cannot be attributed to a single ROI or favorable benchmark. ImageNet- and Ecoset-pretrained models show a consistent ordering: weaker validation accuracy corresponds to weaker mean alignment, and progressively more accurate models reach higher alignment levels. The compression among the highest-accuracy models mirrors the diminishing returns seen under pretraining scaling (Fig. 2), and the agreement across pretraining datasets suggests the association reflects general properties of successful object recognition rather than ImageNet-specific labels. In Appendix F, we provide a detailed dataset- and ROI-resolved analysis of the relationship between pretraining accuracy and brain predictivity.

### 4.11. Fine-tuning improves alignment at a small cost in task accuracy

We next ask whether fine-tuning gains in alignment co-occur with changes in task accuracy. For ViT-S, moving from the pretrained baseline to the full-data fine-tuned state reduces task accuracy from $0.783$ to $0.769$ while mean alignment rises from $r_{\text{NC}} = 0.529$ to $0.535$ (Fig. 7b). The same directional pattern holds for ViT-B, ViT-L, ConvNeXt-S, and ResNet-50, with the largest alignment increase for ViT-L ($0.539{\rightarrow}0.547$). Resolving the ViT-S sweep by fine-tuning dataset (Fig. 7c), task accuracy decreases in all six datasets, while alignment gains are strongest for T-MEG and T-fMRI, more moderate for NSD-fMRI, T-EEG2, and TVSD-EP, and negligible for T-EEG1. The aggregate trend is therefore not driven by any single dataset, although its magnitude is dataset-dependent. These figures show that fine-tuning-

### 4.12. LoRA retains most of the alignment gain of full fine-tuning

To assess whether full-weight updates are required, we fine-tune ViT-S with LoRA (Hu et al., 2022), adding trainable low-rank updates to the attention and feed-forward transformations while keeping the rest of the backbone fixed and allowing the classification head to adapt (details in Appendix I). Comparing a single LoRA configuration ($r = 32$, lr $10^{-4}$, 20 epochs) against full fine-tuning (20 epochs, lr $10^{-5}$) across all $48$ fine-tuning–evaluation pairs, full fine-tuning yields the larger mean alignment gain ($\Delta r_{\text{NC}} = 0.0059$ vs. $0.0042$), but LoRA exceeds it on $21/48$ pairs and matches it for several same- and cross-dataset transfers (Fig. 7d). LoRA's average is pulled down primarily by a large loss after fine-tuning on T-EEG1. A shared LoRA configuration thus recovers most of the alignment benefit of full fine-tuning at a fraction of the trainable parameters.

## 5. Discussion & Future Work

Our results provide a consistent, unified view on the scaling laws for models' multimodal brain alignment. Our key findings pertain to the training stages of building a brain model: ML pretraining, fine-tuning, and mapping.

*Pretraining* scale – across compute, (non-neural) data samples, and parameters – reliably improves brain alignment, but exhibits clear saturation (Fig. 2). At larger scales, training trajectories across architectures markedly converge toward similar plateaus (Fig. 3a), and pretrained models with higher validation accuracy reach correspondingly higher mean alignment with the same diminishing returns at the top (Fig. 7a). This convergence suggests that simply scaling conventional pretraining on natural images pushes models toward a common representational regime which is shared

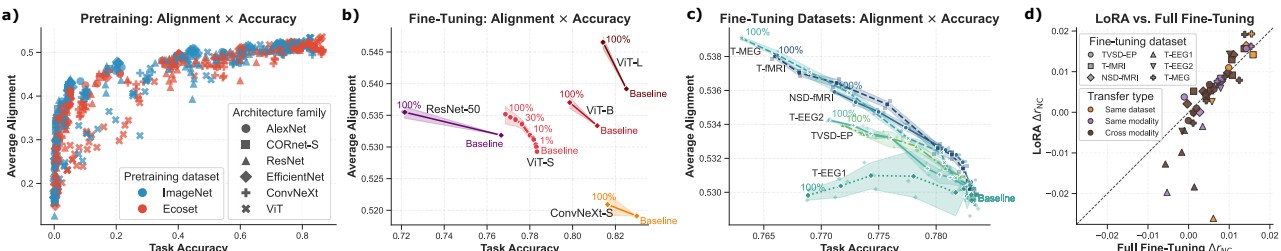

*Figure 7.* **Task accuracy and neural alignment across pretraining and fine-tuning.** **(a)** Mean alignment ($r_{NC}$, averaged over ROIs and benchmarks) vs. validation accuracy for ImageNet- and Ecoset-pretrained models. **(b)** Task accuracy vs. mean alignment for ViT-S across fine-tuning-data fractions, and for ViT-B/L, ConvNeXt-S, and ResNet-50 between baseline and full-data fine-tuned states (95% CI). **(c)** Same view for ViT-S, separated by fine-tuning dataset and data fraction. **(d)** Per-pair neural-alignment gain for LoRA vs. full fine-tuning across all 48 fine-tuning–evaluation transfers; LoRA remains competitive on many pairs while full fine-tuning is stronger on average.

across many architecture families ([Huh et al., 2024](#)). Architecture matters most when pretraining resources are scarce (Fig. 3b). In low-data regimes, families separate sharply, indicating that inductive bias can substitute for missing supervision. However, these differences shrink with scale, and larger pretraining budgets partially compensate for architectural choices, eventually yielding a narrower band of high-scale performance.

*Fine-tuning.* Our second key finding is that neural supervision consistently improves multimodal brain alignment beyond pretraining. Fine-tuning task-optimized backbones on paired stimulus–neural recordings improves alignment monotonically with neural sample count across datasets (except T-EEG1), with no saturation at the scale of current datasets (Fig. 4a). Moreover, the benefits generalize beyond the training benchmark: fine-tuning on a single dataset typically transfers positively across benchmarks, even across modalities (Fig. 5). These results suggest that the saturation observed under generic pretraining does not imply an intrinsic ceiling of the architecture alone, but rather limitations of the pretraining signal. Neural fine-tuning acts as a representation-shaping stage that can move models toward more benchmark-agnostic, biologically predictive features. These gains come at a small but consistent cost in task accuracy across architectures and fine-tuning datasets (Fig. 7b,c), and most of the alignment benefit can be recovered with parameter-efficient LoRA updates rather than full-weight fine-tuning (Fig. 7d).

*Mapping.* At the mapping level, our neural-sample scaling experiments show strong, reliable improvements as more stimulus–response pairs are available for fitting readouts, well-captured by a simple scaling law (Fig. 6, S62). In this context, attention-based probes offer a practical advantage: they are substantially more parameter-efficient than per-ROI linear maps while remaining competitive at full data, and our proposed multi-subject sharing consistently improves their performance (Tab. 2, Fig. S63). This supports the view that useful structure in stimulus-to-brain mappings is shared across subjects and can be captured with modest, structured

sharing rather than large, ROI-specific parameterization.

**Limitations.** First, our benchmarks only use *static* visual stimuli; we do not evaluate alignment for dynamic, naturalistic inputs (i.e., videos) where temporal computations, recurrence, and motion-sensitive pathways may play a larger role. Prior work has shown that video models can outperform static image models, particularly for predicting responses in motion-selective areas ([Tang et al., 2025](#); [Sartzetaki et al., 2025](#)). Second, our datasets rely on a single stimulus modality (vision), whereas some unexplained neural variance may reflect multisensory interactions that are not driven by visual input alone. Third, we largely abstract away temporal structure by averaging over time (EP), using single-trial beta estimates (fMRI), or flattening across time points (M/EEG). Incorporating temporally structured mappings and evaluation could better capture the dynamics of these modalities. Finally, our evaluation is benchmark-driven and correlational: improved predictivity alone does not establish mechanistic correspondence.

**Future work.** Our work suggests the development of *stronger decoder maps* as a promising direction. Training the mapping jointly on *multi-modal neural data* (e.g., fMRI + EEG/MEG + electrophysiology) should learn modality-invariant structure while remaining sensitive to modality-specific constraints (temporal resolution, spatial coverage, and noise). This could be combined with multi-subject and anatomy-aware parameter sharing to improve sample efficiency and generalization. More broadly, our results motivate training objectives that blend task and neural optimization in a multi-objective framework that maintains task performance while improving biological predictivity.

## Software and Data

We open-source our training and analysis code, along with benchmark results. All resources are available at: [https://github.com/epflneuroailab/multimodal-brain-scaling](https://github.com/epflneuroailab/multimodal-brain-scaling).

## Acknowledgements

This work was supported by the Swiss National Science Foundation (SNSF; Grant No. 10.003.772; A.G. and Y.T.) and the Swiss AI Initiative (2025 Fellowship Program; A.G.). We thank the members of the EPFL NeuroAI Lab for their valuable discussions and feedback throughout the project.

## Impact Statement

This work systematically characterizes how brain–model alignment changes with pretraining scale, neural fine-tuning, and the amount of neural data used to fit model-to-brain mappings across recording modalities. By identifying where scaling yields diminishing returns and where targeted supervision or improved mappings help most, our results support more data- and compute-efficient approaches to building brain-predictive models. The main broader impact is resource-related: large-scale training can be costly and energy-intensive. We do not anticipate ethical risks unique to this work beyond standard considerations around computational footprint and responsible use of public human/animal neuroscience datasets, which we use in accordance with their existing licenses and protocols.

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

*Table S1.* Overview of datasets used in this paper (all visual stimuli). We report concise dataset handles, noise ceilings, subject counts, ROI counts, and per-subject train/test split characteristics. Note that NSD-fMRI uses a subject-specific training set.

| Dataset | Modality | Handle | # Stimuli | Avg. Noise ceiling | # Subjects | # ROIs | Train samples | Test samples | Train:Test repetitions |
|---|---|---|---|---|---|---|---|---|---|
| Freeman et al. (2013) | Ephys | FZ-EP | 135 | 0.881070 | 1 | 2 | 121 | 14 | 20 : 20 |
| Majaj et al. (2015) | Ephys | MH-EP | 3,200 | 0.869264 | 2 | 2 | 2,880 | 320 | 50 : 50 |
| Papale et al. (2025) | Ephys | TVSD-EP | 22,248 | 0.976046 | 2 | 3 | 22,248 | 100 | 1 : 30 |
| Hebart et al. (2023) | fMRI | T-fMRI | 8,640 | 0.506084 | 3 | 28 | 8,640 | 100 | 1 : 12 |
| Allen et al. (2021) | fMRI | NSD-fMRI | 70,000 | 0.551898 | 8 | 28 | ~9,000 | ~1,000 | 3 : 3 |
| Hebart et al. (2023) | MEG | T-MEG | 22,248 | 0.350308 | 4 | 7 | 22,248 | 200 | 1 : 12 |
| Grootswagers et al. (2022) | EEG | T-EEG1 | 22,248 | 0.303142 | 10 | 7 | 22,248 | 200 | 1 : 12 |
| Gifford et al. (2022) | EEG | T-EEG2 | 16,740 | 0.490773 | 10 | 7 | 16,540 | 200 | 4 : 80 |

# A. Datasets and Preprocessing

## A.1. Natural Scenes Dataset

The Natural Scenes Dataset (NSD) (Allen et al., 2021) is a large-scale, high-resolution 7T fMRI dataset designed to bridge cognitive neuroscience and artificial intelligence through extensive sampling of visual and memory-related brain responses. Eight participants each viewed over 9,000 natural scenes, amounting to more than 70,000 unique images across all participants, across 30–40 scan sessions while performing a continuous recognition task. The dataset is distinguished by its high spatial resolution (1.8 mm isotropic), rigorous participant screening, and an advanced analysis pipeline incorporating voxel-specific hemodynamic response functions (HRFs), data-driven denoising via GLMdenoise (Kay et al., 2013), and ridge regression for robust single-trial response estimation.

The NSD is released in multiple spatial representations (volumetric and surface-based; native and group-level) and with several preprocessing pipelines of increasing sophistication for estimating single-trial beta responses, referred to as `b1`, `b2`, and `b3`. In our experiments, we use the `b3` beta estimates, which combine voxel-wise HRF fitting using a library of HRFs with GLMdenoise and ridge regularization, yielding the highest signal-to-noise ratio and most reliable single-trial estimates.

We use data in the `func1pt8mm` space, corresponding to each participant's native volumetric functional space. This choice is motivated by computational considerations, as the number of vertices in the `nativesurface` and `fsaverage` surface spaces is substantially larger.

For each subject, we use the provided region-of-interest (ROI) masks to select voxels and apply a 10% noise-ceiling threshold. Noise ceilings are computed from the released `ncsnr` files following the procedure described in (Allen et al., 2021). We restrict voxel selection to voxels within the `nsdgeneral` ROI to focus analyses on reliably visually responsive cortex. Within this constraint, we include the following anatomically defined and functionally localized ROIs in our analyses: `early`, `midventral`, `midlateral`, `midparietal`, `ventral`, `lateral`, `parietal`, `V1v`, `V1d`, `V2v`, `V2d`, `V3v`, `V3d`, `hV4`, `nsdgeneral`, `OWFA`, `VWFA-1`, `VWFA-2`, `OPA`, `PPA`, `RSC`, `OFA`, `FFA-1`, `FFA-2`, `EBA`, `FBA-1`, and `FBA-2`. Noise-ceiling distributions for these regions are reported in Figure S1.

Each stimulus was nominally presented three times per subject; however, due to incomplete scan sessions for four participants (`subject3`, `subject4`, `subject6`, and `subject8`), some stimuli were presented only once or twice. We use all completed scans from 8 subjects. In addition to the subject-specific set of unique stimuli, the dataset includes a separate set of 1,000 images that were viewed by all subjects. We split the data into training and test sets, where the training set consists of the subject-specific unique stimuli and the test set consists of the shared, disjoint image set. Within each session, voxel responses are z-score standardized. For all analyses, voxel responses are averaged across available repetitions.

## A.2. THINGS-fMRI

The THINGS fMRI dataset provides a densely sampled, event-related collection of brain responses from three participants, each viewing 8,640 naturalistic object images spanning 720 distinct object concepts (12 exemplars per concept) across 12 scan sessions (Hebart et al., 2023). Individually molded head casts were used to minimize head motion, and an advanced preprocessing pipeline, including ICA-based denoising and voxel-wise hemodynamic response function (HRF) modeling, was applied. Together, these design choices yield high spatial reliability and signal-to-noise ratio, enabling precise estimation of BOLD responses across early visual, ventral visual, and category-selective cortical regions.

We use the preprocessed version of the dataset released by the authors, which is provided in native volumetric space for each subject. To select reliable voxels, we use the provided `nc_testset` noise-ceiling estimates and apply a 10% noise-ceiling

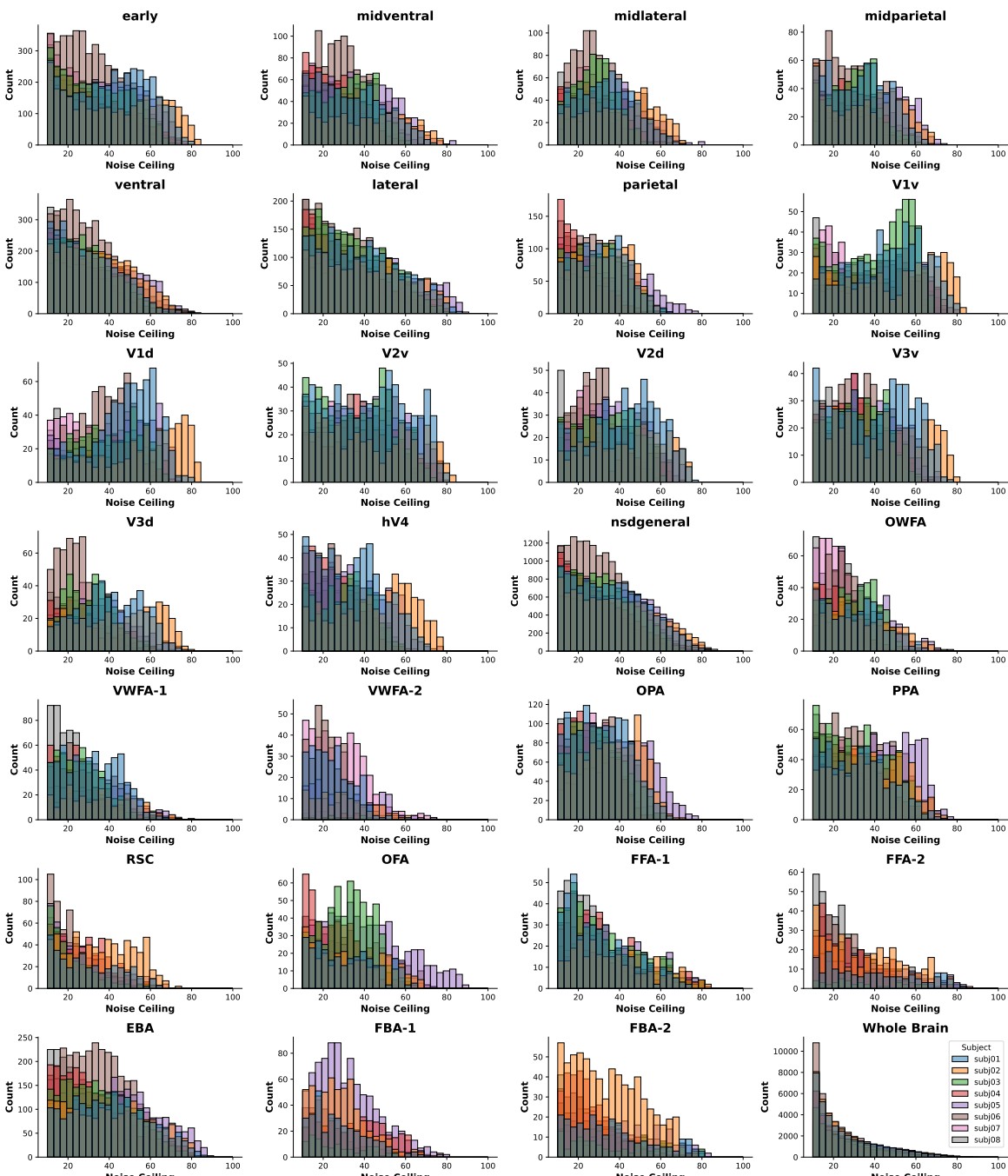

*Figure S1.* Noise-ceiling histograms across ROIs and subjects for NSD in `func1pt8mm` space. The `Whole Brain` ROI denotes the set of all voxels passing the noise-ceiling threshold.

threshold. Analyses are restricted to the following regions of interest (ROIs) included with the dataset release: `V1`, `V2`, `V3`, `V4`, `VO1`, `VO2`, `LO1`, `LO2`, `TO1`, `TO2`, `V3a`, `V3b`, `lEBA`, `rEBA`, `lFFA`, `rFFA`, `lOFA`, `rOFA`, `lPPA`, `rPPA`, `lRSC`, `rRSC`, `lTOS`, `rTOS`, `lLOC`, `rLOC`, and `IT`.

The dataset provides disjoint training and test splits: training images are presented once, whereas test images are repeated 12 times, enabling reliable noise-ceiling estimation. We use the training set for model layer selection and for fitting decoders from model representations to brain responses, and report performance on the test set. Noise-ceiling distributions for these regions are shown in Figure S2. Voxel responses are z-score standardized across stimuli on a per-voxel basis. For evaluation,

voxel responses in the test set are averaged across repetitions.

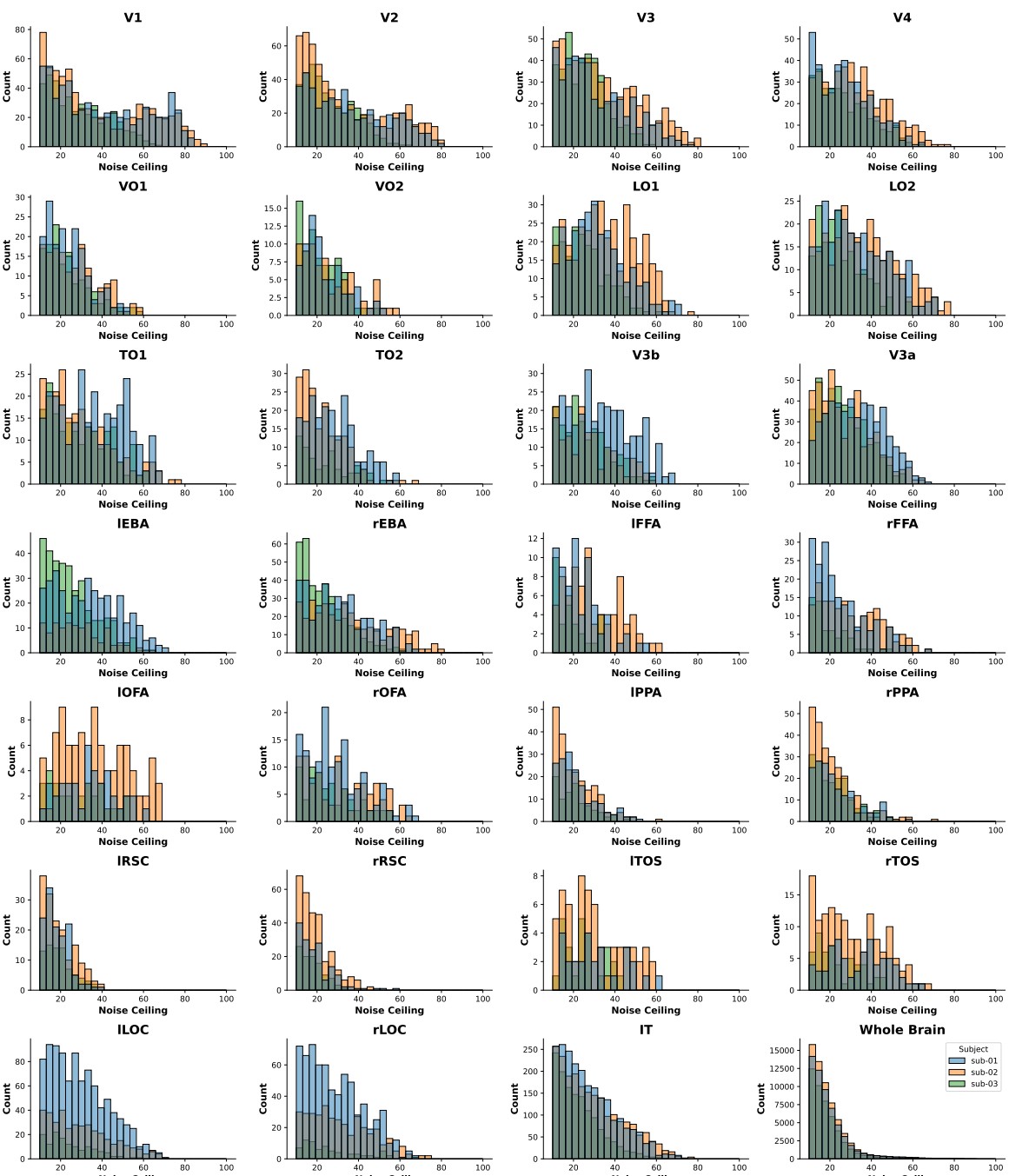

*Figure S2.* Noise-ceiling histograms across ROIs and subjects for THINGS fMRI dataset. The `Whole Brain` ROI denotes the set of all voxels passing the noise-ceiling threshold.

## A.3. THINGS-EEG1

The THINGS-EEG1 dataset comprises large-scale human EEG recordings acquired during rapid serial visual presentation (RSVP) of object images, enabling dense temporal sampling across a broad object space. Fifty participants viewed 22,248 images corresponding to 1,854 object concepts from the THINGS stimulus set, presented at 10 Hz in a single ~1-hour session (Grootswagers et al., 2022). Each image was shown once, with an additional set of 200 test images repeated 12

times to support reliability and noise-ceiling estimation.

We preprocess the EEG data using the MNE-Python package (Gramfort et al., 2013), closely replicating the correction steps described by the dataset authors, originally implemented in MATLAB. Continuous signals are high-pass filtered at 0.1 Hz, epoched, and epochs are resampled to 100 Hz using an anti-aliasing filter. We apply baseline correction using a $[-0.2, 0]$ s pre-stimulus interval and crop epochs to the $[0, 0.8]$ s post-stimulus window. Two of the 50 subjects failed during preprocessing and were excluded from further analysis.

Noise ceilings are computed on the test set following the procedure described in (Allen et al., 2021). Although the dataset includes a relatively large number of participants, noise-ceiling values vary substantially across subjects, with many exhibiting low reliability. To exclude low-quality recordings and to reduce computational cost, we select the top 10 subjects based on mean noise ceiling in occipital and parietal channels. For all subsequent analyses, neural responses are averaged across repeated presentations in the test set. Noise ceilings across subjects and ROIs are shown in Fig. S3.

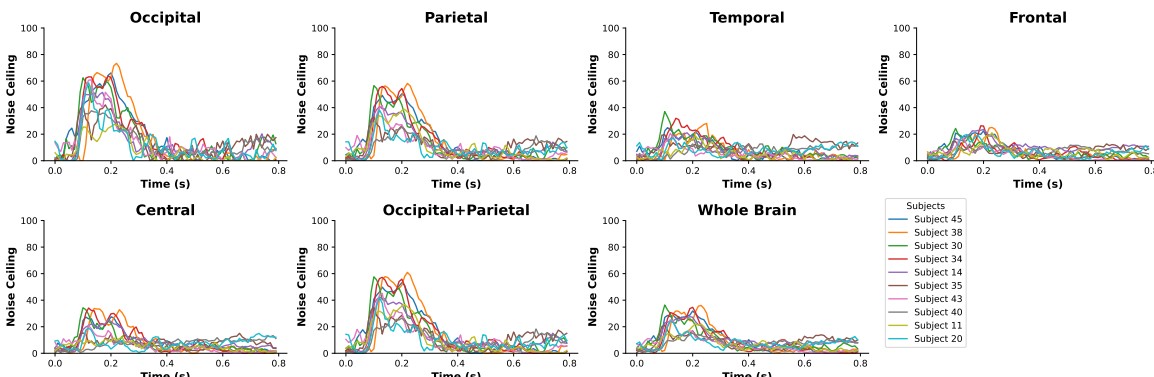

*Figure S3.* Time-resolved noise-ceiling curves averaged across channels, shown for each subject and ROI in the THINGS-EEG1 dataset. The `Whole Brain` ROI includes all available EEG channels, while `Occipital+Parietal` comprises channels located over occipital and parietal scalp regions.

### A.4. THINGS-EEG2

THINGS-EEG2 (Gifford et al., 2022) is a large-scale EEG dataset designed for training and evaluating computational models of human visual object recognition with high temporal resolution. The dataset includes recordings from 10 participants, each completing 82,160 trials acquired over four RSVP sessions (stimulus onset asynchrony: 200 ms) using images from the THINGS stimulus set. Stimuli are split into a non-overlapping training partition of 16,540 image conditions (1,654 concepts $\times$ 10 images; each repeated 4$\times$) and a held-out test partition of 200 image conditions (200 concepts $\times$ 1 image; each repeated 80$\times$), enabling unbiased training and evaluation without concept overlap. EEG was recorded with a 64-channel 10–10 montage at 1 kHz, and the release provides both raw and preprocessed data.

The released preprocessed version of THINGS-EEG2 includes only occipital and parietal channels and applies an additional whitening step, which increases signal-to-noise ratio and inflates noise-ceiling estimates. Because this preprocessing choice is not applied to the other datasets used in our study, we avoid whitening to ensure fair and consistent comparisons across datasets. Accordingly, we start from the source EEG recordings and perform our own preprocessing, closely following the pipeline described in (Gifford et al., 2022).

We epoch the continuous data and resample the resulting epochs to 100 Hz using an anti-aliasing low-pass filter. Note that during acquisition, the data were already band-pass filtered between 0.1 and 100 Hz. Noise ceilings are computed per subject, per channel, and per time point across image conditions and repetitions, following the procedure described in (Allen et al., 2021). Unlike the released preprocessed data, we do not apply whitening. Noise-ceiling time courses are shown in Fig. S4. For all subsequent analyses, neural responses are averaged across repetitions within both the training and test splits.

### A.5. THINGS-MEG

The THINGS-MEG dataset provides large-scale human magnetoencephalography recordings with high temporal resolution for studying visual object representations across a broad semantic space (Hebart et al., 2023). Four participants viewed

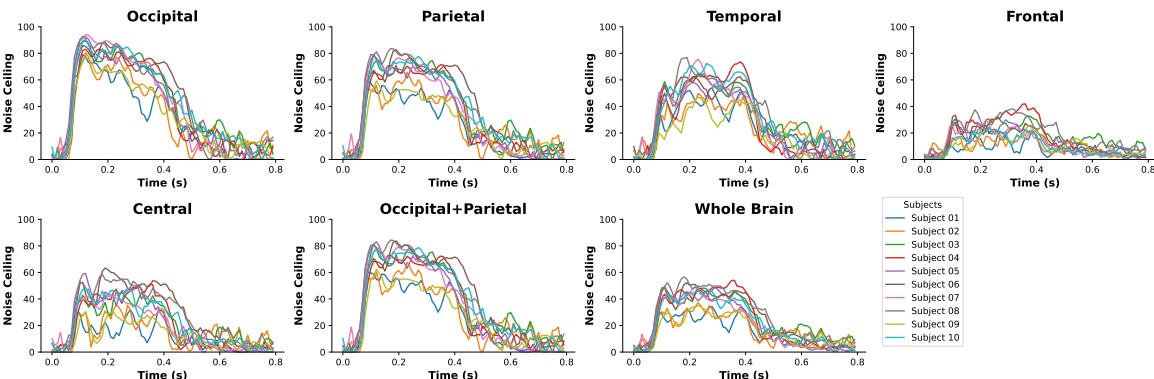

*Figure S4.* Per time point noise-ceiling plots averaged across channels shown across subjects and ROIs for THINGS EEG2 dataset. The `Whole Brain` ROI denotes the set of all available channels and `Occipital+Parietal` is the collection of channels from occipital and parietal channels.

22,448 naturalistic images spanning 1,854 object concepts from the THINGS stimulus set, collected across 12 sessions using a fast event-related design (stimulus onset asynchrony: $1.5 \pm 0.2$ s). A subset of 200 images was repeated in each session to support reliability analysis, noise-ceiling estimation, and held-out model evaluation.

For the MEG data, we use the epoched signals provided by the dataset authors. The authors applied a band-pass filter with cutoffs at 0.1 Hz and 40 Hz, resampled the data to 200 Hz, and epoched the signals with respect to image onset in the interval $[-0.1, 1.3]$ s. Baseline correction was applied using the $[-0.1, 0]$ s pre-stimulus window, where 0 denotes stimulus onset. We further resample the epoched data to 100 Hz and crop the signals to the $[0, 1.3]$ s post-stimulus interval.

Noise ceilings are computed following the same procedure as the dataset authors, based on the analytical approach described in (Allen et al., 2021), and are shown in Fig. S5. For all subsequent analyses, neural responses are averaged across repeated presentations in the test set.

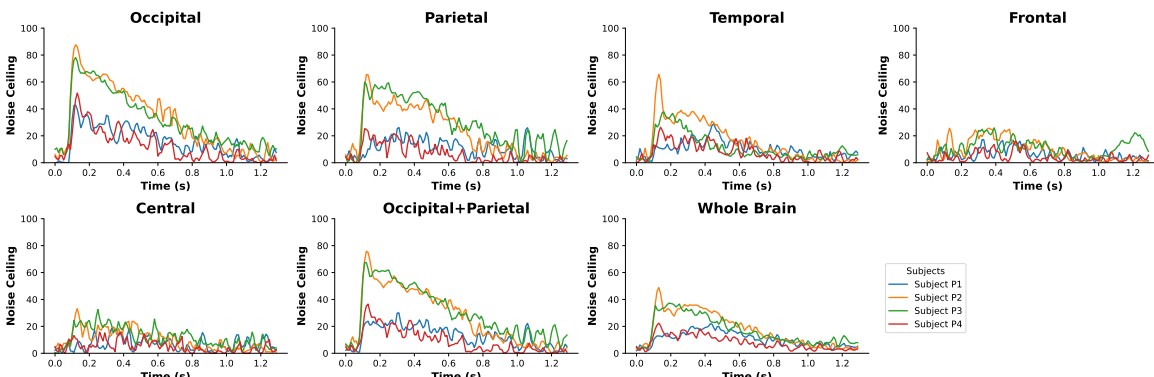

*Figure S5.* Per time point noise-ceiling plots averaged across channels shown across subjects and ROIs for THINGS MEG dataset. The `Whole Brain` ROI denotes the set of all available channels and `Occipital+Parietal` is the collection of channels from occipital and parietal channels.

### A.6. THINGS Ventral-stream Spiking Dataset

The THINGS Ventral-stream Spiking Dataset (TVSD) provides large-scale intracortical electrophysiology recordings from the primate visual ventral stream, enabling high-resolution analyses of neuronal tuning and population dynamics (Papale et al., 2025). Multi-unit spiking activity was recorded from two macaque monkeys using high-channel-count Utah array implants (1,024 electrodes in total) chronically placed in primary visual cortex (V1), area V4, and inferotemporal cortex (IT). Monkeys viewed 25,248 natural images from the THINGS stimulus set, comprising a training partition spanning 1,854 object concepts (12 images per concept, shown once) and a held-out test set of 100 images repeated 30 times to support reliability estimation and cross-validation.

We use the normalized multi-unit activity (MUA) provided with TVSD, which is preprocessed by the authors by z-scoring responses within session and averaging firing rates within an analysis window centered on each site's response peak. We select recording sites using the released `reliability` field, retaining channels with reliability more than 10%. We compute noise-ceiling estimates following (Allen et al., 2021), shown in Fig. S6. For all analyses on the repeated-image test set, responses are averaged across repetitions.

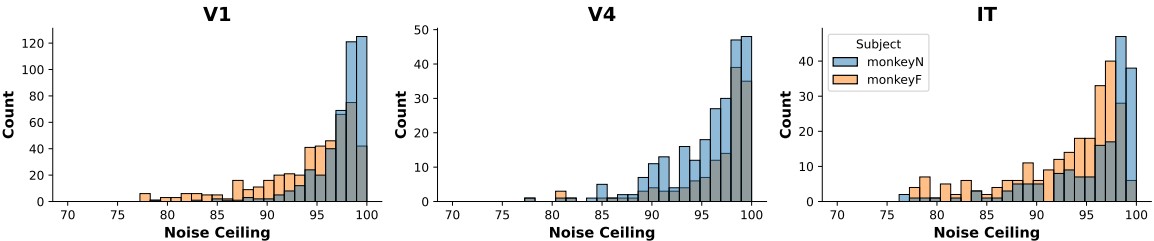

*Figure S6.* Noise-ceiling histograms across ROIs and subjects for THINGS Ventral-stream Spiking Dataset.

### A.7. FreemanZiemba2013

FreemanZiemba2013 (Freeman et al., 2013) is a macaque visual cortex electrophysiology dataset built around a tightly controlled ensemble of synthetic texture stimuli. Extracellular single-unit spiking activity was recorded from areas V1 and V2 across multiple anesthetized macaques using microelectrodes. The stimulus set is organized into texture families, where each family contains naturalistic texture samples synthesized to match higher-order statistics of an original texture photograph, and spectrally matched noise control images generated by phase randomization that preserves the power spectrum while removing higher-order structure. Images were presented briefly (100 ms) with an inter-stimulus gray interval (100 ms) and repeated across trials, enabling trial-averaged firing-rate time courses and summary response measures suitable for encoding analyses, cross-condition comparisons, and reliability/noise-ceiling estimation across neurons and areas.

We use the FreemanZiemba2013 dataset version distributed via the Brain-Score platform (Schrimpf et al., 2018). This release does not provide subject identifiers for individual neurons; accordingly, we treat all recorded units as originating from a single pooled subject. We compute noise ceilings following the procedure of (Allen et al., 2021) and retain channels with noise ceilings exceeding 10%. The resulting noise-ceiling distribution is shown in Fig. S7. As this release does not define an explicit train–test split, we randomly designate 10% of image conditions as a held-out test set stratified by texture type and use the remaining images exclusively for layer selection and decoder fitting. Neural responses are averaged over the 50–200 ms post-stimulus time window and across stimulus repetitions. We note that this dataset is comparatively small, comprising 135 image conditions, and consists of spike-sorted single-unit recordings.

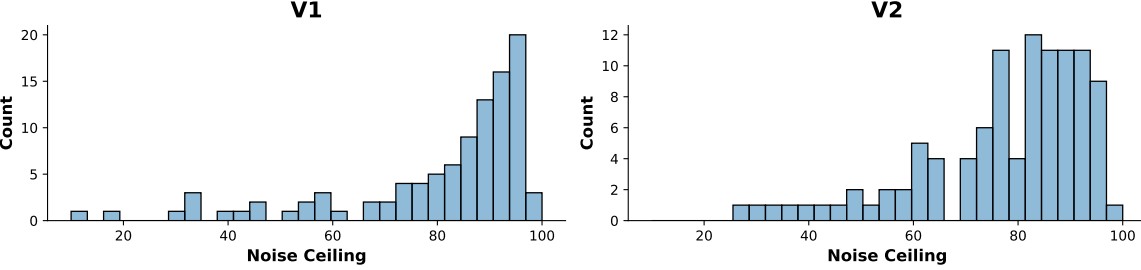

*Figure S7.* Noise-ceiling histograms across ROIs for FreemanZiemba2013 dataset.

### A.8. MajajHong2015

The MajajHong2015 dataset (Majaj et al., 2015) provides large-scale electrophysiological recordings from the primate ventral visual stream collected during controlled object recognition experiments. Neural activity was recorded from macaque visual areas V4 and inferior temporal (IT) cortex using chronically implanted multielectrode arrays, while animals passively viewed a large set of images. The stimulus set consists of 5,760 grayscale images rendered from 64 three-dimensional object models spanning eight basic-level categories, with systematic variation in position, scale, and pose, and superimposed

on randomized natural backgrounds. Each image was presented for 100 ms and repeated multiple times (typically ∼50), enabling reliable estimation of stimulus-evoked responses. Neural responses are provided primarily as multi-unit activity (MUA), computed as spike counts within fixed post-stimulus time windows aligned to response onset, and normalized per recording session.

We use the version of the dataset released on the Brain-Score platform (Schrimpf et al., 2018). Noise ceilings are computed following the procedure of (Allen et al., 2021), and we retain channels with noise ceilings exceeding 10%. The resulting noise-ceiling distribution is shown in Fig. S8. As this release does not define an explicit train–test split, we randomly designate 10% of image conditions as a held-out test set stratified by category and use the remaining images exclusively for layer selection and decoder fitting. Neural responses are averaged across stimulus repetitions.

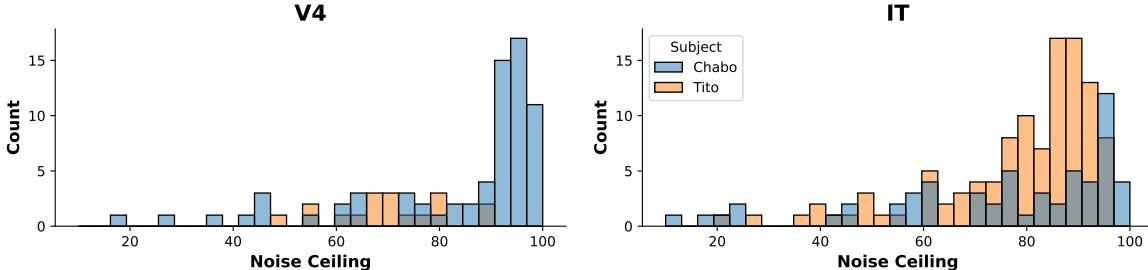

*Figure S8.* Noise-ceiling histograms across ROIs and subjects for MajajHong2015 dataset.

## B. Model inventory and pretraining regimes

We evaluate brain alignment using pretrained vision models drawn from two complementary model zoos (Table S2). Our primary suite is the controlled scaling collection introduced in Gokce & Schrimpf (2025) (spvvs), which contains 600+ task-optimized models spanning multiple architecture families, including AlexNet (Krizhevsky et al., 2012), ResNet (He et al., 2016), EfficientNet (Tan & Le, 2019), CORnet (recurrent) (Kubilius et al., 2019), ConvNeXt (Liu et al., 2022), and Vision Transformers (Dosovitskiy et al., 2021). These models are trained primarily with supervised classification objectives while systematically varying pretraining scale—including the number of training samples $D$, model capacity $N$, and total training compute $C$—making spvvs well suited for estimating empirical scaling relationships between pretraining and brain alignment.

To complement this controlled suite, we additionally evaluate selected models from the timm repository (Wightman, 2019), which broadens coverage to larger scales and alternative objectives and datasets. In particular, our timm subset includes ConvNeXt and ViT checkpoints trained with supervised ImageNet pretraining, image–language contrastive supervision, as well as self-supervised DINO-style models (including DINOv2 (Oquab et al., 2024) and DINOv3 (Siméoni et al., 2025)). We use spvvs for fitting scaling curves in the main text and treat the timm models as an external validation set that probes generalization to architectures and pretraining regimes outside the controlled scaling suite.

Table S2 summarizes the model inventory by source and architecture family. We report the number of evaluated checkpoints (*Runs*) and the number of unique training configurations (*Configs*), where configurations collapse runs that differ only by random seed when a seed is encoded in the model identifier. We additionally report the number of distinct architecture variants within each family (*Archs*), the number of distinct parameter scales (*Sizes*), and the coverage of pretraining scale in terms of parameter count $N$, pretraining samples $D$, and an estimated compute proxy $C$ derived from available model metadata. Objective counts are summarized using coarse labels (SUP: supervised classification; SSL: self-supervised; I–L: image–language contrastive); for timm, these labels are inferred from model identifiers and/or pretraining dataset tags.

**Compute proxy.** To enable consistent compute-scale comparisons across model sources (timm and spvvs), we use a unified compute proxy defined as

$$C \triangleq D \times \text{FLOPs}_{\text{fwd}}, \tag{8}$$

i.e., the product of the reported pretraining dataset size $D$ and the model's floating-point operations per forward pass. Using this proxy throughout avoids dependence on heterogeneous training pipelines and removes the need to reconstruct full training procedures when estimating total training compute.

*Table S2.* **Model inventory by backbone source and architecture family.** *Runs* correspond to distinct checkpoints; *Configs* collapse runs that only differ by random seed (when available). $N$ is parameter count (millions), $D$ is the number of pretraining samples, and $C$ is an estimated compute proxy from the metadata. Objective labels: SUP = supervised classification, SSL = self-supervised, I–L = image–language contrastive.

| Source | Family | Runs | Configs | Archs | Sizes | $N$ (M) | $D$ | $C$ (est.) | Pretrain | Obj. |
|---|---|---|---|---|---|---|---|---|---|---|
| spvvs | AlexNet | 30 | 14 | 1 | 1 | 61.1 | 565–1.4M | $8.07\times10^{11}$–$2.06\times10^{15}$ | ecoset, imagenet | SUP:30 |
| spvvs | CORnet-S | 30 | 14 | 1 | 1 | 53.4 | 565–1.4M | $1.86\times10^{13}$–$4.76\times10^{16}$ | ecoset, imagenet | SUP:30 |
| spvvs | ConvNeXt | 128 | 64 | 8 | 8 | 0.579–198 | 565–1.4M | $5.05\times10^{12}$–$9.94\times10^{16}$ | ecoset, imagenet | SUP:128 |
| spvvs | EfficientNet | 87 | 42 | 3 | 3 | 5.29–9.11 | 565–1.4M | $4.77\times10^{11}$–$3.39\times10^{15}$ | ecoset, imagenet | SUP:87 |
| spvvs | ResNet | 217 | 137 | 39 | 38 | 0.149–181 | 565–1.4M | $2.06\times10^{12}$–$7.99\times10^{16}$ | ecoset, imagenet | SUP:217 |
| spvvs | ViT | 128 | 64 | 8 | 8 | 0.436–304 | 565–1.4M | $1.22\times10^{12}$–$1.73\times10^{17}$ | ecoset, imagenet | SUP:128 |
| timm | ConvNeXt | 4 | 4 | 4 | 4 | 198–846 | 14M–2.2B | $9.36\times10^{17}$–$8.72\times10^{20}$ | in22k, laion2b | SUP:2, I–L:2 |
| timm | ViT | 8 | 8 | 7 | 7 | 85.6–632 | 1.3M–2.2B | $4.32\times10^{16}$–$7.13\times10^{20}$ | in1k, laion2b, lvd142m, lvd1689m, openai | SUP:1, SSL:3, I–L:4 |

## C. Robustness to Random Feature Projection

To reduce the computational and memory cost of fitting encoding models, we apply a fixed random projection to the model features before fitting the mapping for layer assignment and the pretraining scaling analyses (Figure 2). Figure S9 shows that projecting to 30k dimensions yields essentially identical alignment to using the full feature vectors across architectures and benchmarks. To characterize sensitivity to projection size, Figure S10 sweeps projection dimensionality for DeiT-S: alignment improves rapidly at low dimensions and then plateaus, with ~30k–50k dimensions matching the no-projection baseline. Based on this robustness, we use a 30k-dimensional projection as the default when scaling analyses to many models and datasets. Random projections are not applied for neural fine-tuning (Figures 4–5) or for mapping comparisons (Figure 6, Table 2).

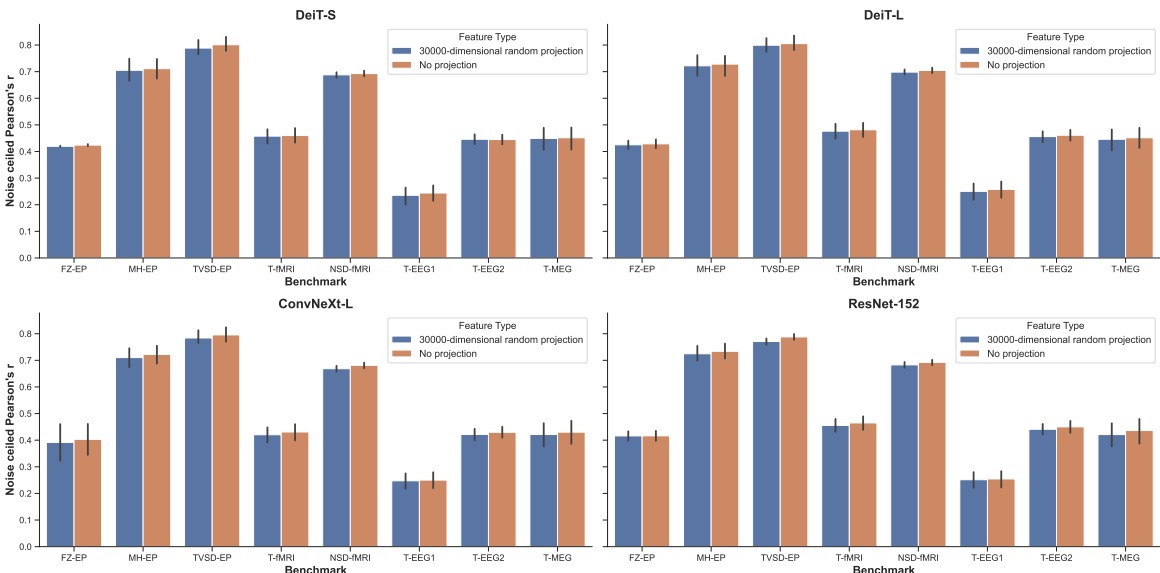

*Figure S9.* Random feature projections preserve alignment across architectures and benchmarks. Noise-ceiled Pearson $r$ for four backbones (DeiT-S/L, ConvNeXt-L, ResNet-152) evaluated on each benchmark, comparing a 30k-dimensional random projection of the feature vectors to using the full features. Scores are nearly unchanged across modalities, indicating that random projections can reduce feature dimensionality without materially affecting alignment.

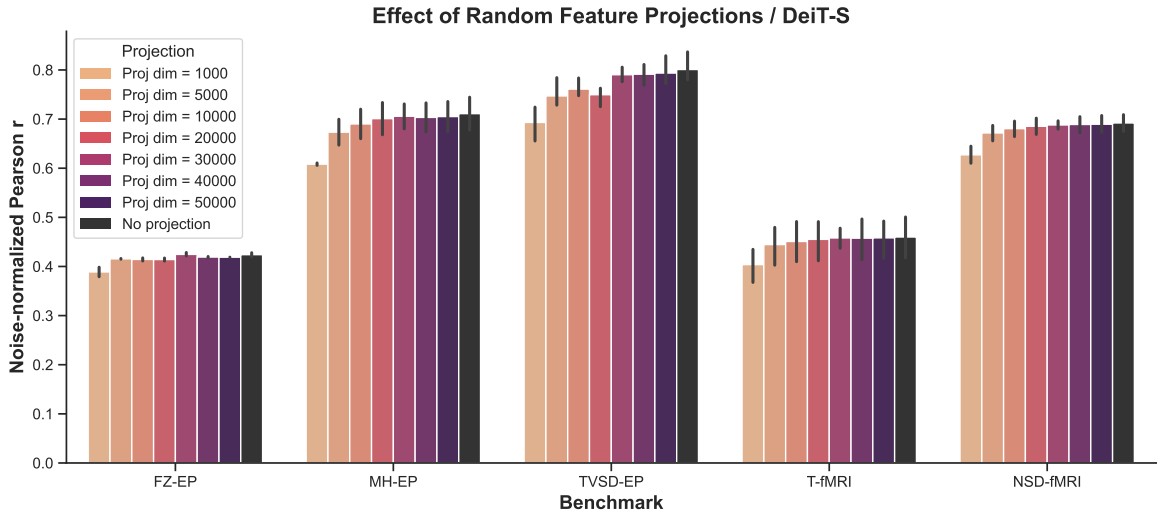

*Figure S10.* Alignment is stable across projection dimensionalities (DeiT-S). Noise-normalized Pearson $r$ as a function of random projection dimension for five benchmarks, including the no-projection baseline. Performance improves rapidly with dimension and then saturates, with ~30k–50k dimensions matching the no-projection scores.

## D. Layer–ROI Hierarchy and Cross-Dataset Consistency

**Cross-dataset consistency of the layer–ROI hierarchy.**    To assess whether the relative depth at which models best align with different stages of the visual hierarchy is stable across datasets, we compared layer-wise predictivity profiles between NSD-fMRI and two independent benchmarks (THINGS-fMRI and TVSD-EP). Because ROI taxonomies differ across datasets, we map dataset-specific ROIs into three coarse super-ROIs that capture successive stages of the visual hierarchy: *Early* (primary/secondary visual cortex; e.g., V1/V2 and dataset-specific variants, including NSD's composite `early`), *Mid* (intermediate visual areas; e.g., V3/V4 and variants, including NSD composites such as `midventral`, `midlateral`, and `midparietal`), and *High* (all remaining higher-level ROIs not assigned to Early or Mid). This mapping yields a common partition that enables cross-dataset comparison despite differences in ROI definitions.

**Order-based vs. score-based agreement.**    We quantify cross-dataset agreement in two complementary ways (Fig. S11). First, in an *order-based* (rank) comparison, we rank layers *within each model* by their predictivity for a given super-ROI (best layer has rank 1), normalize the ranks to $[0, 1]$ to account for differing numbers of layers, and then correlate normalized ranks between datasets. This analysis tests whether two datasets prefer the *same layers in the same relative order* within each model, and is therefore sensitive to shifts in the location of the optimal (or near-optimal) layer. Second, in a *score-based* comparison, we correlate the *raw layerwise predictivity values* between datasets. Score-based agreement captures whether layers that score highly in one dataset also score highly in the other and whether the overall layerwise profile co-varies with depth, and can remain high even when the precise argmax layer differs (e.g., when both datasets share a broad monotonic increase and plateau but peak at slightly different depths).

**Results.**    Across NSD-fMRI and THINGS-fMRI, we observe strong agreement in both metrics, indicating a largely consistent layer–hierarchy correspondence across the two fMRI benchmarks. Rank correlations are high across the hierarchy (Early $r = 0.83$, Mid $r = 0.89$, High $r = 0.88$), and score correlations are similarly strong (Early $r = 0.92$, Mid $r = 0.95$, High $r = 0.94$), suggesting that the overall layerwise profile is preserved and that the preferred layers are broadly consistent. In contrast, NSD-fMRI and TVSD-EP show a dissociation between the two metrics: while score-based agreement remains high (Early $r = 0.77$, Mid $r = 0.92$, High $r = 0.96$), rank-based agreement is weaker in early and mid stages (Early $r = 0.54$, Mid $r = 0.68$) and improves for higher-level regions (High $r = 0.83$). This pattern suggests that NSD and TVSD often agree on which depth ranges are broadly predictive (yielding high score correlation), but differ more in the fine-grained ordering and the precise depth of the optimum, particularly for earlier and intermediate visual areas.

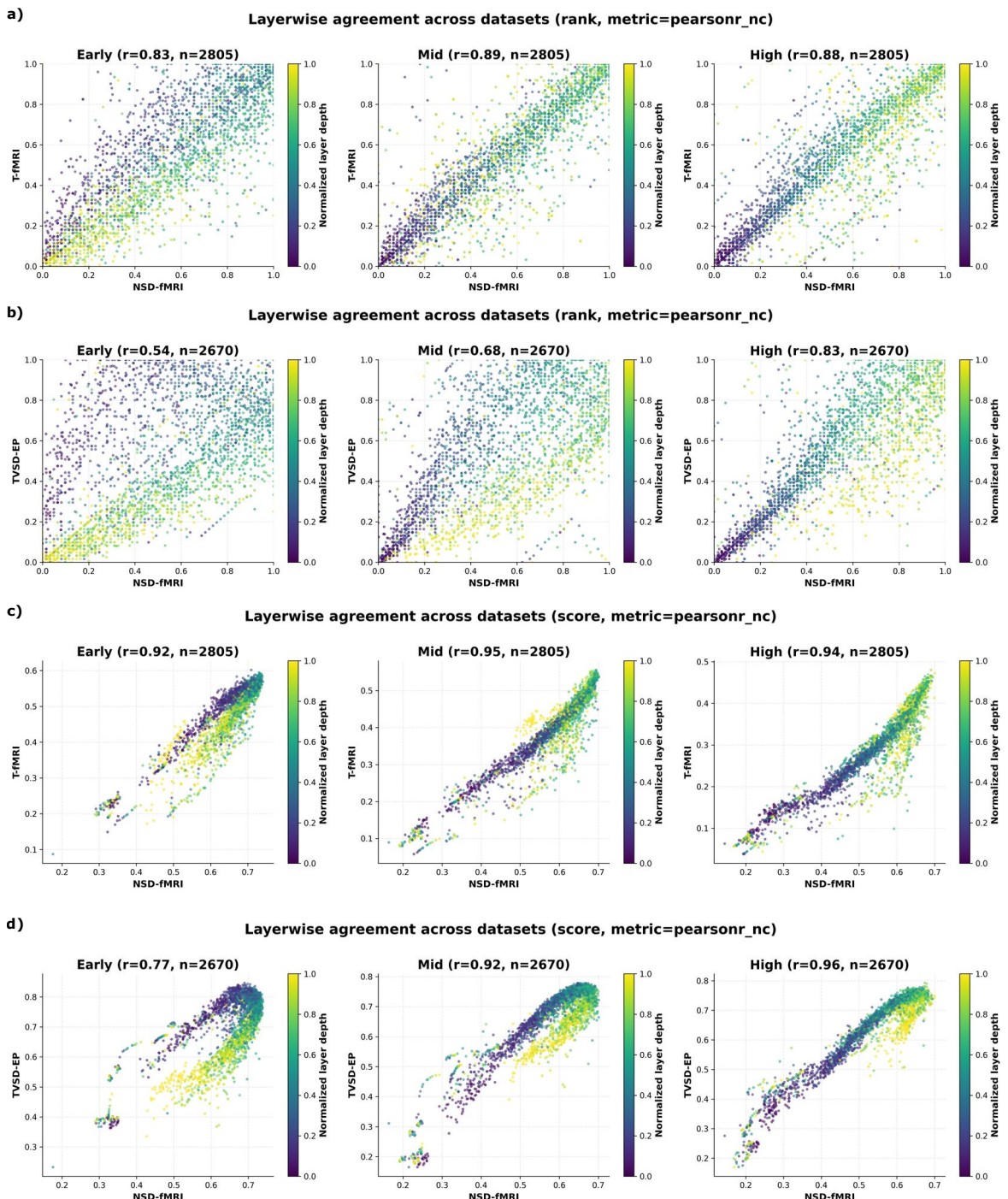

*Figure S11.* **Cross-dataset consistency of the layer–ROI hierarchy.** We compare layerwise brain-alignment profiles between NSD-fMRI and two independent benchmarks using noise-normalized Pearson $r$. ROIs are grouped into **Early**, **Mid**, and **High** super-ROIs (see text). Each point corresponds to a model layer (colored by normalized layer depth), plotted for matched layers across datasets. **a–b)** *Order-based* agreement: layers are ranked within each model by predictivity and compared across datasets (NSD vs. THINGS-fMRI in **a**; NSD vs. TVSD-EP in **b**). **c–d)** *Score-based* agreement: raw layerwise predictivity values are compared (NSD vs. THINGS-fMRI in **c**; NSD vs. TVSD-EP in **d**). Titles report Pearson correlation $r$ and the number of matched model-layer points $n$ for each super-ROI.

## FZ-EP

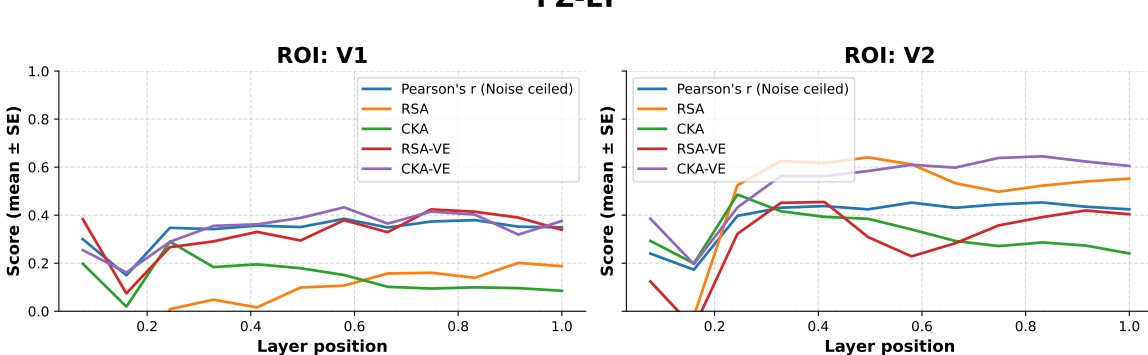

*Figure S12.* Layerwise progression for ViT-S on FZ-EP: ROI alignment across normalized layer position, comparing noise-ceiled Pearson *r* with RSA/CKA and variance-explained variants.

## MH-EP

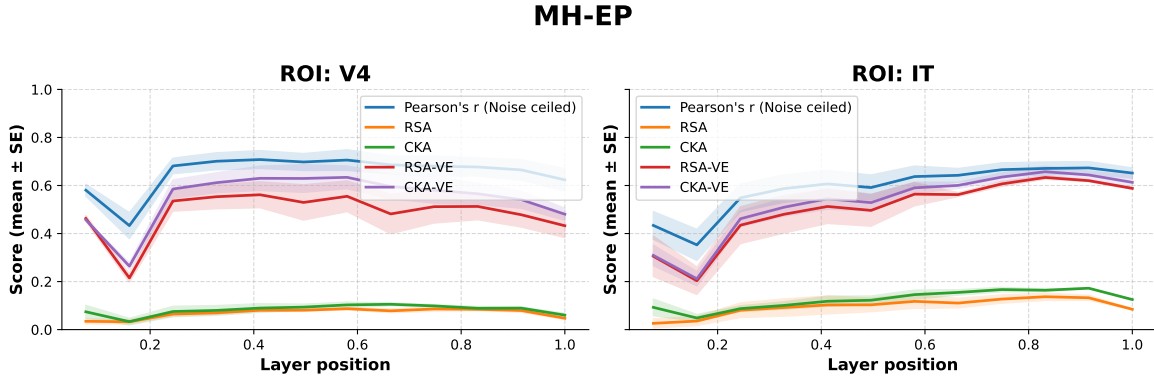

*Figure S13.* Layerwise progression for ViT-S on MH-EP: ROI alignment across normalized layer position, comparing noise-ceiled Pearson *r* with RSA/CKA and variance-explained variants.

## TVSD-EP

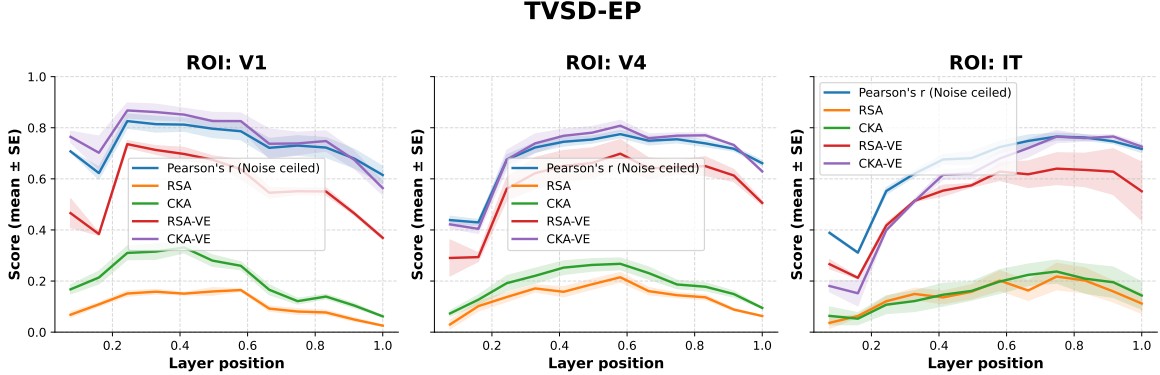

*Figure S14.* Layerwise progression for ViT-S on TVSD-EP: ROI alignment across normalized layer position, comparing noise-ceiled Pearson *r* with RSA/CKA and variance-explained variants.

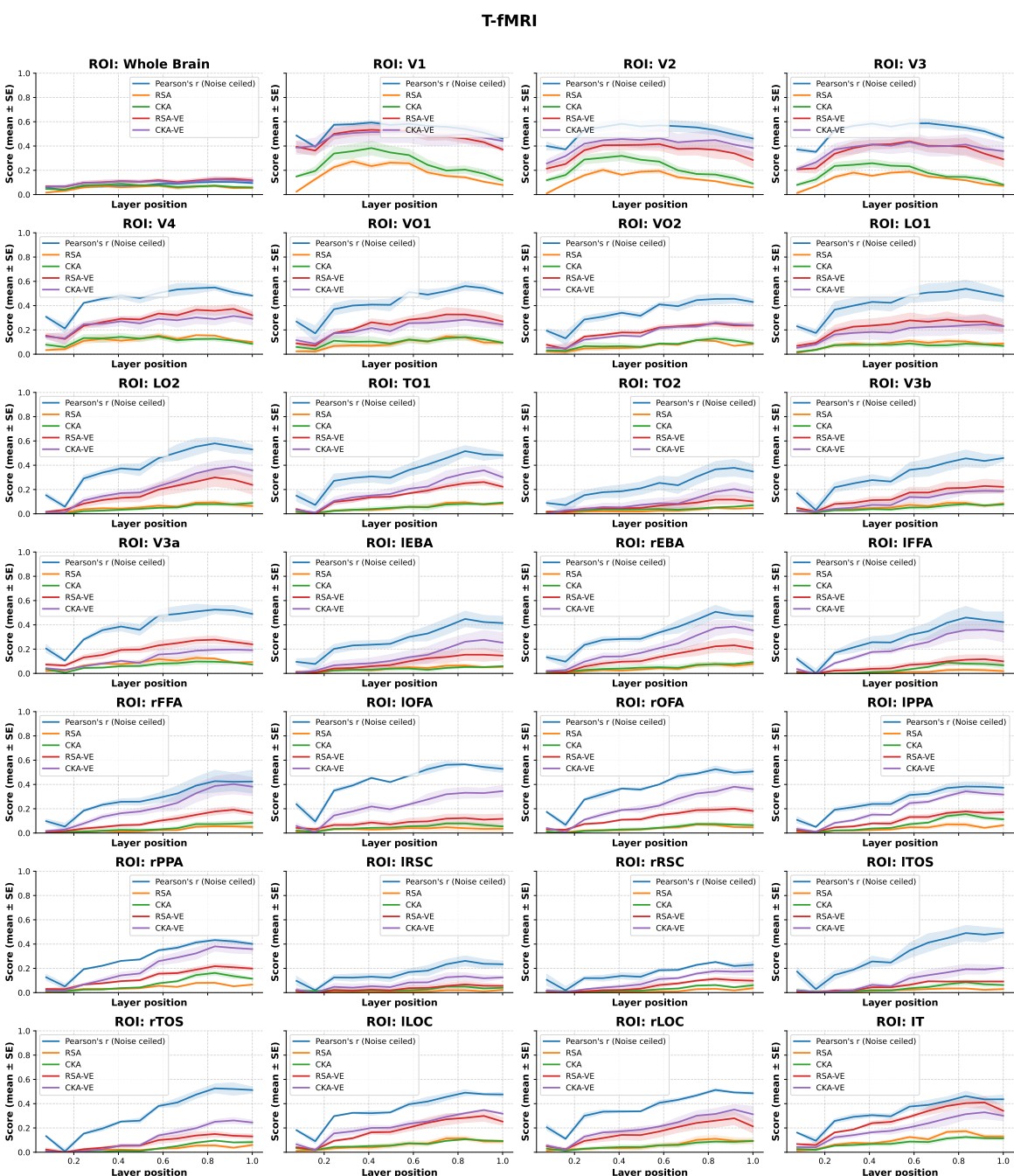

*Figure S15.* Layerwise progression for ViT-S on T-fMRI: ROI alignment across normalized layer position, comparing noise-ceiled Pearson *r* with RSA/CKA and variance-explained variants.

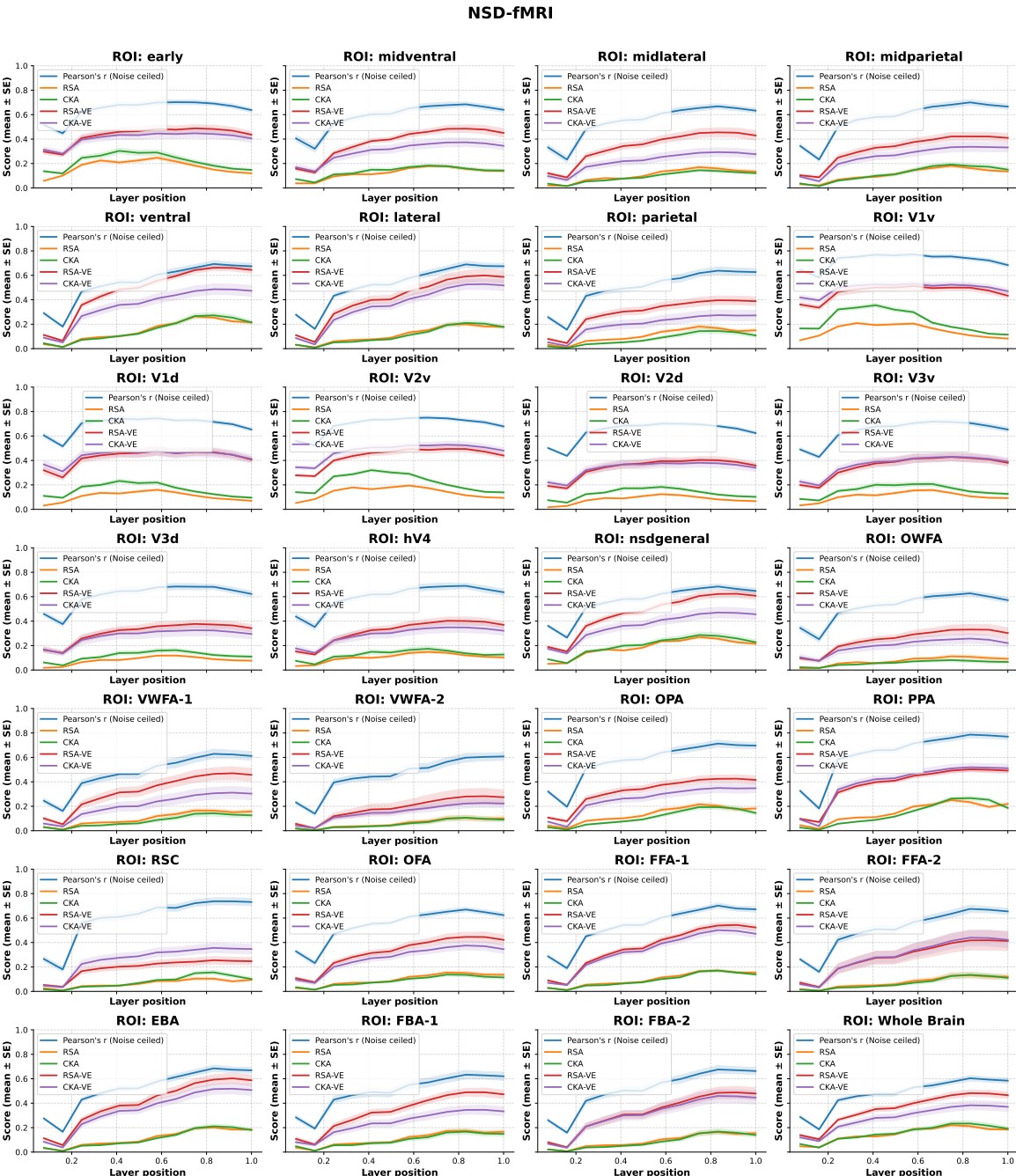

*Figure S16.* Layerwise progression for ViT-S on NSD-fMRI: ROI alignment across normalized layer position, comparing noise-ceiled Pearson $r$ with RSA/CKA and variance-explained variants.

**T-EEG1**

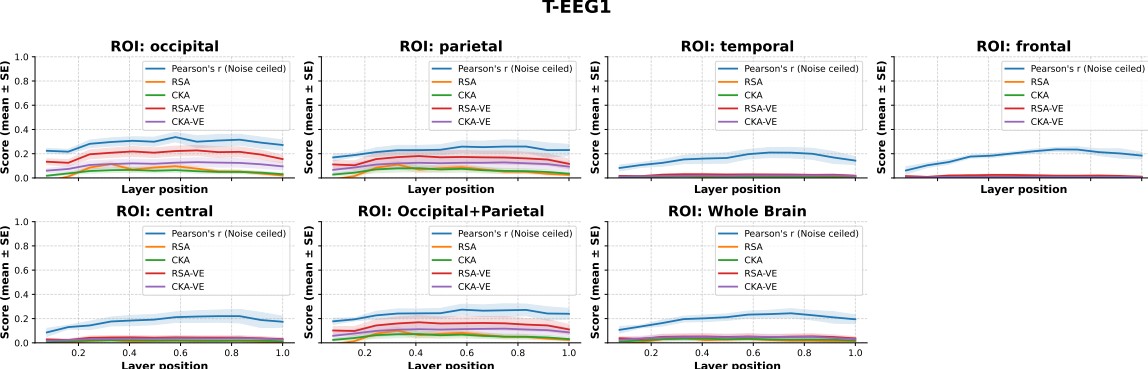

*Figure S17.* Layerwise progression for ViT-S on T-EEG1: ROI alignment across normalized layer position, comparing noise-ceiled Pearson $r$ with RSA/CKA and variance-explained variants.

**T-EEG2**

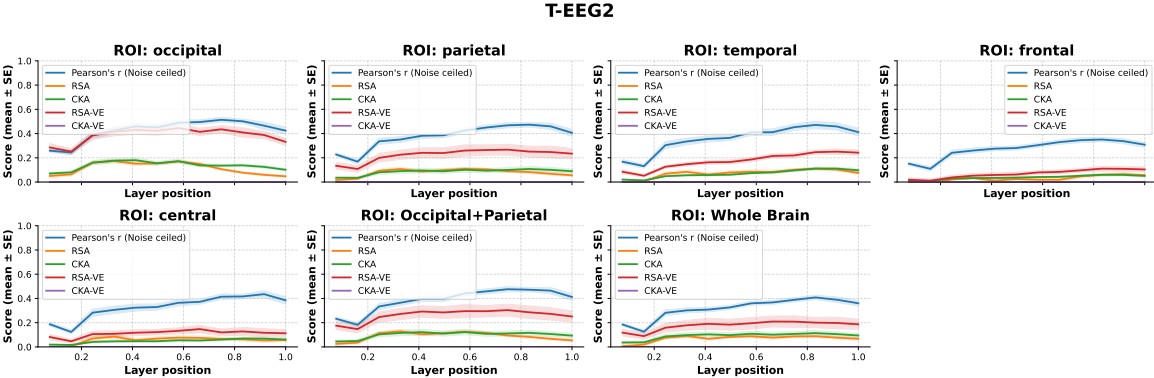

*Figure S18.* Layerwise progression for ViT-S on T-EEG2: ROI alignment across normalized layer position, comparing noise-ceiled Pearson $r$ with RSA/CKA and variance-explained variants.

**T-MEG**

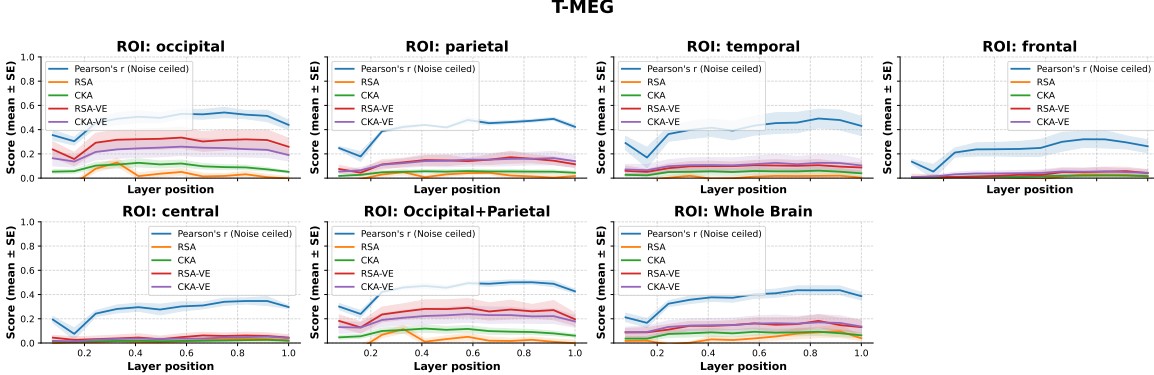

*Figure S19.* Layerwise progression for ViT-S on T-MEG: ROI alignment across normalized layer position, comparing noise-ceiled Pearson $r$ with RSA/CKA and variance-explained variants.

# E. Pretraining Scaling and Metric Robustness

## E.1. Pretraining scale effects across architectures

Figures S20 and S21 provide a detailed breakdown of alignment across architecture families and benchmarks as we vary (i) the number of pretraining samples and (ii) backbone parameter count, without fitting parametric scaling curves.

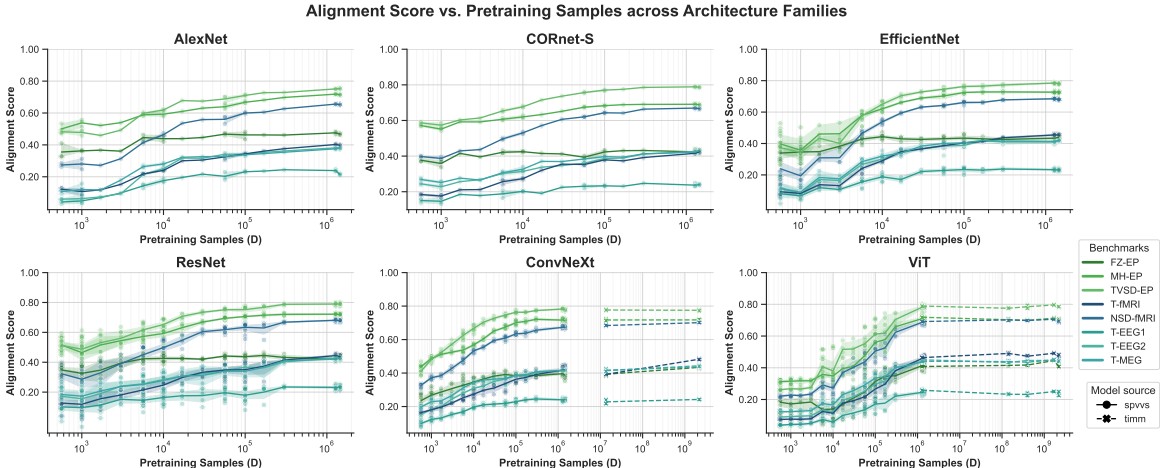

*Figure S20.* Alignment across benchmarks as a function of pretraining dataset size, shown separately for each architecture family.

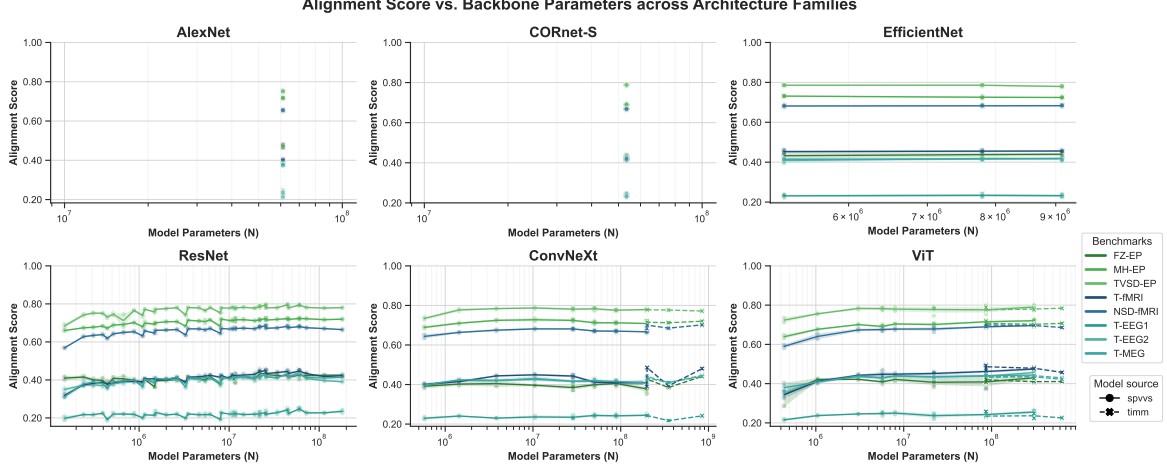

*Figure S21.* Alignment across benchmarks as a function of backbone parameter count, shown separately for each architecture family.

## E.2. Supplementary results on diverse model classes

In the main text, we primarily evaluate image-encoder models. Here, we present additional results for several models of independent interest: (i) a ResNet-152 from Gokce & Schrimpf (2025) that is first trained on ImageNet for 100 epochs and then adversarially fine-tuned for 10 epochs using the FFGSM attack (Goodfellow et al., 2015; Wong et al., 2020); (ii) the self-supervised world/video model V-JEPA2-L (Assran et al., 2025); and (iii) the vision–language model Qwen3-VL-2B-Instruct (Bai et al., 2025).

Figure S22 shows that, after 0–1 normalization, the overall alignment levels of these models are relatively close, further supporting the pretraining saturation claim. Figures S23 and S24 provide a benchmark-resolved view: adversarial fine-tuning improves over the standard ImageNet-trained ResNet-152 and yields the strongest performance on datasets dominated by early visual regions, such as electrophysiology, as well as on EEG benchmarks. In contrast, Qwen3-VL-2B-Instruct achieves the highest scores on datasets emphasizing higher-level cortical regions, such as fMRI and MEG. Finally, Figure S25 puts

model size into perspective, showing that increased parameter count does not necessarily translate into improved neural alignment.

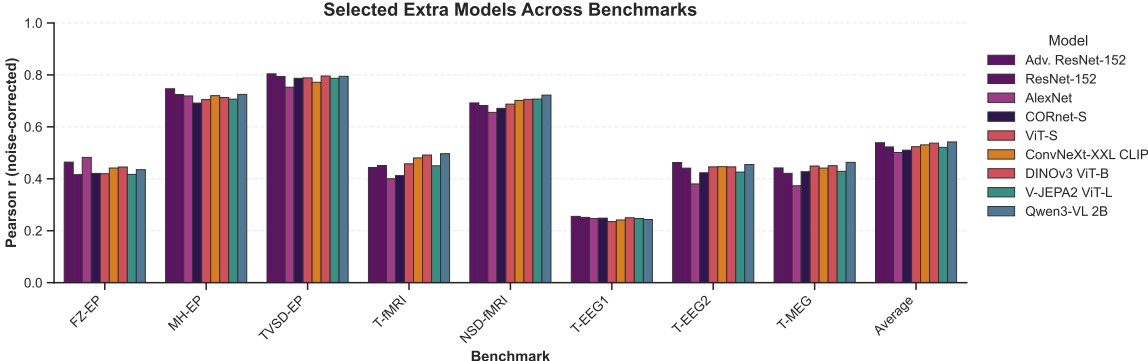

*Figure S22.* Alignment across selected models, shown on a single 0–1 normalized scale.

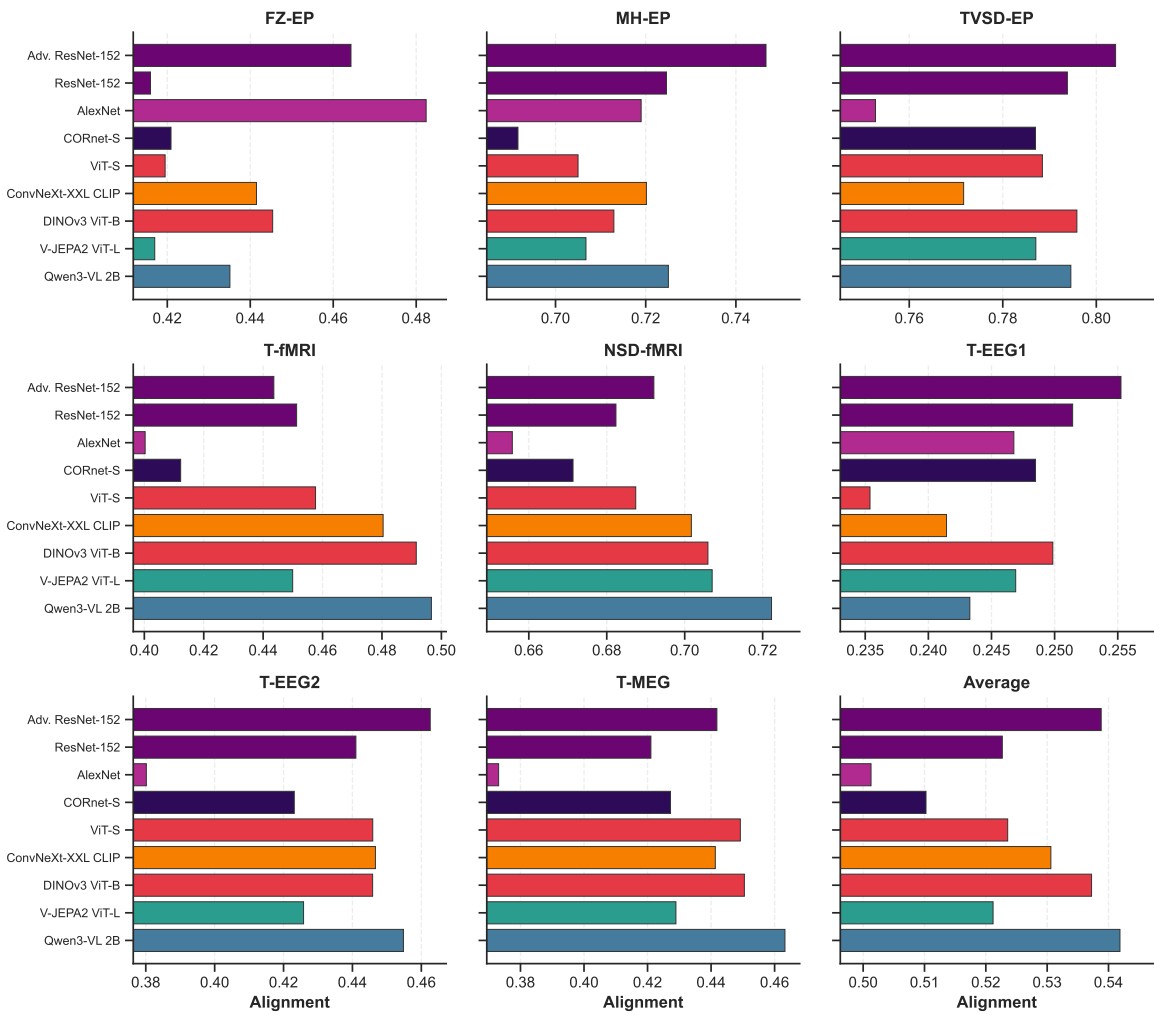

*Figure S23.* Benchmark-resolved alignment for selected models (zoomed view).

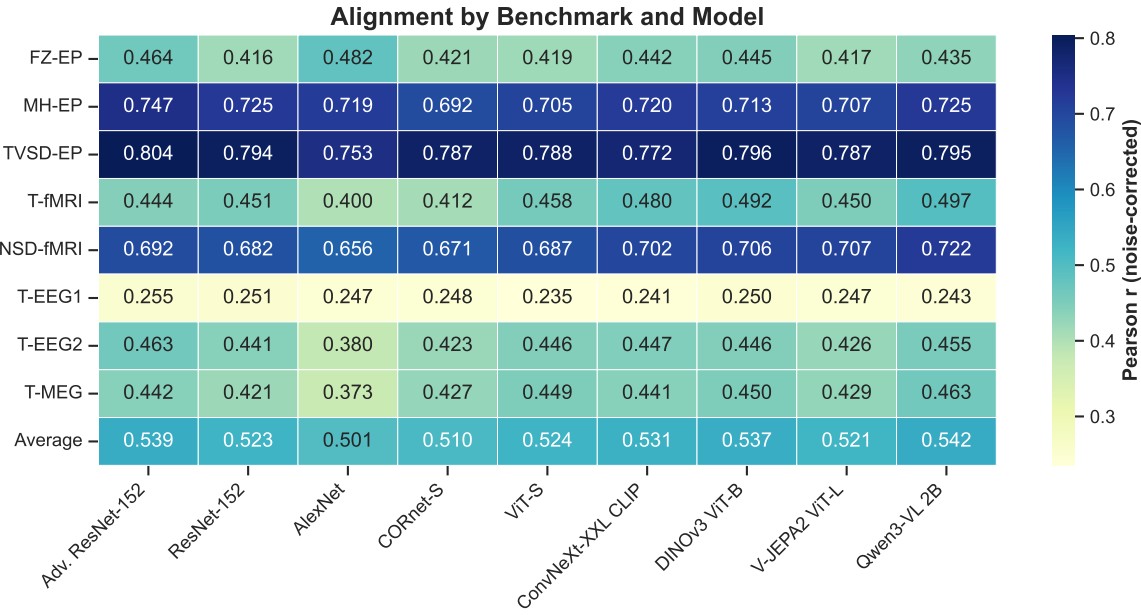

*Figure S24.* Alignment scores for each selected model and benchmark.

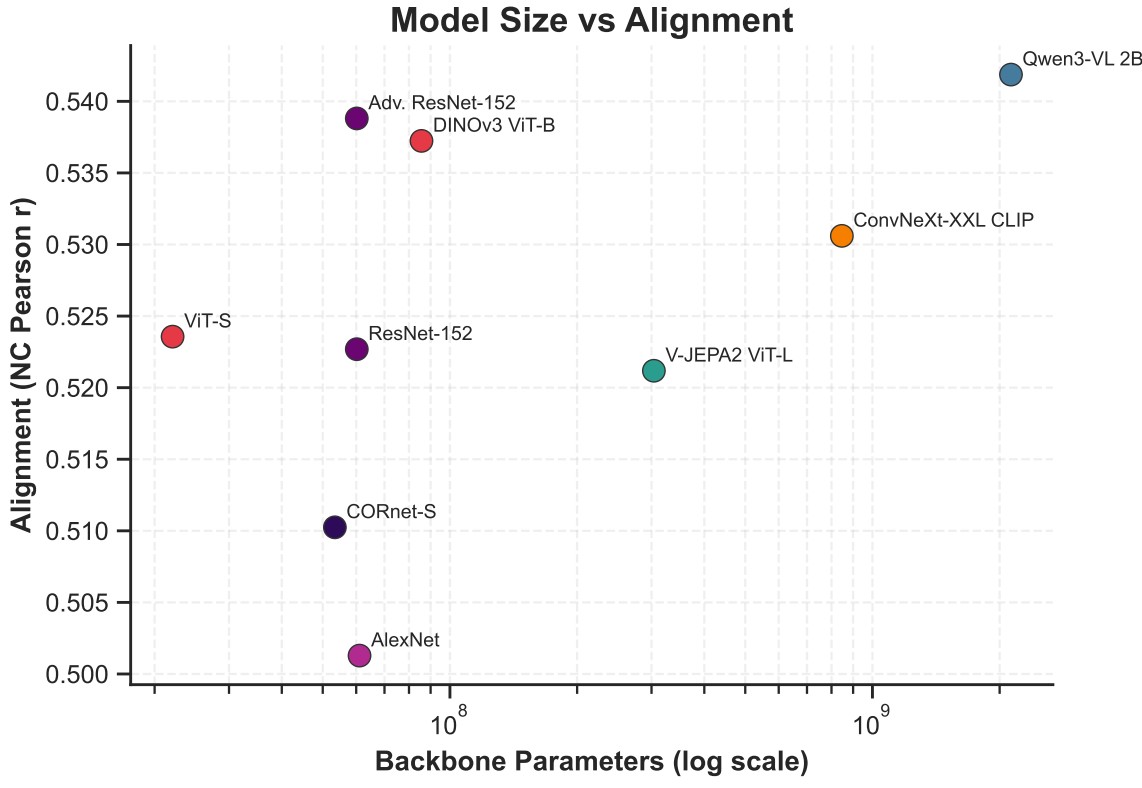

*Figure S25.* Relationship between model parameter count and benchmark-average alignment for the selected extra models. The plot separates model scale from empirical alignment and highlights cases where larger parameter count does not directly imply higher neural alignment.

### E.3. Robustness to alignment metric choice: RSA and CKA

In the main text, we quantify model–brain alignment with an encoding-model pipeline (noise-ceiled Pearson $r$). Here we provide two complementary, *mapping-free* metrics—Representational Similarity Analysis (RSA) and Centered Kernel Alignment (CKA)—and mirror the same layerwise and scaling analyses with these alternatives. For each dataset, subject $s$, ROI $r$, and stimulus set of size $n$, let $Y_{r,s} \in \mathbb{R}^{n \times p_{r,s}}$ denote neural responses (voxels/channels) and $Z_\ell \in \mathbb{R}^{n \times d_\ell}$ denote model features from layer $\ell$ (both arranged by stimulus).

**Representational similarity analysis (RSA).**   RSA compares the geometry of representational spaces via representational dissimilarity matrices (RDMs). We form RDMs for model and neural representations using correlation distance,

$$D_{ij}^{Z_\ell} \;=\; 1 - \mathrm{corr}(Z_\ell[i,:],\, Z_\ell[j,:])\,, \qquad D_{ij}^{Y_{r,s}} \;=\; 1 - \mathrm{corr}(Y_{r,s}[i,:],\, Y_{r,s}[j,:])\,, \tag{9}$$

for stimuli $i \neq j$. We then vectorize the upper triangles and compute similarity as a rank correlation

$$\mathrm{RSA}(Z_\ell, Y_{r,s}) \;=\; \rho_{\mathrm{Spearman}}\big(\mathrm{vec}\big(D^{Z_\ell}\big),\, \mathrm{vec}\big(D^{Y_{r,s}}\big)\big)\,. \tag{10}$$

**Unbiased linear CKA.**   As an alternative, mapping-free alignment metric, we compute (debiased) linear centered kernel alignment (CKA) between model features and neural responses. For layer $\ell$, we arrange model features $Z_\ell \in \mathbb{R}^{n \times d_\ell}$ and neural responses $Y_{r,s} \in \mathbb{R}^{n \times p_{r,s}}$ across the same $n$ stimuli, form linear Gram matrices $K = Z_\ell Z_\ell^\top$ and $L = Y_{r,s} Y_{r,s}^\top$, and estimate dependence using the unbiased (U-statistic) HSIC estimator $\mathrm{HSIC}_u(K, L)$ (with diagonals removed). We then define alignment as

$$\mathrm{CKA}(Z_\ell, Y_{r,s}) = \frac{\mathrm{HSIC}_u(K, L)}{\sqrt{\mathrm{HSIC}_u(K, K)\, \mathrm{HSIC}_u(L, L)}}\,. \tag{11}$$

We report both classical RSA/CKA variants (cRSA/cCKA), which compare pairwise stimulus relationships via RDMs or Gram matrices, and *voxel-encoding* variants (veRSA/veCKA). While originally introduced in the fMRI literature, voxel-encoding here simply denotes applying a lightweight linear readout from model features to the response space (e.g., voxels/channels) and then computing the corresponding similarity score (RSA/CKA) on these linearly transformed predictions. We apply veRSA/veCKA uniformly across all benchmarks for consistency. Empirically, the trends in our main figures are most closely matched by the voxel-encoding metrics: veRSA and veCKA yield clearer, more consistent scaling behavior, whereas the classical variants show the same overall direction but with reduced contrast.

To keep this supplementary analysis computationally tractable, we do not recompute layer assignments under RSA/CKA. Instead, we evaluate cRSA/cCKA and veRSA/veCKA at the same layer selected by our primary encoding-model criterion in the main text. We additionally report raw RSA/CKA scores (without noise normalization). Under these choices, veRSA and veCKA closely track the patterns reported in the main text, while cRSA and cCKA exhibit similar but less pronounced trends (Figures S26, S27, S28, S29).

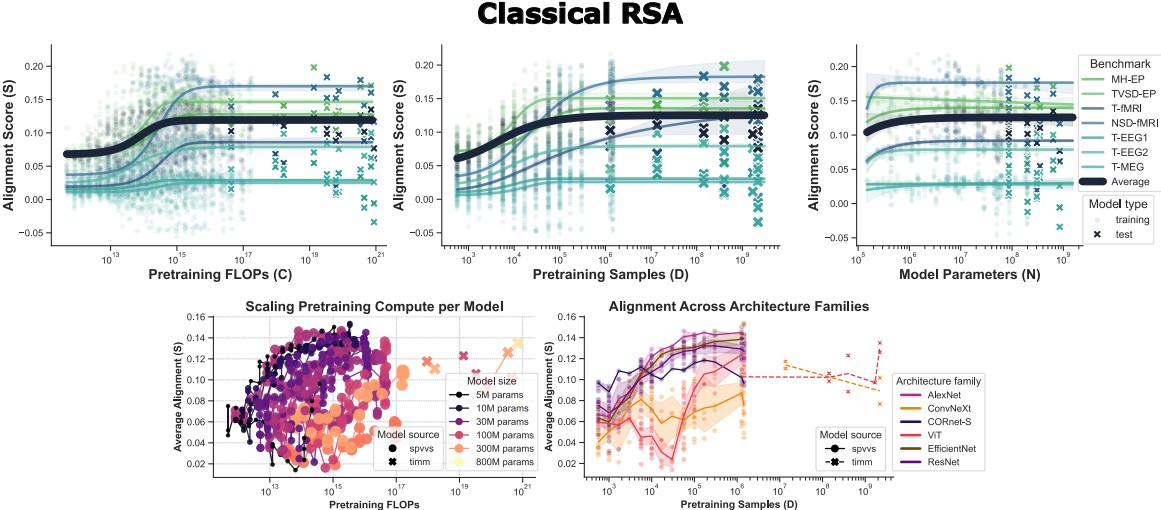

*Figure S26.* Pretraining scaling analysis using classical RSA (cRSA).

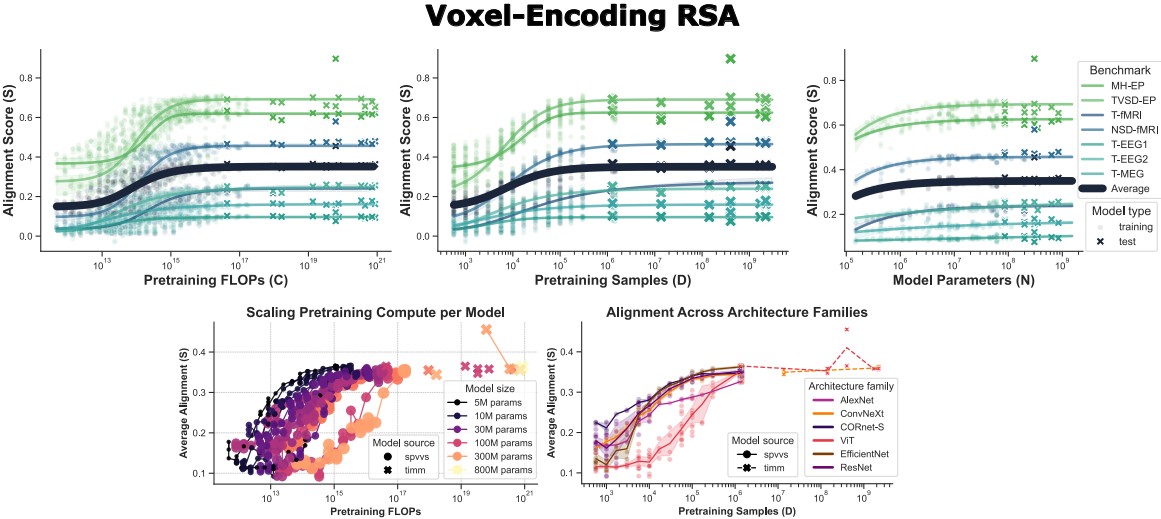

*Figure S27.* Pretraining scaling analysis using voxel-encoding RSA (veRSA).

## Classical CKA

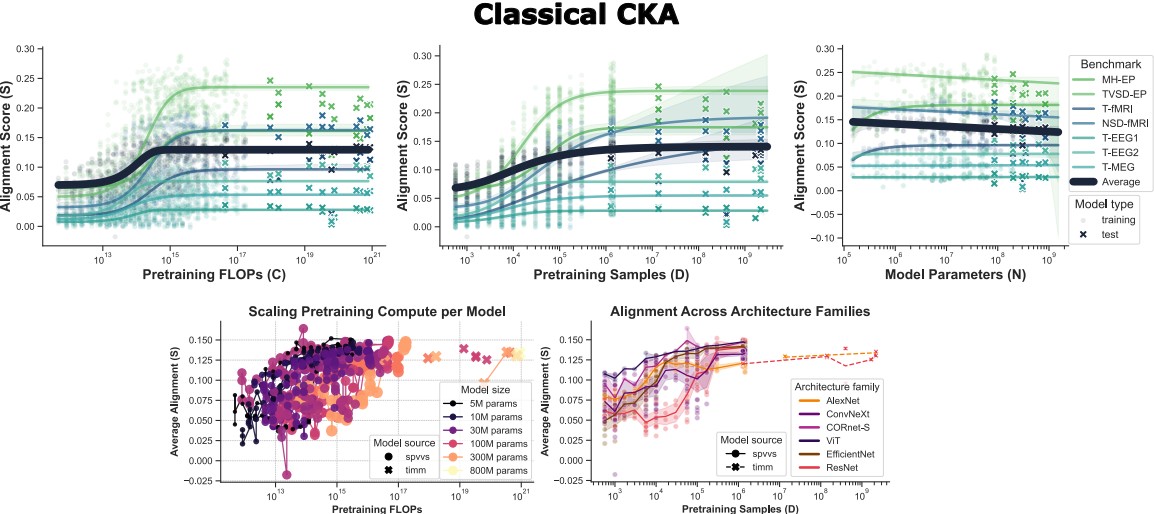

*Figure S28.* Pretraining scaling analysis using classical CKA (cCKA).

## Voxel-Encoding CKA

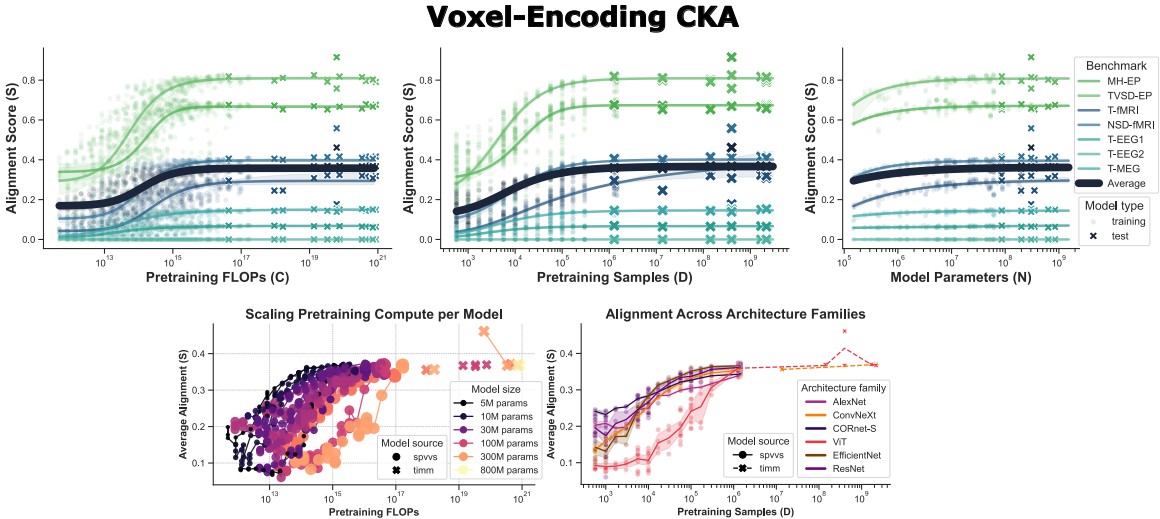

*Figure S29.* Pretraining scaling analysis using voxel-encoding CKA (veCKA).

## F. Relationship Between Task Accuracy and Neural Alignment

**Benchmark-level trends.**    We first examine the relationship between supervised pretraining accuracy and neural alignment at the benchmark level. Figure S30 plots alignment averaged across ROIs for each of the eight benchmarks, showing ImageNet- and Ecoset-pretrained models from multiple architecture families without imposing a fitted trend. Across benchmarks, models with higher validation accuracy tend to obtain higher neural alignment, motivating a more detailed examination of the regional structure of this association.

**ROI-resolved relationships.**    We next resolve the relationship within individual ROIs. Figures S31, S32, and S33 show positive relationships in FZ-EP, MH-EP, and TVSD-EP, with stronger accuracy–alignment coupling in higher-level visual ROIs. For ImageNet-pretrained models, correlations increase from V1 to V2 in FZ-EP ($r = 0.376$ to $0.533$), from V4 to IT in MH-EP ($r = 0.700$ to $0.831$), and from V1 to V4 to IT in TVSD-EP ($r = 0.616, 0.776$, and $0.820$); Ecoset-pretrained models show the same ordering. Figures S34 and S35 extend the positive relationship across the ROI sets of T-fMRI and NSD-fMRI, and Figures S36, S37, and S38 show corresponding positive associations across T-EEG1, T-EEG2, and T-MEG sensor-region groupings. Across the 84 ROIs evaluated for each pretraining dataset, all 168 ImageNet- or Ecoset-specific correlations are positive and significant after Benjamini–Hochberg correction ($p_{\mathrm{FDR}} < 10^{-11}$).

**High-accuracy regime.**    The full accuracy range suggests a non-linear relationship: alignment increases rapidly at low accuracy levels and appears to increase with a shallower slope among higher-accuracy models. Figure S39 isolates models with validation accuracy $\geq 0.40$ and averages alignment across all eight benchmarks. Even within this range, the direction remains strongly positive for both Ecoset-pretrained models ($r = 0.72$, $p_{\mathrm{FDR}} = 5.8 \times 10^{-21}$, $n = 124$) and ImageNet-pretrained models ($r = 0.67$, $p_{\mathrm{FDR}} = 3.7 \times 10^{-20}$, $n = 142$). Thus, although the full data suggest diminishing gains in alignment as accuracy increases, alignment continues to increase with task performance among high-accuracy models.

**Summary across ROIs.**    Finally, Figures S40, S43, S41, and S42 directly summarize ROI-wise correlation strengths for ImageNet-pretrained models with bootstrap confidence intervals. In FZ-EP, MH-EP, and TVSD-EP, correlations range from $r = 0.38$ to $0.83$ and visibly reinforce the stronger coupling in higher-level visual ROIs. In the T-EEG1, T-EEG2, and T-MEG benchmarks, they range from $r = 0.65$ to $0.86$. T-fMRI and NSD-fMRI also show broadly strong effects across ROIs, with ranges of $r = 0.68$–$0.93$ and $r = 0.69$–$0.87$, respectively. Taken together, these analyses show that higher supervised pretraining task accuracy is a reliable correlate of stronger neural alignment across the evaluated model collection, with particularly strong correspondence in higher-level visual ROIs.

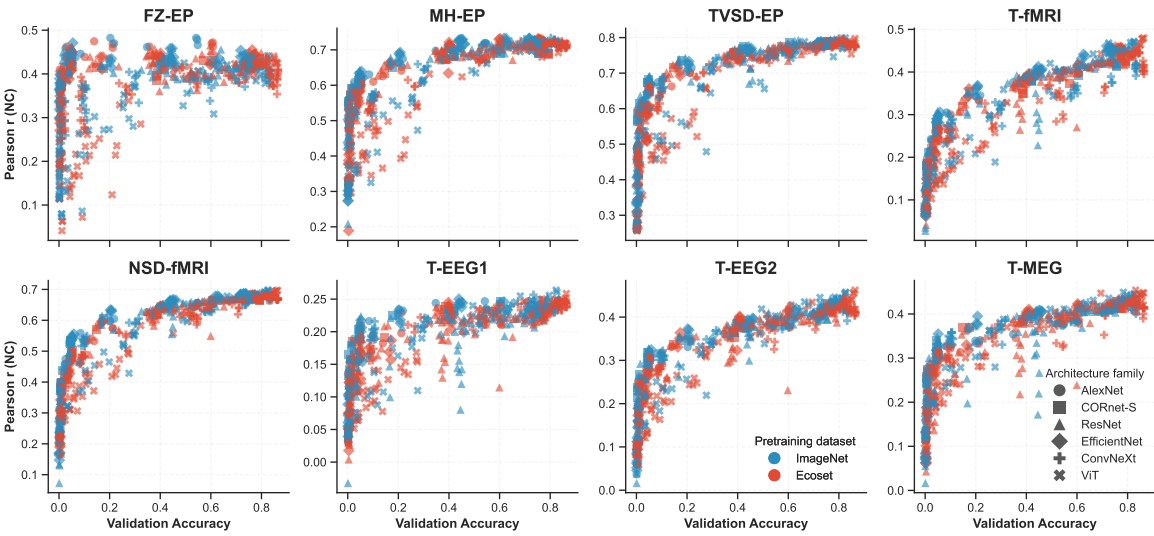

*Figure S30.* Validation accuracy versus neural alignment averaged across ROIs for each benchmark. Colors indicate pretraining dataset and markers indicate architecture family.

## FZ-EP

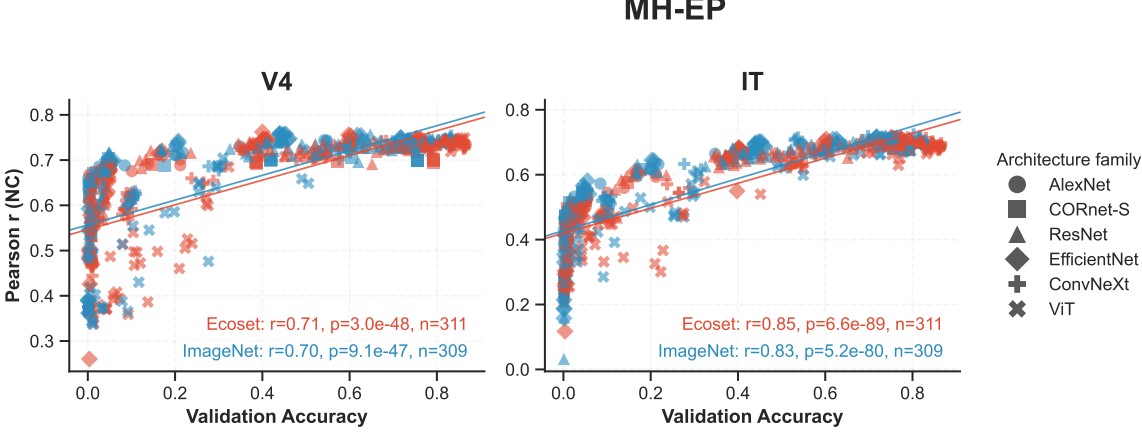

*Figure S31.* Validation accuracy versus ROI-level neural alignment ($r_{\mathrm{NC}}$) in FZ-EP for supervised ImageNet- and Ecoset-pretrained `spvvs` models.

## MH-EP

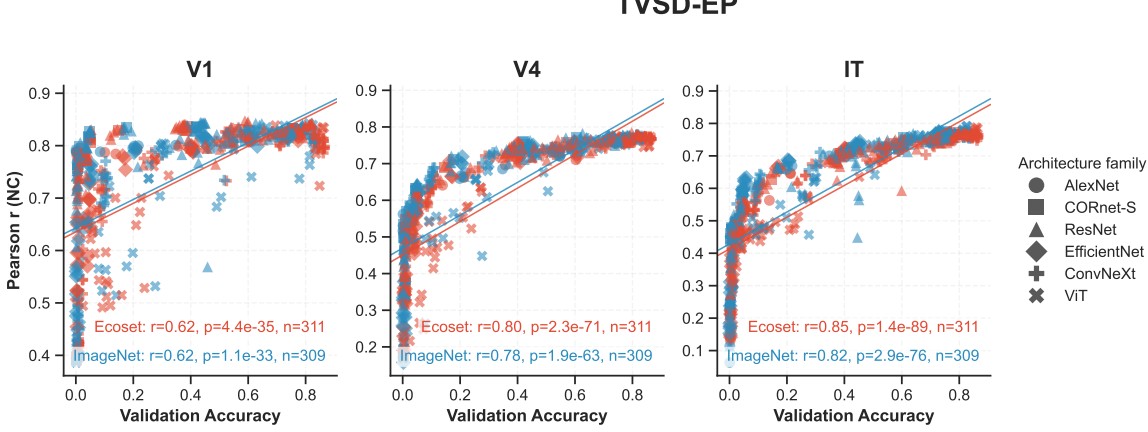

*Figure S32.* Validation accuracy versus ROI-level neural alignment ($r_{\mathrm{NC}}$) in MH-EP for supervised ImageNet- and Ecoset-pretrained `spvvs` models.

## TVSD-EP

*Figure S33.* Validation accuracy versus ROI-level neural alignment ($r_{\mathrm{NC}}$) in TVSD-EP for supervised ImageNet- and Ecoset-pretrained `spvvs` models.

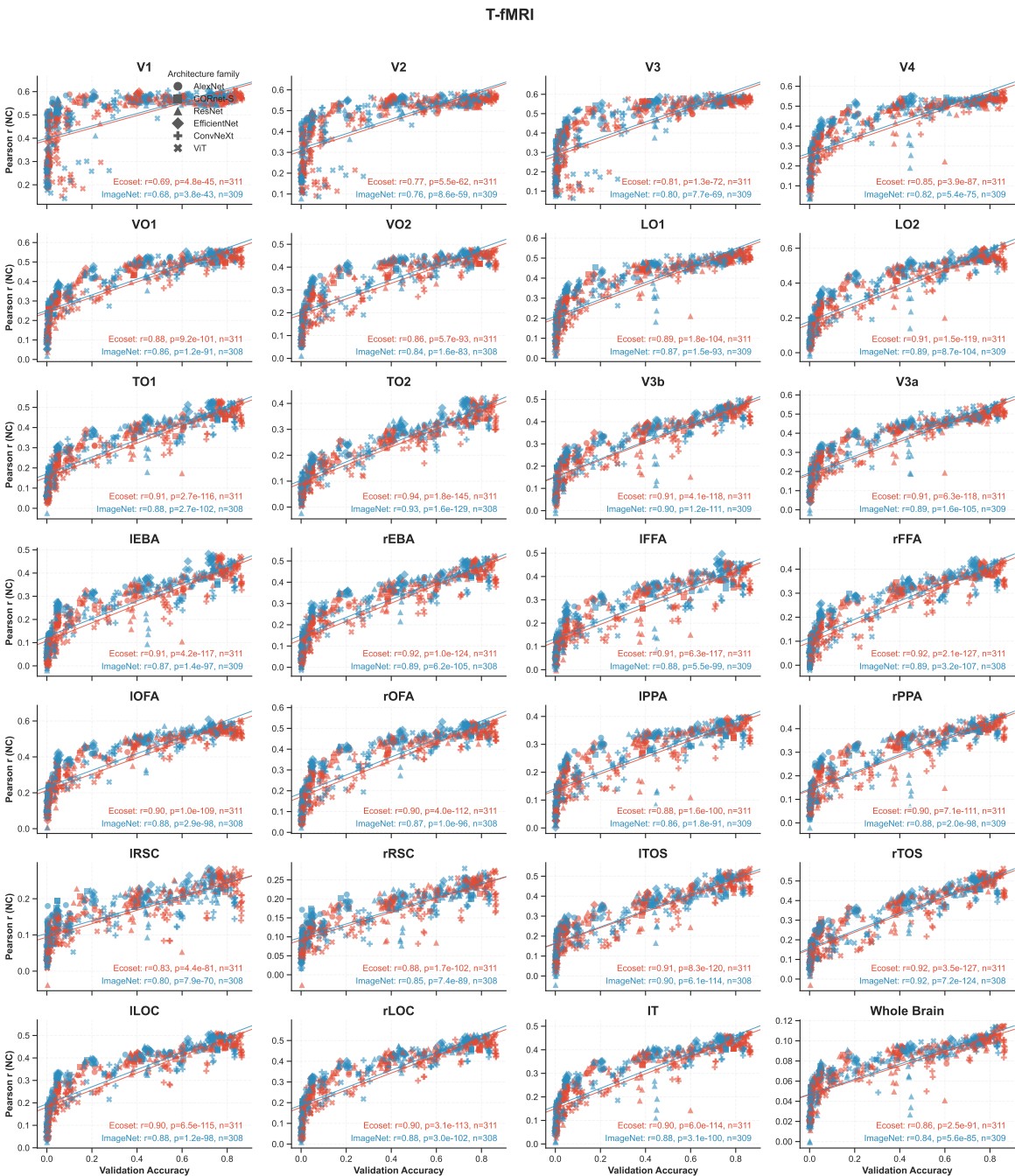

*Figure S34.* Validation accuracy versus ROI-level neural alignment ($r_{\mathrm{NC}}$) in T-fMRI for supervised ImageNet- and Ecoset-pretrained `spvvs` models.

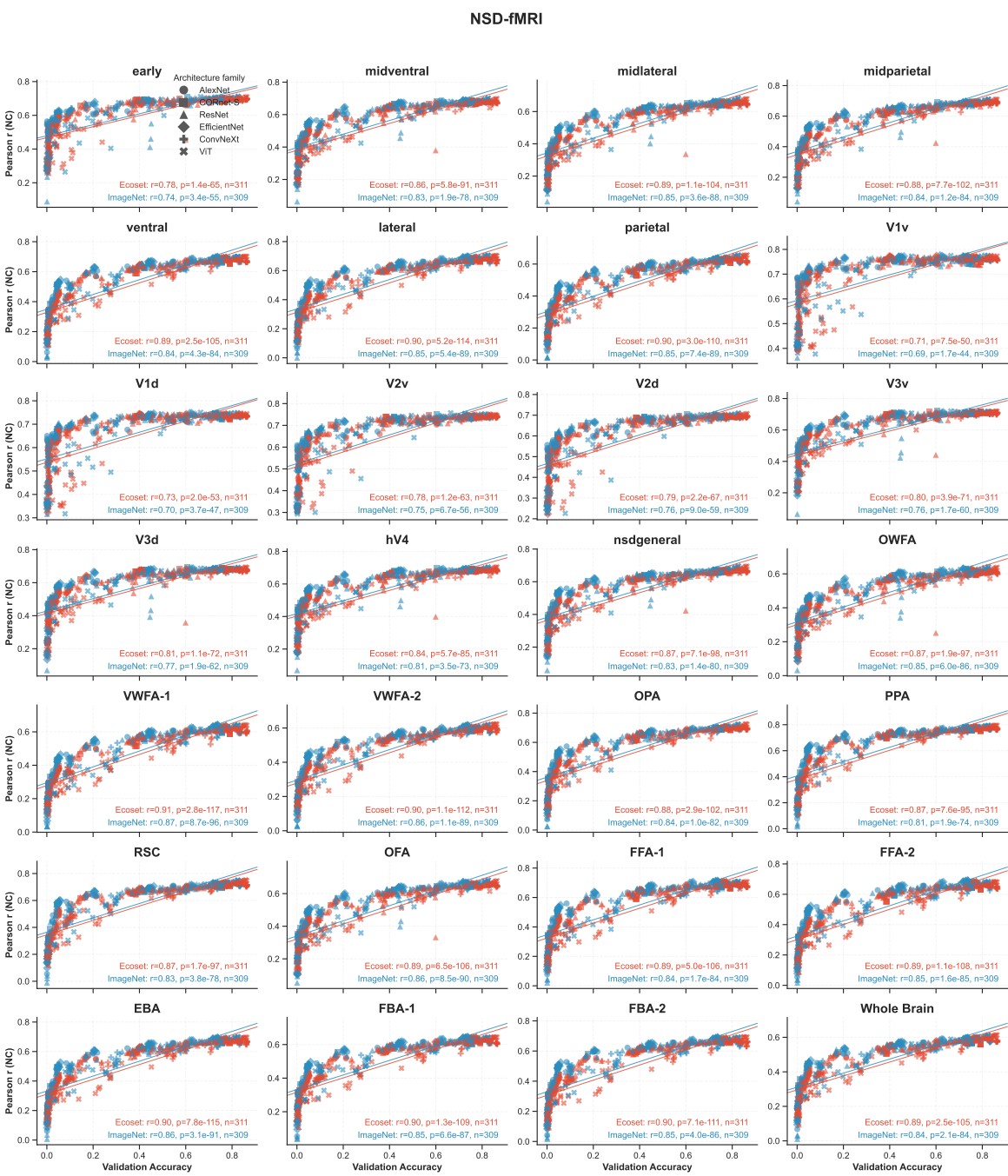

*Figure S35.* Validation accuracy versus ROI-level neural alignment ($r_{\mathrm{NC}}$) in NSD-fMRI for supervised ImageNet- and Ecoset-pretrained `spvvs` models.

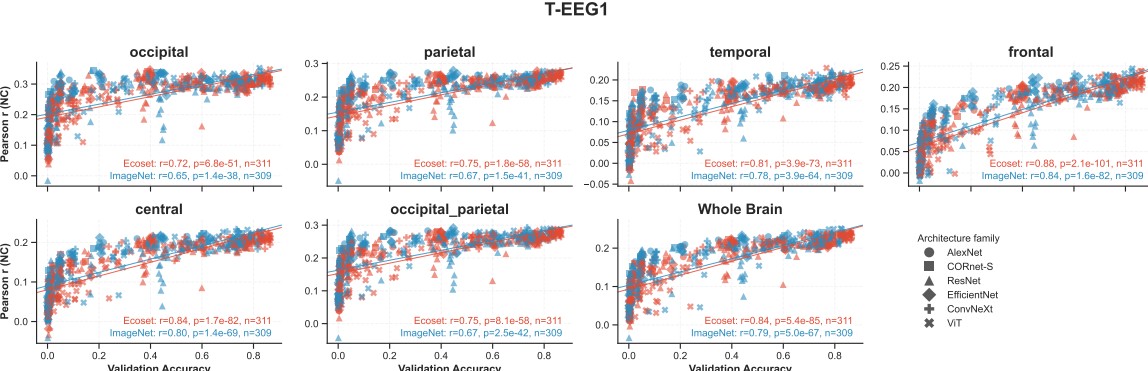

*Figure S36.* Validation accuracy versus ROI-level neural alignment ($r_{\mathrm{NC}}$) in T-EEG1 for supervised ImageNet- and Ecoset-pretrained `spvvs` models.

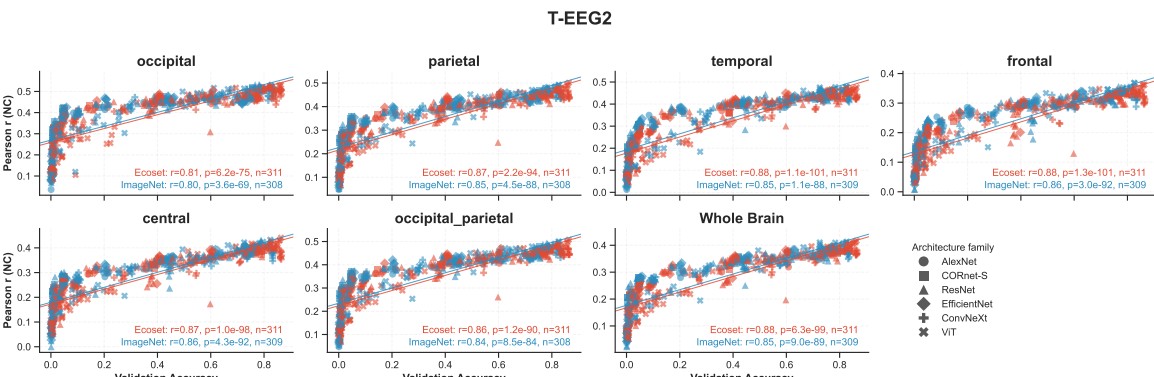

*Figure S37.* Validation accuracy versus ROI-level neural alignment ($r_{\mathrm{NC}}$) in T-EEG2 for supervised ImageNet- and Ecoset-pretrained `spvvs` models.

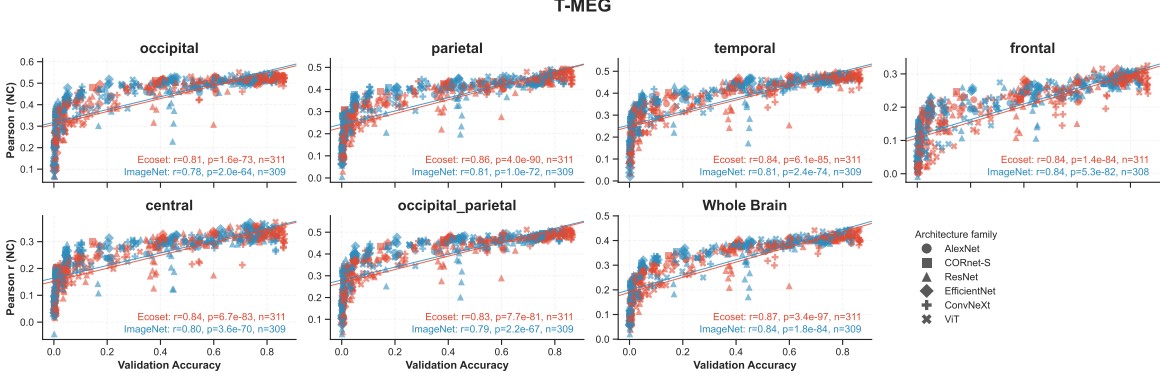

*Figure S38.* Validation accuracy versus ROI-level neural alignment ($r_{\mathrm{NC}}$) in T-MEG for supervised ImageNet- and Ecoset-pretrained `spvvs` models.

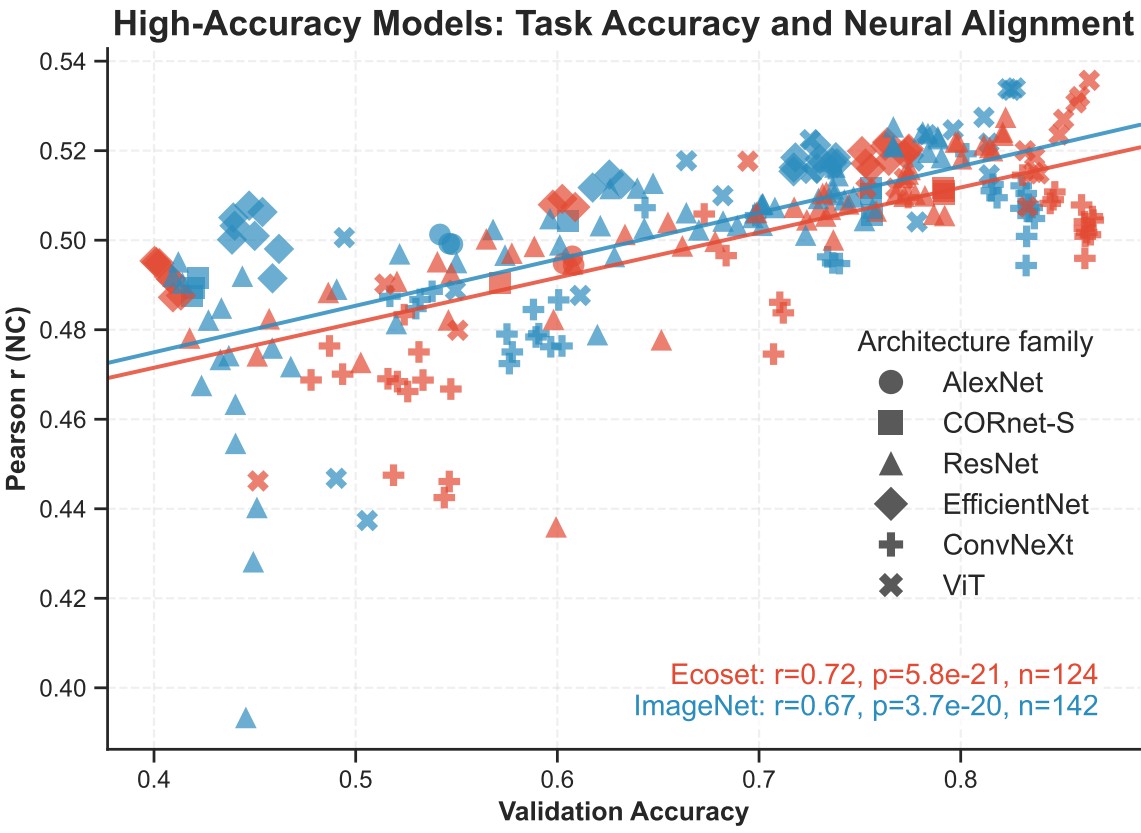

*Figure S39.* Validation accuracy versus mean neural alignment across benchmarks for high-accuracy ImageNet- and Ecoset-pretrained models (accuracy $\geq 0.40$). Alignment continues to increase with accuracy in this restricted range, although with a shallower trend than in the full accuracy range.

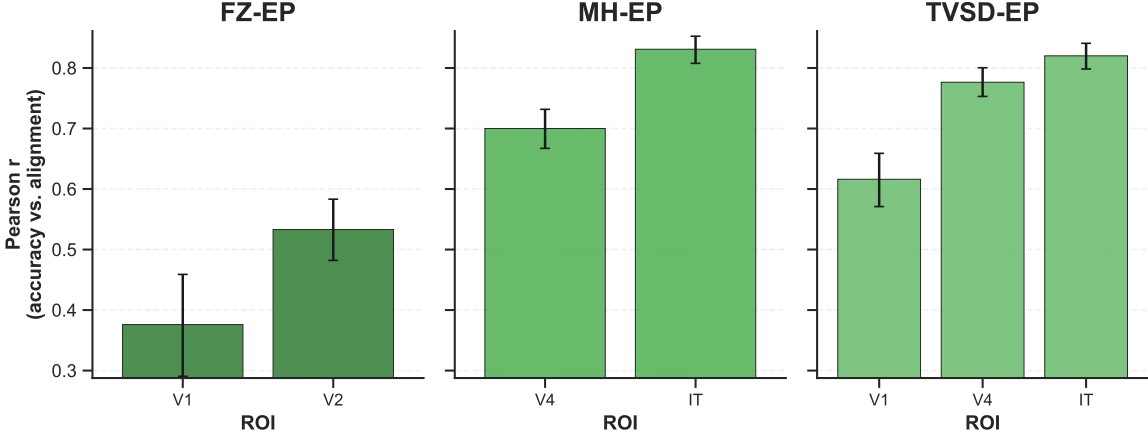

*Figure S40.* ROI-wise accuracy–alignment correlations for FZ-EP, MH-EP, and TVSD-EP in ImageNet-pretrained models; error bars show 95% bootstrap confidence intervals.

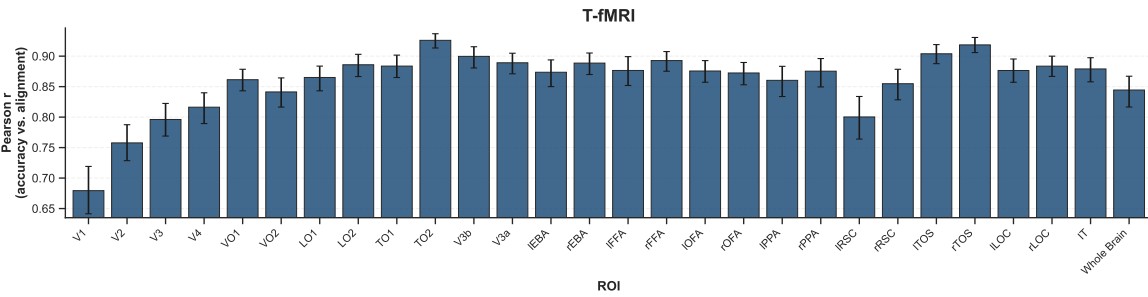

*Figure S41.* ROI-wise accuracy–alignment correlations for T-EEG1, T-EEG2, and T-MEG in ImageNet-pretrained models; error bars show 95% bootstrap confidence intervals.

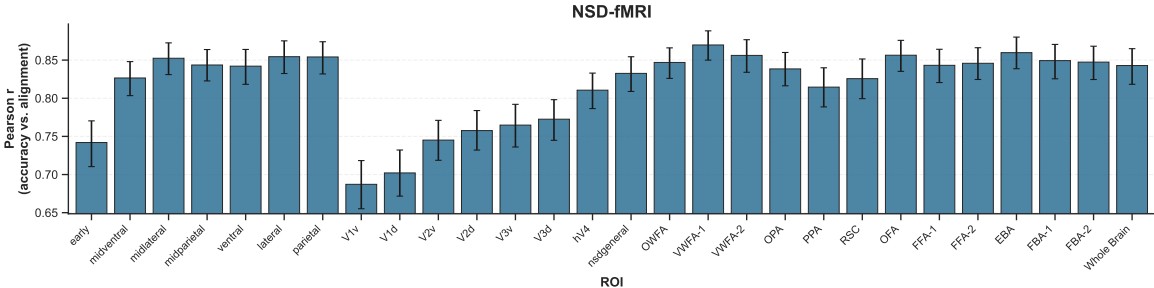

*Figure S42.* ROI-wise accuracy–alignment correlations for NSD-fMRI in ImageNet-pretrained models; error bars show 95% bootstrap confidence intervals.

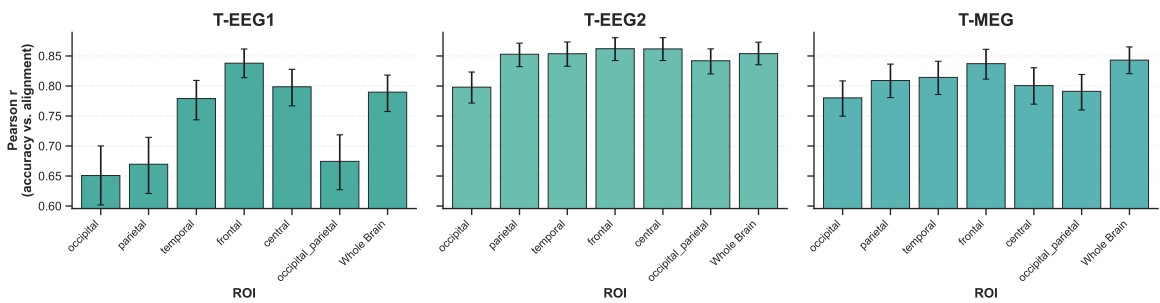

*Figure S43.* ROI-wise accuracy–alignment correlations for T-fMRI in ImageNet-pretrained models; error bars show 95% bootstrap confidence intervals.

# G. Neural Fine-Tuning Details, Transfer, and Controls

### G.1. Implementation details

Table S3 summarizes the fine-tuning hyperparameters used during joint neural-data fitting and task optimization. The configuration is chosen to improve neural predictivity without inducing catastrophic forgetting of the model's original task performance, while enabling effective adaptation over a short training horizon. Epochs are defined with respect to complete passes over the neural dataset, whereas ImageNet batches are sampled continuously throughout training.

For image transformations, we apply the same preprocessing and data augmentations used during the original ImageNet pretraining of the backbone for the ImageNet batches. For neural-data fitting, we adopt the stimulus augmentations described in Dapello et al. (2023), including spatial jitter corresponding to $\pm 0.5°$ translations in both the vertical and horizontal directions, $\pm 0.5°$ rotational jitter, and $\pm 0.1°$ scaling jitter, assuming an $8°$ model field of view.

*Table S3.* Neural data fitting hyperparameters.

| Batching | |
|---|---|
| Image batch size | 512 |
| Neural batch size | 256 |
| **Optimization** | |
| Optimizer | AdamW |
| $\beta_1, \beta_2$ | 0.9, 0.999 |
| $\epsilon$ | $1 \times 10^{-8}$ |
| Encoder learning rate | $1 \times 10^{-5}$ |
| Decoder learning rate | 0 (frozen) |
| Encoder weight decay | 0.05 |
| Decoder weight decay | 0 |
| Max epochs | 20 |
| Precision | bf16-mixed |
| **Learning rate schedule** | |
| Schedule | Cosine annealing + warmup |
| Warmup | 5 epochs |
| Min learning rate | $1 \times 10^{-6}$ |
| **Losses** | |
| Neural loss | MSE |
| Image loss weight ($\lambda_{\text{img}}$) | $10^{-1}$ |
| Neural loss weight ($\lambda_{\text{neural}}$) | 100 |

### G.2. Detailed scaling effects of neural fine-tuning

Figure S44 shows how fine-tuning affects each benchmark as a function of the available neural training data. We find that T-fMRI benefits primarily from fine-tuning on T-fMRI itself, whereas NSD-fMRI can also be improved by fine-tuning on other datasets. Several other benchmarks likewise show cross-dataset transfer, while the trends for T-EEG1 and FZ-EP are less consistent. We attribute this to higher measurement noise in T-EEG1 and the limited number of fitting stimuli in FZ-EP.

### G.3. Sanity checks and the alignment–accuracy trade-off

We perform two additional analyses to validate that fine-tuning gains reflect meaningful stimulus–response learning and to quantify their impact on task performance. First, Fig. S47a shows that breaking the correspondence between stimuli and neural recordings (by shuffling stimulus–response pairs) substantially reduces average brain alignment. This sanity check indicates that our improvements are not driven by fitting noise or dataset-specific artifacts, but instead rely on the correct stimulus-conditioned neural signal.

Second, Fig. S45 reports the effect of neural fine-tuning on downstream task performance, measured as ImageNet (Deng et al., 2009) classification accuracy. We observe a consistent decrease in accuracy after fine-tuning, highlighting a trade-off:

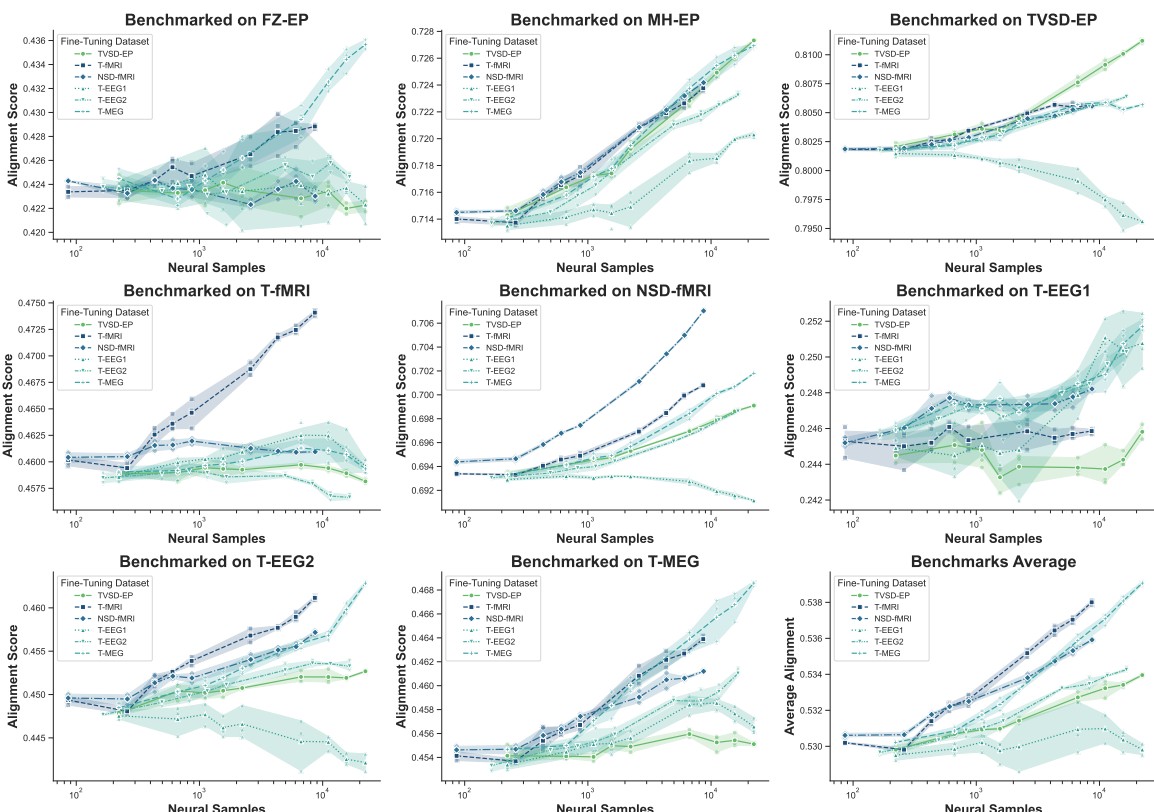

*Figure S44.* Per-benchmark alignment improvements from neural fine-tuning as a function of neural training set size, comparing fine-tuning datasets (within-dataset vs. cross-dataset transfer).

stronger neural supervision can improve brain alignment while partially degrading task performance. In line with this, we find that more aggressive optimization settings (e.g., higher learning rates and longer training) can yield larger within-dataset alignment gains, but typically at the cost of a larger drop in ImageNet accuracy. Together, these results underscore that neural fine-tuning learns stimulus-dependent structure, but its benefits must be balanced against preserving task-level competence.

Figure S46 further separates the ViT-S relationship by the benchmark used to evaluate neural alignment. Averaged over finetuning datasets, alignment increases from the 1% to the 100% data setting in every testing benchmark. The largest increases appear in MH-EP (0.714 to 0.724), T-MEG (0.454 to 0.461), T-EEG2 (0.448 to 0.455), and NSD-fMRI (0.693 to 0.700), while FZ-EP, TVSD-EP, and T-fMRI show smaller positive changes. Individual dataset curves vary within benchmarks, but the benchmark-average panel retains the negative task-accuracy–alignment trajectory observed in the summary figure.

This detailed breakdown shows that the aggregate increase in mean alignment is broad across evaluated benchmarks rather than confined to one measurement modality. It also shows that the size and consistency of the increase vary by benchmark and finetuning dataset, while the average trend remains an increase in alignment alongside decreasing task evaluation accuracy.

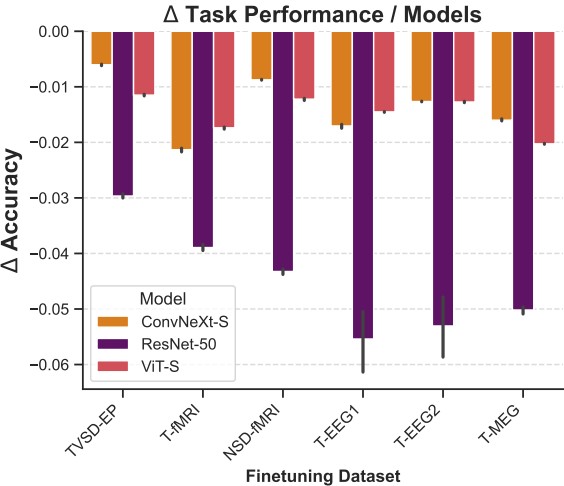

*Figure S45.* Neural fine-tuning is accompanied by a decrease in task (ImageNet) classification accuracy.

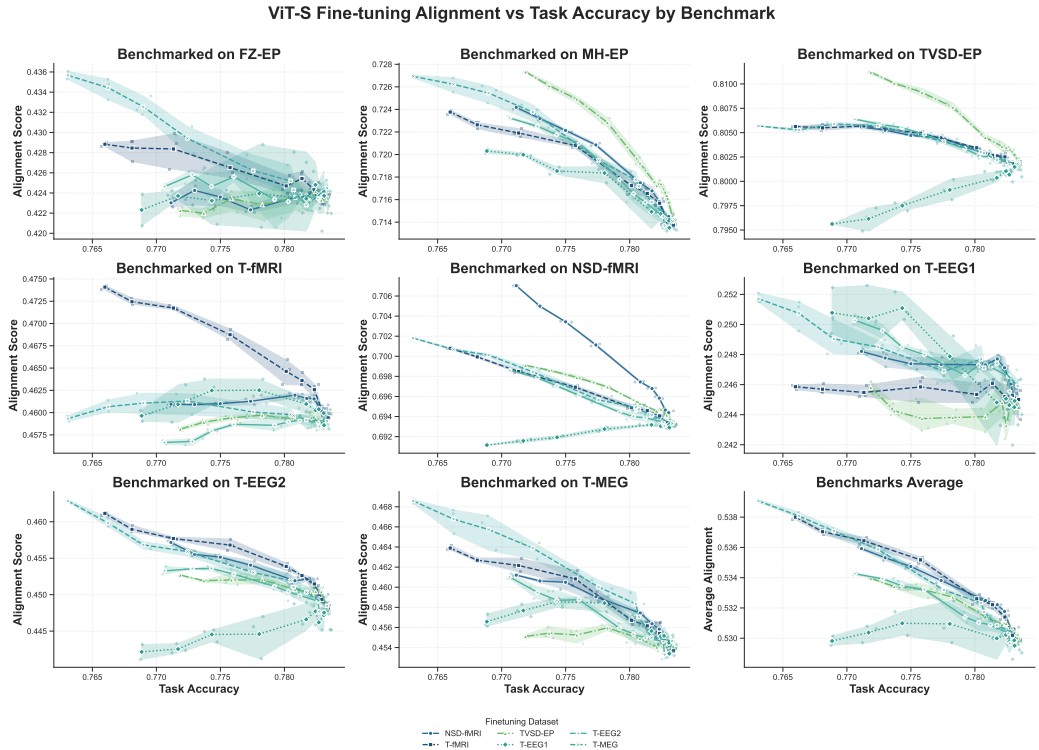

*Figure S46.* Task evaluation accuracy versus neural alignment ($r_{\mathrm{NC}}$) for ViT-S data-fraction sweeps, separated by testing benchmark and finetuning dataset; the final panel averages alignment across benchmarks.

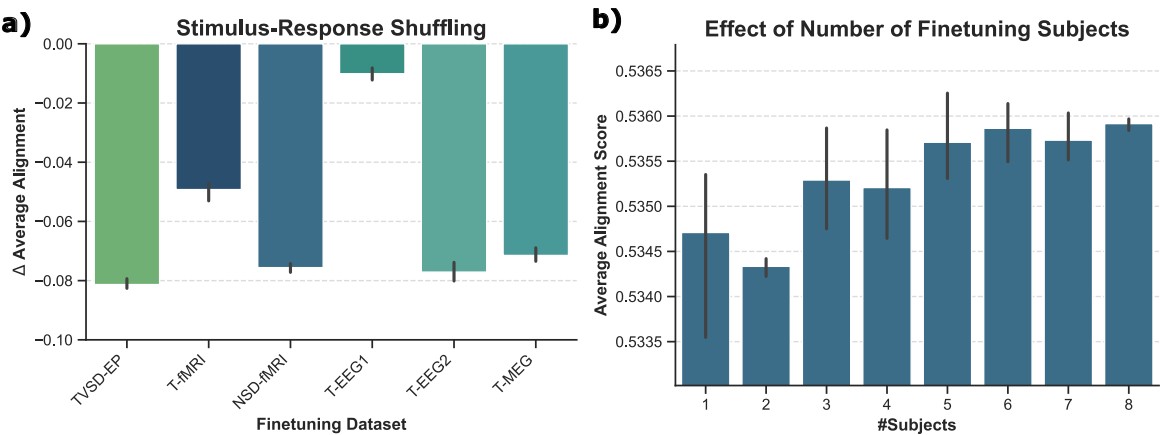

*Figure S47.* **Fine-tuning ablations.** (a) Permuting the stimulus–response correspondence by randomly shuffling pairs before fine-tuning reduces alignment, indicating the gains depend on meaningful supervision; a dataset-wise breakdown is shown in Fig. S48. (b) Average alignment across benchmarks as a function of the number of NSD subjects used for fine-tuning; a dataset-wise breakdown is shown in Fig. S49.

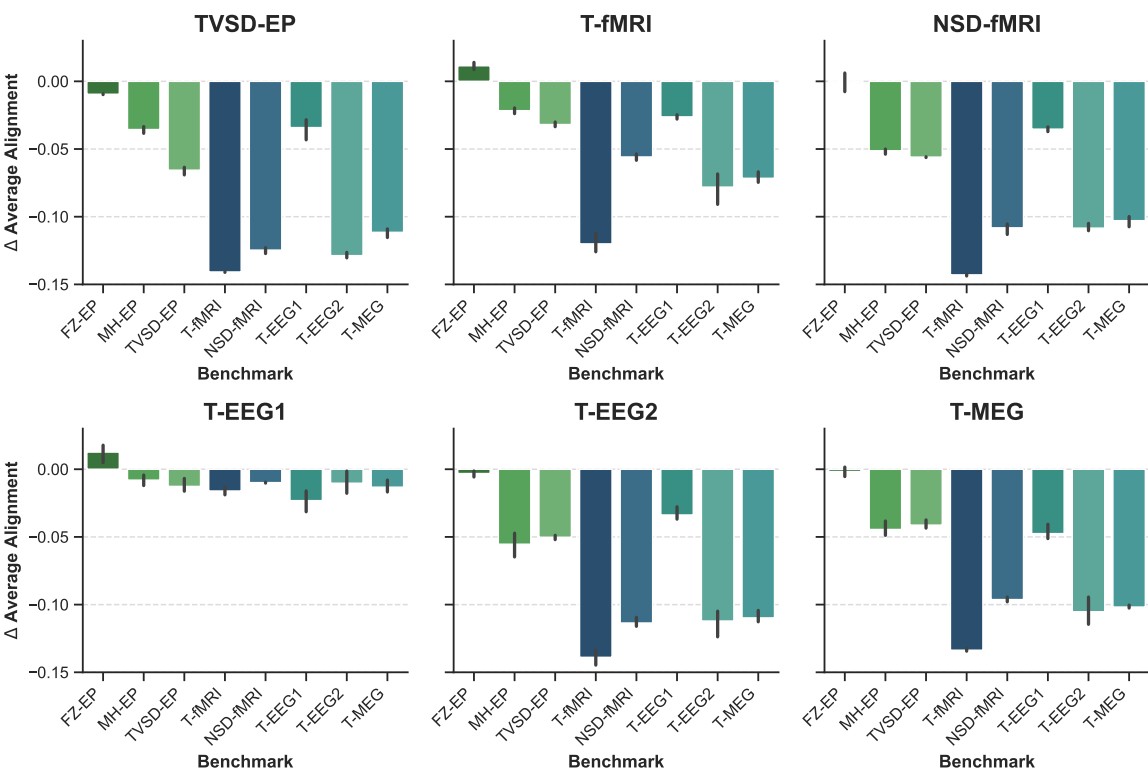

*Figure S48.* Detailed breakdown of permuted-fitting experiments. We find that performance drops not only on the fine-tuned dataset but also on other benchmarks, indicating that shuffling deteriorates the learned representation rather than merely disrupting within-dataset mapping.

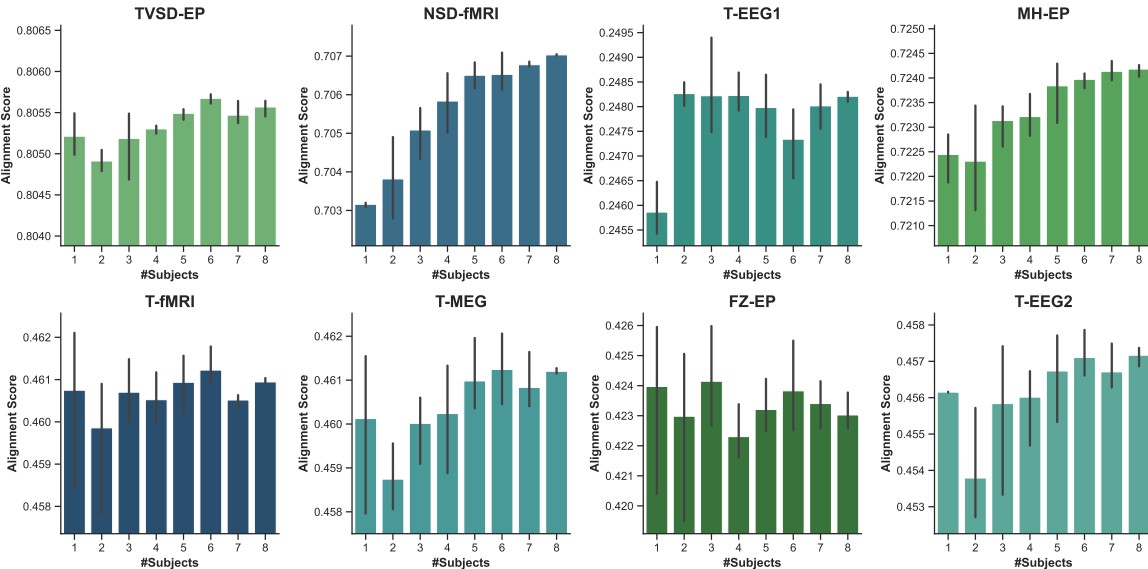

*Figure S49.* Fine-tuning on NSD-fMRI with more subjects further improves alignment, with gains driven primarily by NSD itself. We fine-tune on a subset of NSD subjects and evaluate on all benchmarks; for NSD, test performance is measured on held-out stimuli as before (and thus includes the fine-tuned subjects but not the held-out test stimulus set).

## H. Fine-Tuning Hyperparameter and Model-Size Sensitivity

**Model size.**   We first test whether the neural-alignment gain from fine-tuning varies with ViT backbone size at the common setting of 20 epochs and learning rate $10^{-5}$. Figure S51 resolves each combination of fine-tuning dataset and evaluation benchmark, and Figure S50 averages over fine-tuning datasets. The benchmark-average gain is largest for ViT-L ($\Delta r_{\mathrm{NC}} = 0.00727 \pm 0.00088$ SEM), followed by ViT-S ($0.00582 \pm 0.00137$) and ViT-B ($0.00368 \pm 0.00085$); model size therefore does not yield a monotonic improvement. Capacity helps most clearly for T-fMRI, where mean gains increase from ViT-S to ViT-B to ViT-L ($0.00327$, $0.01491$, and $0.01907$). Conversely, MH-EP and T-MEG favor ViT-S on average ($0.01110$ and $0.00784$, respectively), showing that the benefit of the larger model depends on the evaluation benchmark.

**Hyperparameter sweep.**   Figure S52 provides evidence for the 20-epoch, $10^{-5}$ learning-rate setting used in the fine-tuning experiments. This setting yields a benchmark-average gain of $\Delta r_{\mathrm{NC}} = 0.00591 \pm 0.00073$ SEM, which is effectively identical to the largest observed gain at 50 epochs and $5 \times 10^{-6}$ ($0.00594 \pm 0.00086$; difference $= 0.00003$), while requiring less than half as many training epochs. It is also comparable to 100 epochs at $10^{-6}$ ($0.00581 \pm 0.00070$). Thus, 20 epochs at $10^{-5}$ provides a near-maximal alignment gain with the shortest training schedule among the top-performing settings. In contrast, stronger updates substantially reduce alignment: 100 epochs at $5 \times 10^{-4}$ produces $\Delta r_{\mathrm{NC}} = -0.05775 \pm 0.00823$. Figure S53 shows that the precise best low-rate setting differs by fine-tuning source, but the drop at larger learning rates recurs across datasets. Figure S54 further localizes this drop: T-fMRI, T-EEG2, and T-MEG decrease most strongly as learning rate and training duration increase, whereas FZ-EP remains comparatively tolerant of larger learning rates.

**Matching and transfer evaluations.**   Figure S55 shows that aggressive updates also harm alignment when models are evaluated on the dataset used for fine-tuning, especially for T-fMRI, T-EEG2, and T-MEG. Figure S56 summarizes this effect by transfer relationship. At the selected setting of 20 epochs and learning rate $10^{-5}$, gains are positive for same-dataset ($0.0112$), same-modality ($0.0058$), and cross-modality ($0.0049$) comparisons. At 100 epochs and learning rate $5 \times 10^{-4}$, all three become negative, with the largest decrease in the same-dataset comparison ($-0.1615$, compared with $-0.0511$ for same-modality and $-0.0404$ for cross-modality comparisons). Thus, the selected configuration combines near-maximal average performance with positive transfer across all comparison types, whereas stronger updating generally degrades alignment. Overall, the model-size analysis indicates benchmark-dependent capacity effects, and the hyperparameter sweep supports the 20-epoch, $10^{-5}$ setting used for the fine-tuning experiments.

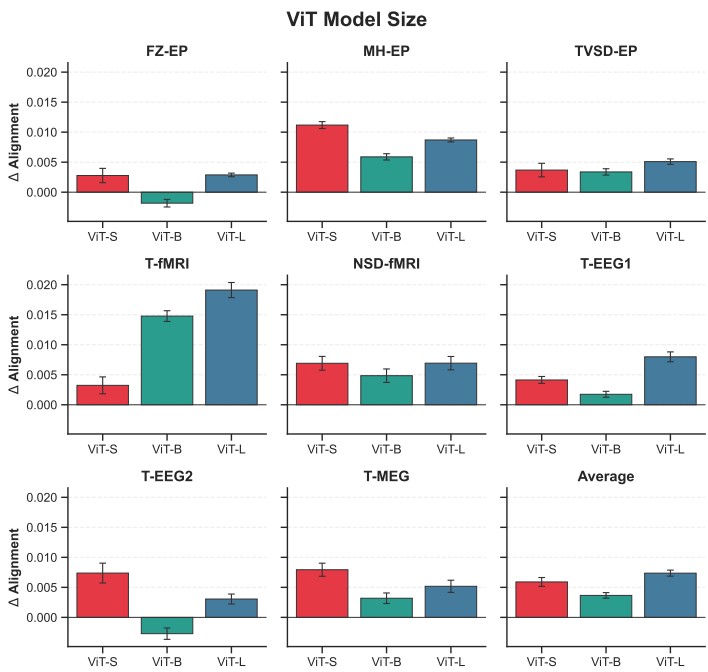

*Figure S50.* Change in neural alignment, $\Delta r_{\mathrm{NC}}$, for ViT model sizes at the shared training setting, averaged across six fine-tuning datasets and three seeds. Error bars indicate SEM across fine-tuning datasets and seeds ($n = 18$).

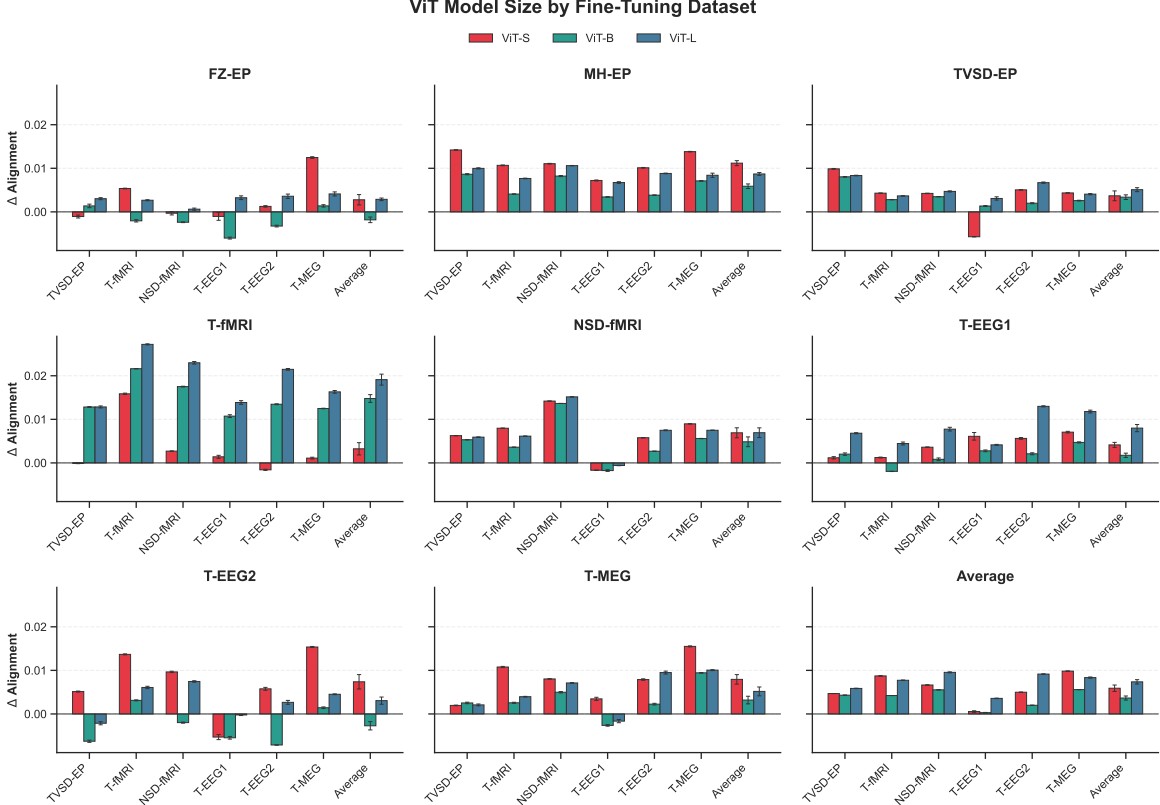

*Figure S51.* Change in neural alignment, $\Delta r_{\mathrm{NC}}$, for ViT model sizes at the shared training setting, averaged across 6 fine-tuning datasets and 3 seeds. Error bars: SEM across 6 datasets × 3 seeds.

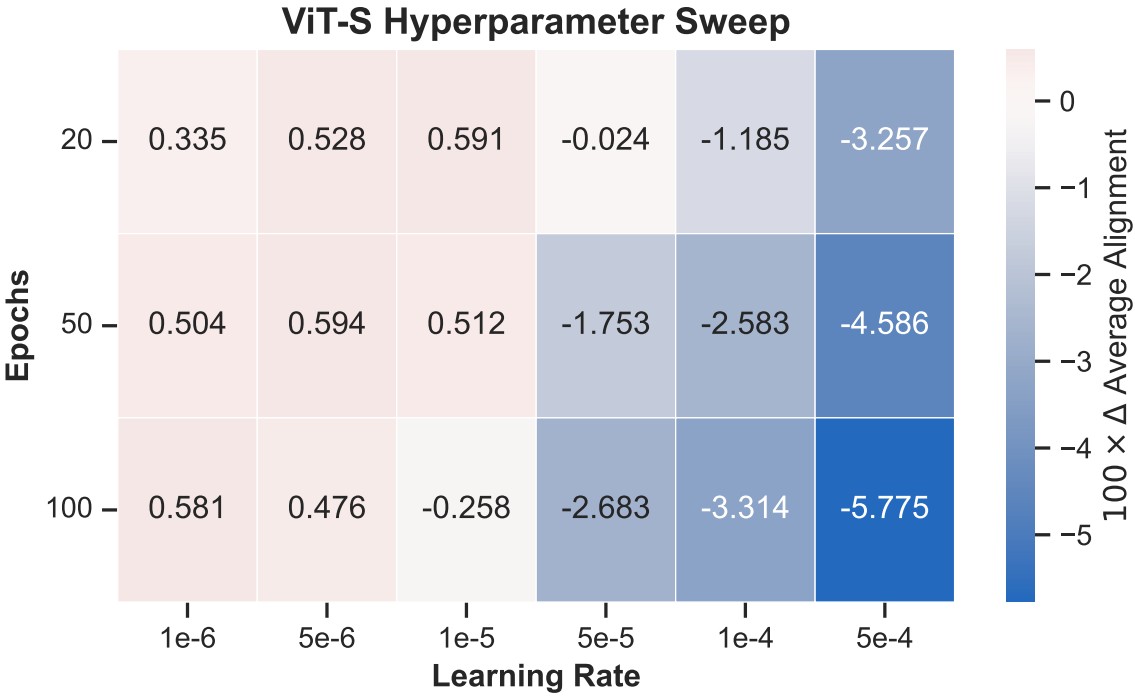

*Figure S52.* ViT-S hyperparameter sweep showing benchmark-average $100 \times \Delta r_{\mathrm{NC}}$ for each epoch and learning-rate setting, averaged across fine-tuning datasets and seeds.

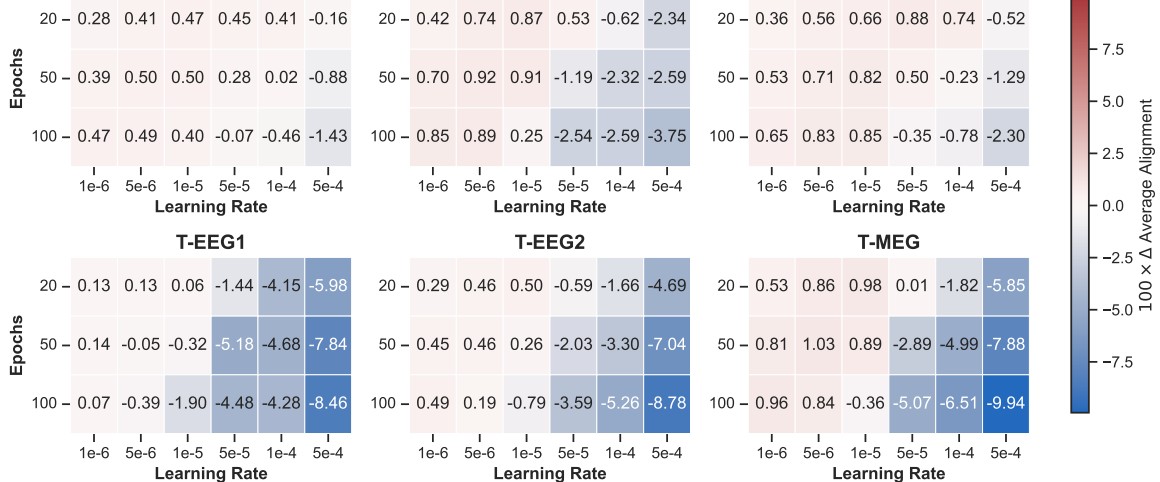

*Figure S53.* Benchmark-average $100 \times \Delta r_{\mathrm{NC}}$ in the ViT-S sweep, shown separately for each fine-tuning dataset and averaged across seeds.

### ViT-S Hyperparameter Sweep by Evaluation Benchmark

*Figure S54.* ViT-S hyperparameter effects, $100 \times \Delta r_{\mathrm{NC}}$, separated by evaluation benchmark and averaged across fine-tuning datasets and seeds.

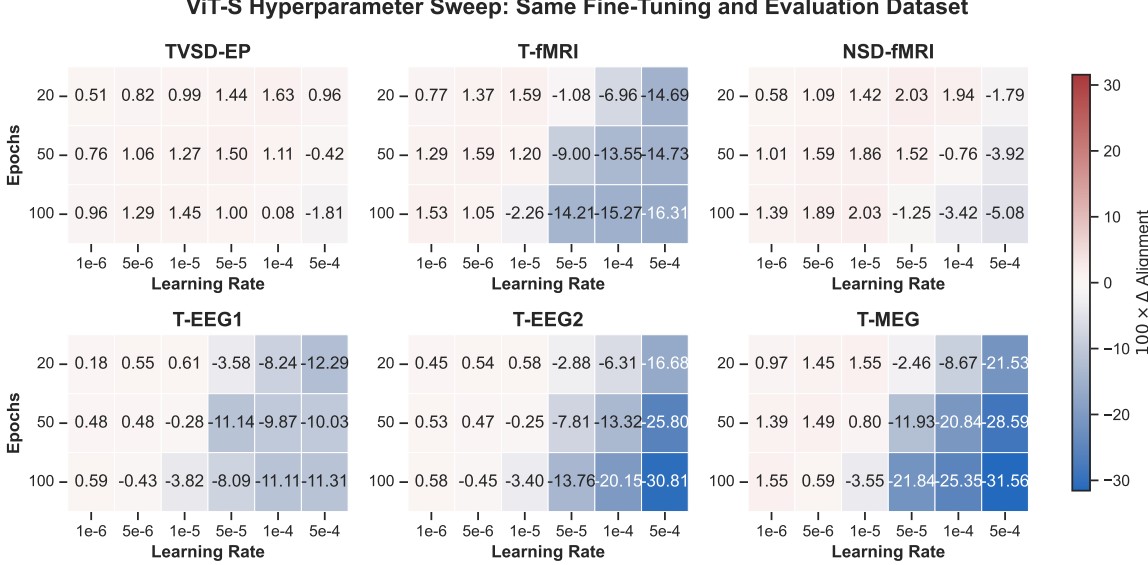

*Figure S55.* ViT-S hyperparameter effects, $100 \times \Delta r_{\mathrm{NC}}$, when the fine-tuning and evaluation dataset match, averaged across seeds.

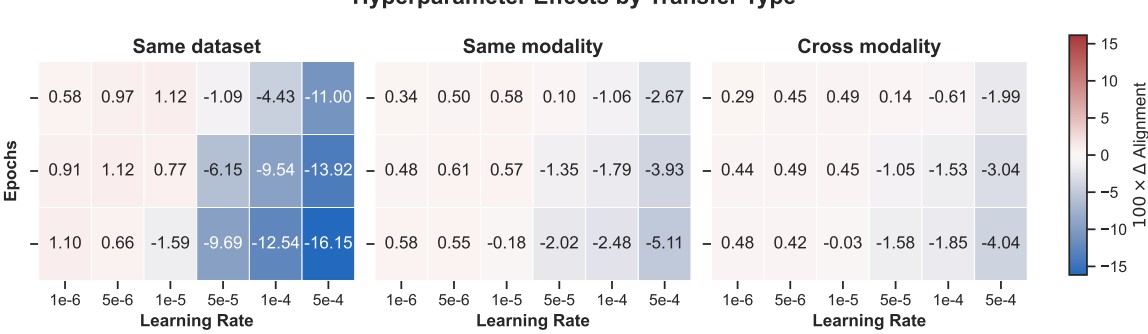

*Figure S56.* ViT-S hyperparameter effects, $100 \times \Delta r_{\mathrm{NC}}$, grouped into same-dataset, same-modality, and cross-modality transfer comparisons.

# I. Parameter-Efficient Neural Fine-Tuning With LoRA

**LoRA implementation.** We apply low-rank updates to the attention projection and both feed-forward transformations within each transformer block, while the classification head remains trainable. All LoRA runs use a scaling factor of 32 and adapter dropout of 0.1, introduce no adapted bias terms, and exclude the neural readout modules from low-rank adaptation. The sweep varies adapter rank over $\{8, 16, 32\}$ and learning rate over $\{10^{-5}, 10^{-4}, 10^{-3}\}$ at 20 epochs.

**LoRA hyperparameter sensitivity.** Figure S57 shows that the LoRA learning rate has a larger effect than rank across fine-tuning datasets. Learning rate $10^{-4}$ gives the highest benchmark-average gain for four of six datasets, while NSD-fMRI favors $10^{-3}$ and T-EEG1 favors $10^{-5}$. When averaged across datasets, the largest gain occurs at $r = 32$ and learning rate $10^{-4}$ ($\Delta r_{\mathrm{NC}} = 0.00421$), closely followed by the other ranks at the same learning rate (0.00417 for $r = 16$ and 0.00408 for $r = 8$). Figure S58 shows the same learning-rate dependence across evaluation benchmarks: $10^{-4}$ generally supports positive transfer, whereas $10^{-3}$ substantially reduces average neural alignment. Figure S59 further shows that the optimal setting is not uniform when fine-tuning and evaluation use the same dataset, motivating the use of a shared overall LoRA setting for the main comparison rather than selecting a configuration independently for each reported outcome.

**Dataset-specific sensitivity analysis.** Figure S61 reports the post-hoc upper-bound comparison in which each fine-tuning dataset uses its highest-scoring LoRA setting. This descriptive analysis increases the mean LoRA gain to $\Delta r_{\mathrm{NC}} = 0.00680$, compared with 0.00591 for fixed full fine-tuning, but it uses the reported evaluation outcomes for selection and therefore does not define the main comparison.

**Shared LoRA versus full fine-tuning.** Figure S60 compares the overall LoRA setting with full fine-tuning for each fine-tuning dataset and evaluation benchmark. Overall LoRA produces larger benchmark-average gains after fine-tuning on TVSD-EP, T-fMRI, NSD-fMRI, and T-MEG, is slightly lower for T-EEG2, and is substantially lower for T-EEG1 ($\Delta r_{\mathrm{NC}}$ difference $= -0.01155$). Consequently, the shared LoRA setting is competitive in several transfer regimes, but full fine-tuning has the larger average gain across datasets and benchmarks.

### LoRA Hyperparameter Sweep by Fine-Tuning Dataset

*Figure S57.* Benchmark-average $100 \times \Delta r_{\mathrm{NC}}$ for each LoRA rank and learning rate, shown separately for each fine-tuning dataset. Learning rate has a stronger influence than rank, with $10^{-4}$ performing best for most datasets.

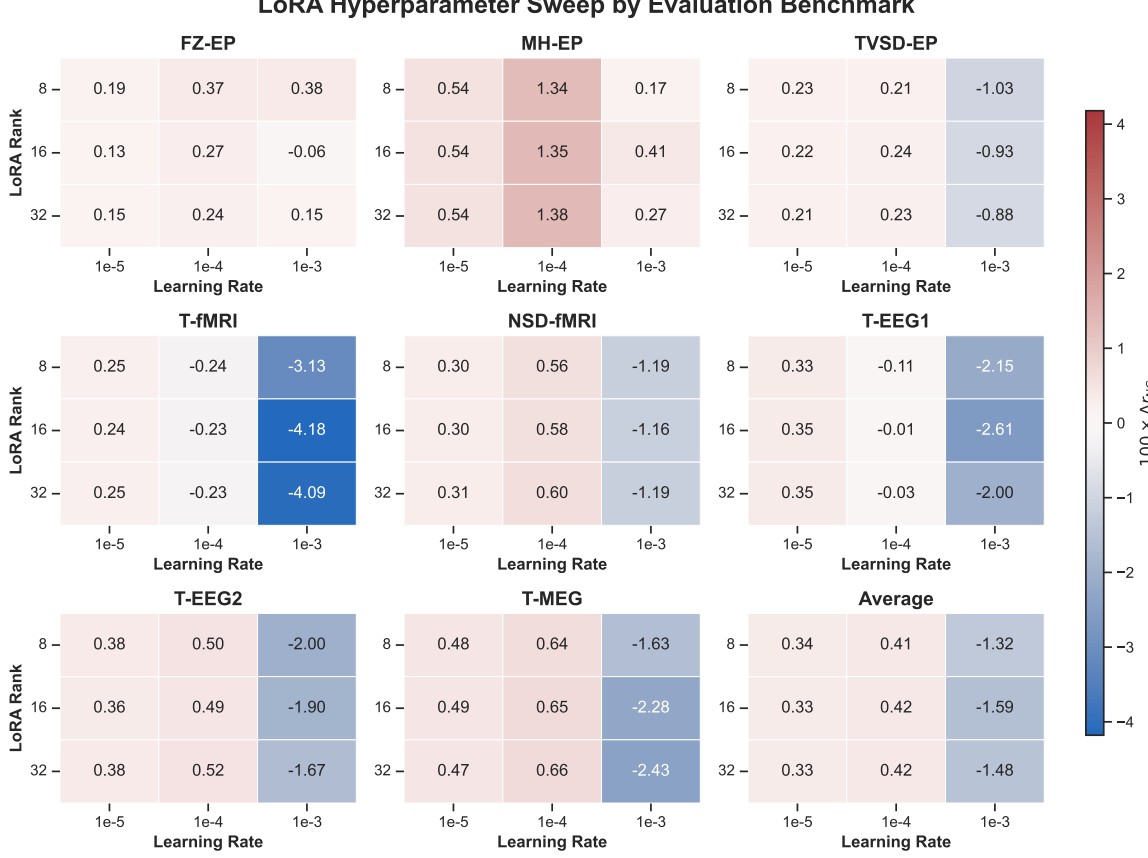

*Figure S58.* LoRA hyperparameter effects, $100 \times \Delta r_{\mathrm{NC}}$, separated by evaluation benchmark and averaged across fine-tuning datasets. The shared $10^{-4}$ setting gives the strongest average transfer, whereas $10^{-3}$ generally reduces alignment.

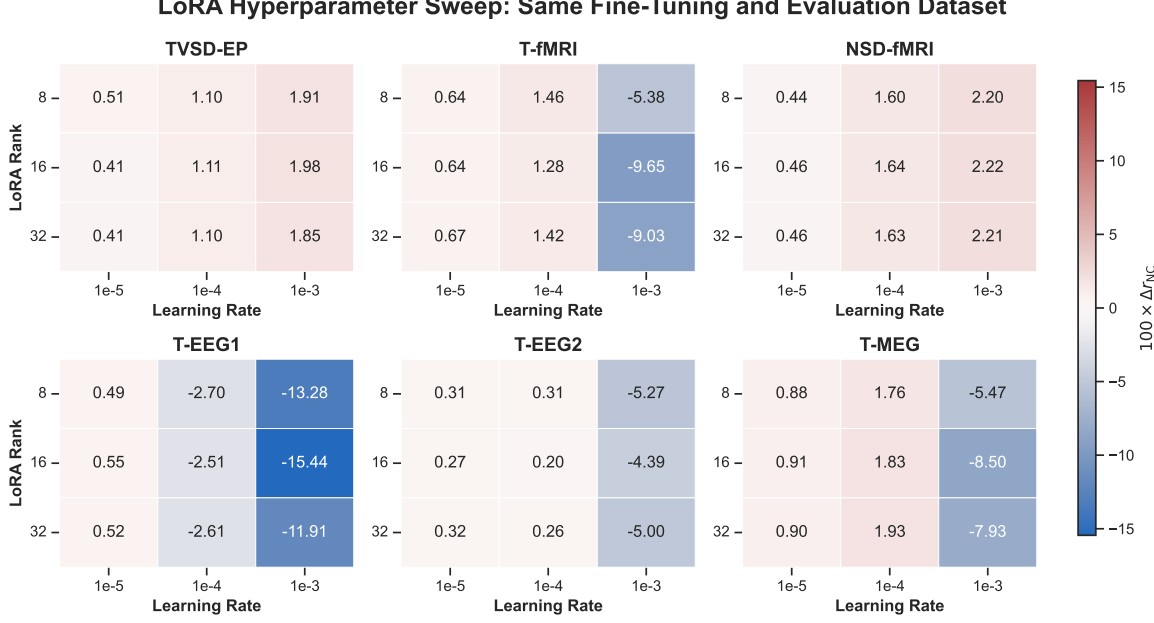

*Figure S59.* LoRA hyperparameter effects, $100 \times \Delta r_{\mathrm{NC}}$, when the fine-tuning and evaluation dataset match. Matching-dataset gains remain sensitive to learning rate and do not support one dataset-specific optimum as a general evaluation setting.

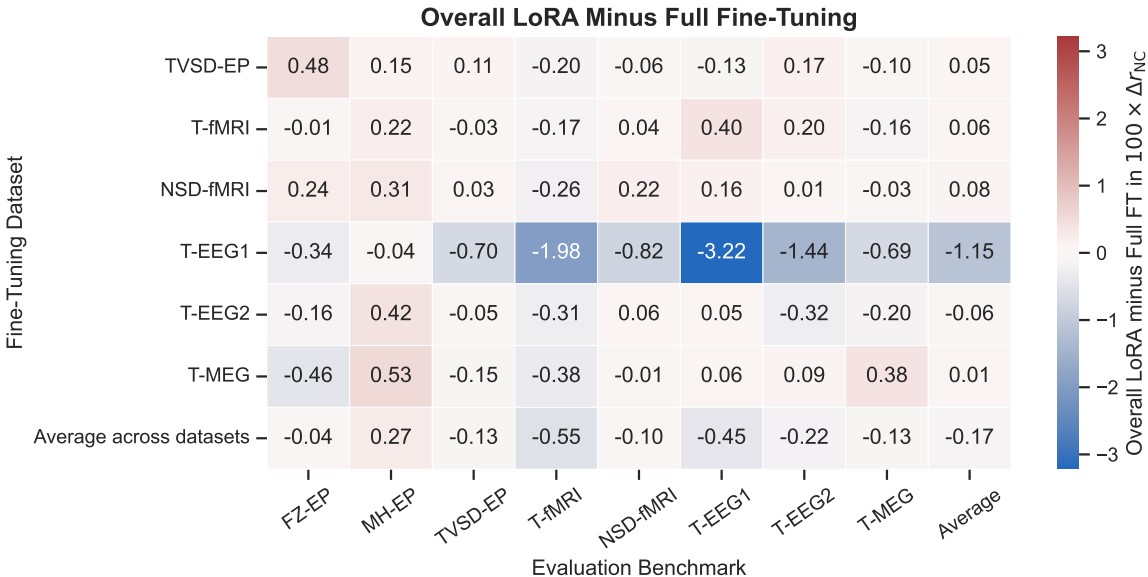

*Figure S60.* Difference in $100 \times \Delta r_{\mathrm{NC}}$ between overall LoRA (20 epochs, $r = 32$, learning rate $10^{-4}$) and full fine-tuning (20 epochs, learning rate $10^{-5}$) across fine-tuning datasets and evaluation benchmarks. Full fine-tuning has the larger mean gain, driven primarily by the T-EEG1 fine-tuning condition.

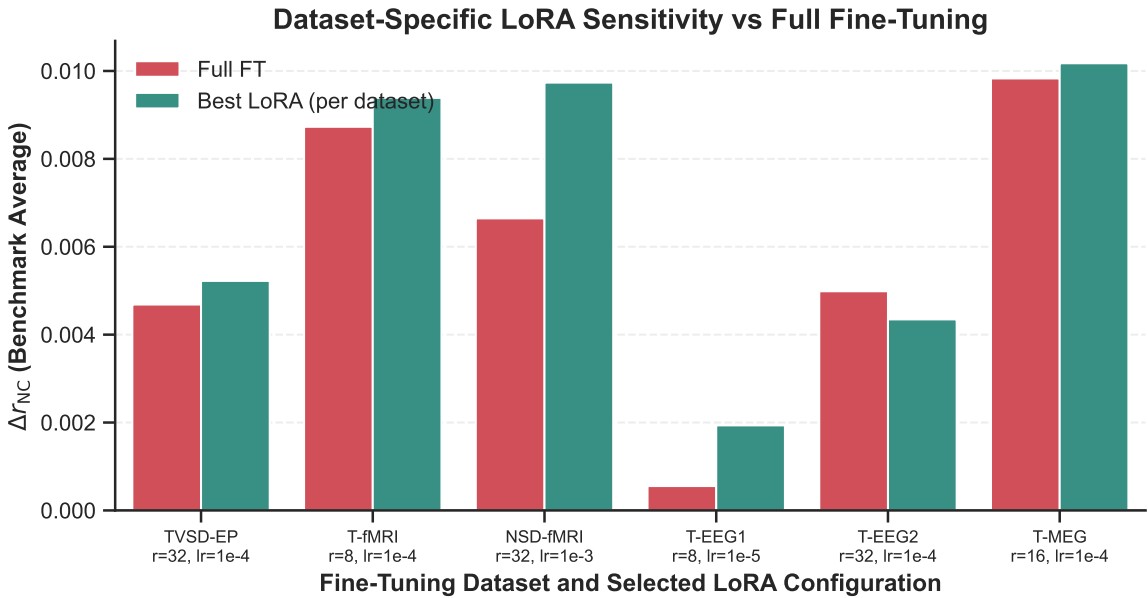

*Figure S61.* Benchmark-average $\Delta r_{\mathrm{NC}}$ for post-hoc dataset-specific LoRA selections and fixed full fine-tuning at 20 epochs and learning rate $10^{-5}$. Dataset-specific selection raises LoRA's descriptive upper bound but is not used for the primary comparison.

## J. Mapping scaling and readout comparisons

Across the fMRI benchmarks, attention-based probes match or exceed linear mappings as more neural data becomes available. On TVSD-EP, both attention variants overtake the linear baseline at moderate-to-large $F$, with the multi-subject probe showing the steepest scaling (largest exponent $\alpha$), indicating faster returns per additional sample. On NSD-fMRI, the multi-subject variant delivers the most consistent improvements at larger $F$, while the single-subject attention probe is competitive but not uniformly better than linear. On T-fMRI, linear mapping performs best in the lowest-data regime, but attention-based mappings catch up and surpass it at larger sample sizes, again with attention-multi trending highest.

For EEG/MEG, scaling remains monotonic but is noticeably slower: the exponents $\alpha$ are smaller (e.g., $\sim 0.05$ for T-EEG1), implying that substantially more fitting data is required to achieve comparable gains in $S$. Correspondingly, differences between mapping families are attenuated. On T-EEG1 and T-MEG, linear mapping is consistently strong and attention-based variants yield only marginal changes, whereas on T-EEG2 the attention-multi probe scales more rapidly and can achieve the best performance at higher $F$ despite underperforming in the lowest-data regime.

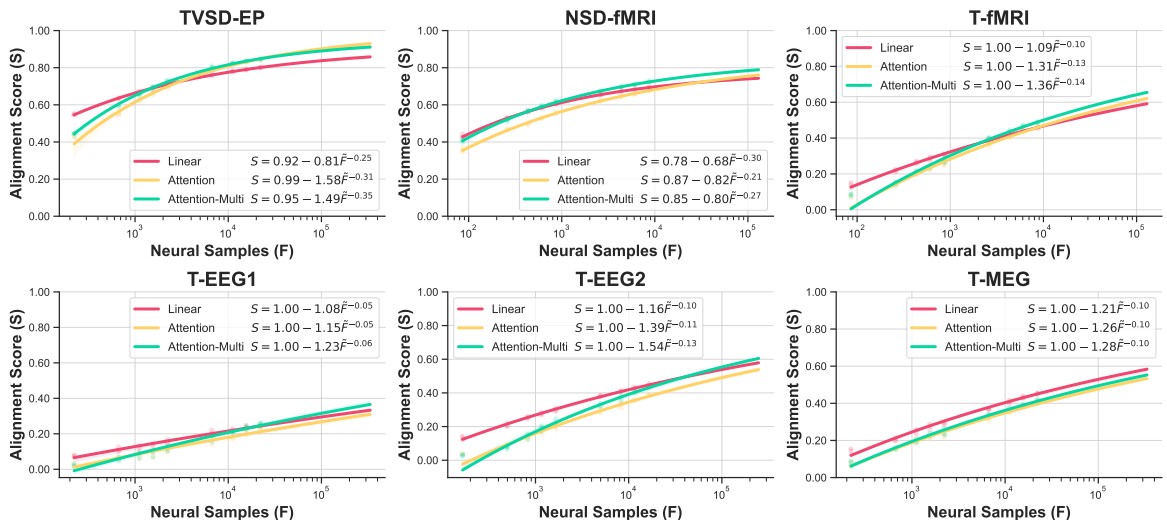

*Figure S62.* **Scaling neural samples for fitting a mapping.** Across benchmarks, increasing the number of stimulus–neural pairs used to fit the mapping yields consistent gains in alignment score. Points show empirical results, and solid curves show scaling-law fits $S(F) = E - AF^{-\alpha}$ for linear, attention, and multi-subject attention probes. For numerical stability, we fit using the rescaled variable $\tilde{F} = F/10$.

### J.1. Additional Readout Baselines

To compare attention-based mappings with simpler alternatives, we evaluated a factorized linear (low-rank) readout and a shallow MLP readout. Both operate on the mean-pooled encoded representation: the low-rank readout uses two linear projections, while the shallow MLP adds one nonlinear hidden layer before the subject-specific neural prediction layer.

For the low-rank and shallow MLP readouts, given token features $\mathbf{Z}_\ell(\mathbf{x}) \in \mathbb{R}^{T \times d}$ from layer $\ell$, both readouts first compute the mean-pooled representation

$$\bar{\mathbf{z}}_\ell(\mathbf{x}) = \frac{1}{T} \sum_{t=1}^{T} \mathbf{Z}_\ell(\mathbf{x})_{t,:} \ \in \ \mathbb{R}^d. \tag{12}$$

**Low-rank readout.** For each subject $s$ and ROI $r$, the prediction is produced by a factorized linear map with rank $\rho$,

$$\widehat{\mathbf{y}}_{r,s}(\mathbf{x}) = B_{r,s} A_{r,s} \bar{\mathbf{z}}_\ell(\mathbf{x}) + \mathbf{b}_{r,s}, \qquad A_{r,s} \in \mathbb{R}^{\rho \times d}, \quad B_{r,s} \in \mathbb{R}^{n_{r,s} \times \rho}, \tag{13}$$

where $n_{r,s}$ is the number of recorded units for $(r, s)$, the first projection has no bias, and $\mathbf{b}_{r,s}$ is the subject-/ROI-specific bias.

*Table S4.* Noise-ceiled Pearson $r$ across benchmarks for different mapping readouts. SS = single-subject; MS = multi-subject. Scores are averaged across subjects, ROIs, and three random seeds. Parameter counts are summarized with the geometric mean across all instantiations of each probe type. Best per row is shown in **bold**.

| Benchmark | Linear (SS) | Shallow MLP (SS) | Shallow MLP (MS) | Low-rank (SS) | Attention (SS) | Attention (MS) |
|---|---|---|---|---|---|---|
| TVSD-EP | 0.801 | 0.665 | 0.673 | 0.656 | 0.844 | **0.846** |
| NSD-fMRI | 0.693 | 0.625 | 0.658 | 0.623 | 0.677 | **0.723** |
| T-fMRI | 0.458 | 0.390 | 0.403 | 0.370 | 0.461 | **0.487** |
| T-EEG1 | 0.245 | 0.220 | 0.217 | 0.216 | 0.234 | **0.257** |
| T-EEG2 | **0.447** | 0.367 | 0.395 | 0.365 | 0.390 | 0.434 |
| T-MEG | **0.453** | 0.374 | 0.386 | 0.339 | 0.410 | 0.414 |
| Average | 0.516 | 0.440 | 0.455 | 0.428 | 0.503 | **0.527** |
| Geom. #Parameters ($\times 10^7$) | 23.03 | 2.53 | 2.06 | 3.31 | 2.53 | **2.06** |

**Shallow MLP readout.** Mean-pooled features are transformed by a one-hidden-layer MLP before a subject- and ROI-specific linear output head:

$$\mathbf{h}(\mathbf{x}) = W_2 \, \text{Dropout}_p(\text{GELU}(W_1 \, \text{LN}(\bar{\mathbf{z}}_\ell(\mathbf{x})) + \mathbf{b}_1)) + \mathbf{b}_2, \qquad \widehat{\mathbf{y}}_{r,s}(\mathbf{x}) = C_{r,s} \, \mathbf{h}(\mathbf{x}) + \mathbf{c}_{r,s}. \tag{14}$$

The dropout probability $p$ is selected per benchmark configuration.

Table S4 reports noise-ceiled Pearson $r$ across benchmarks for these baselines alongside the linear and attention-based readouts. Across the six benchmarks, the multi-subject attention probe achieves the highest average alignment, while the low-rank and shallow MLP readouts trail both the linear and attention variants. This indicates that the gains of the attention-based mapping are not simply a function of added readout capacity or non-linearity acting on pooled features.

## J.2. Multi-subject sharing improves the Pareto frontier

Comparing single-subject and multi-subject attention probes, the multi-subject variant tends to shift the parameter–performance trade-off in a favorable direction (top-left). Because the cross-attention block is shared across subjects, the model can pool statistical strength across subjects while keeping the subject-specific components small. This sharing yields a more data-efficient mapping that remains compact yet competitive, and in aggregate (as reflected by the centroids) matches or improves upon single-subject attention while operating at similar parameter scales. Overall, Figure S63 highlights that carefully structured parameter sharing can deliver strong alignment without relying on large, ROI-specific linear readouts.

Both attention-probe variants operate in a low-parameter regime while achieving alignment scores comparable to linear probes. This efficiency is expected: linear readouts scale with the product of feature dimensionality and the number of recorded channels, whereas the attention probe learns a data-driven dimensionality reduction, amortizes most capacity in a shared cross-attention module, and uses lightweight subject-/ROI-specific heads. Consistent with this design, attention probes achieve competitive predictivity without the large per-ROI parameter counts characteristic of linear decoders. The centroid markers summarize this trend at the family level, highlighting that attention-based mappings offer a favorable trade-off between parameter count and alignment.

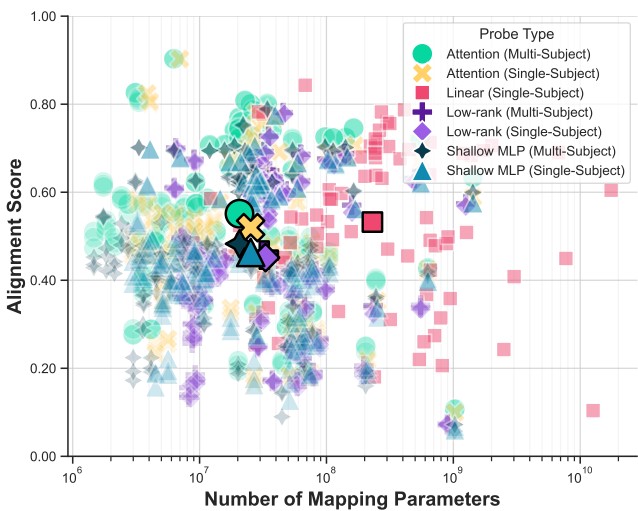

*Figure S63.* **Alignment vs. mapping parameters.** Each point corresponds to a fitted probe for a single ROI from one dataset, with larger symbols showing probe-family centroids. Attention-based probes achieve comparable alignment to linear probes with substantially fewer parameters, and the multi-subject variant further improves the Pareto frontier. The factorized low-rank and shallow MLP readouts (single- and multi-subject) operate at parameter counts comparable to the attention probes and roughly an order of magnitude below LINEAR–SS, but achieve lower alignment than both linear and attention readouts, indicating that capacity reduction alone does not account for the attention probe's gains.

# K. Attention-Based Readout Architecture and Training Details

Table S5 reports the cross-attention probe architecture and training hyperparameters. The probe design is inspired by the Perceiver and Perceiver IO architectures (Jaegle et al., 2021; 2022). To limit model capacity and mitigate overfitting, we keep the attention module lightweight and adjust the learning rate, number of training epochs, and dropout on a per-dataset basis, as indicated directly in Table S6.

*Table S5.* Attention probe hyperparameters (default configuration).

| **Probe architecture** | |
| --- | --- |
| Model dimension ($d_{\text{model}}$) | 384 |
| Number of attention heads | 16 |
| Feedforward dimension | 0 (no fully connected layers) |
| Number of latent query tokens | 16 |
| Cross-attention layers | 1 |
| Token encoder layers | 0 |
| Query self-attention | Disabled (no self-attention layers) |
| Positional encoding | Sinusoidal |
| Head type | Linear |
| Dropout | 0.3 |
| **Optimization** | |
| Optimizer | AdamW |
| Learning rate | $10^{-4}$ |
| Weight decay | $10^{-4}$ |
| Learning rate schedule | Cosine |
| Gradient clipping | 1.0 |
| Precision | bf16-mixed |
| Batch size | 256 |
| Epochs | 10 |

*Table S6.* Dataset-specific overrides for attention probe training.

| Dataset | Epochs | Learning rate | Dropout |
| --- | --- | --- | --- |
| THINGS EEG-1 | 10 | $10^{-4}$ | 0.5 |
| THINGS EEG-2 | 10 | $10^{-4}$ | 0.5 |
| NSD fMRI | 20 | $10^{-3}$ | 0.8 |
| THINGS MEG | 30 | $10^{-4}$ | 0.8 |
| THINGS fMRI | 30 | $10^{-4}$ | 0.8 |
| TVSD | 30 | $10^{-4}$ | 0.3 |

