# OpenReview forum: "Multimodal Scaling Laws for Task & Data-Optimized Models of Visual Cortex"
_ICML.cc/2026/Conference — ICML 2026 regular_

### Official Review · Reviewer_twXG · 2026-03-05

**Soundness:** 4
**Presentation:** 3
**Significance:** 3
**Originality:** 3
**Overall Recommendation:** 5
**Confidence:** 4

**Summary:**

The paper studies how model–brain alignment scales as a function of three aspects: (1) pretraining scale of vision backbones (e.g., model/compute/data), (2) neural supervision via brain-driven fine-tuning of the backbone, and (3) the capacity/parameterization of the mapping used to predict brain responses from model features. Alignment is operationalized using an encoding-model pipeline: for each neural dataset and brain target (e.g., ROI/electrode/site), the authors fit stimulus-to-response mappings from model layer features and evaluate held-out predictivity. Using a large suite of vision models and multiple neural recording datasets, the paper fits and compares scaling laws for alignment, tests how alignment changes with additional neural data for fitting mappings, and evaluates the effect of neural fine-tuning under controlled training procedures. It also introduces and analyzes attention-based probes as an alternative mapping family to standard linear probes.

**Compliance With Llm Reviewing Policy:**

Affirmed.

**Final Justification:**

Overall, my questions have been adequately addressed, and the rebuttal increases my confidence in the paper’s broader claim. I am therefore raising my score from 4 (Weak accept) to 5 (Accept).

**Key Questions For Authors:**

1. Beyond providing a large-scale benchmark and empirical scaling curves, what is the exact impact this work is intended to have on NeuroAI? Concretely, what decisions should practitioners make differently after reading this paper, and why is this important relative to existing brain-prediction evaluations?

2. What is the rationale and prior-literature support for operationalizing “model–brain alignment” primarily via this encoding-model protocol and the specific choices such as committed layers, noise normalization, and the probe families? Please clarify whether the goal is representation-level alignment (favoring simpler probes) or predictive upper bounds (allowing richer probes), since the probe family changes the operational meaning of alignment. Relatedly, do the main scaling trends persist under alternative plausible alignment notions or mapping families, or are they specific to the protocol used here?

3. Different neural modalities differ substantially in temporal/spatial resolution, number of channels/ROIs, preprocessing abstractions (e.g., fMRI betas vs EEG/MEG flattening), and what aspects of brain function they reflect. How do the authors ensure fair cross-modal comparison and aggregation? In particular, how are datasets and modalities weighted in the aggregate analyses, and would the conclusions change under alternative normalization, weighting, or within-modality stratification schemes?

**Limitations:**

yes

**Strengths And Weaknesses:**

## Strength

1. The paper is original in scope and synthesis. It systematically studies “model–brain alignment” scaling across multiple axes and across multiple datasets/modalities, which is relatively rare in NeuroAI work.

2. The study involves a large number of models and datasets, with extensive sweeps and controlled comparisons. The overall empirical pipeline appears carefully executed.

3. The paper provides detailed descriptions of the datasets, preprocessing, alignment pipeline, and analysis protocol, which improves transparency and makes the large empirical study easier to follow and assess.

4. The paper is well written and well structured. The figures/tables are visually clean, intuitive, and make the main empirical patterns easy to inspect.

5. Key trends are communicated clearly through the visualizations.

## Weaknesses

1. The paper frames its findings as “multimodal scaling laws” for model–brain alignment, but the reported trends are established under one specific alignment protocol and a limited set of mapping choices. Without stronger evidence that the main conclusions are robust to alternative alignment definitions, mapping families, or protocol variations, it is difficult to tell whether the paper has uncovered genuinely general NeuroAI scaling laws or primarily documented benchmark-specific empirical regularities.

2. The paper does not clearly articulate why “model–brain alignment” is the right entry point for studying “NeuroAI scaling laws,” and what unique scientific/ML value this framing provides beyond benchmarking. This weakens the scope of the conclusions and makes the impact feel narrower than the technical effort suggests.

3. The paper could do more to justify why this particular alignment metric/protocol is the appropriate target via clearer theoretical intuition and stronger grounding in prior neuro/computational literature, and discussion of alternative plausible alignment notions and why they were not used.

4. The attention-based mapping is compared primarily against a linear mapping, which makes the “parameter efficiency” argument feel only partially established. Without broader comparisons to other compact but more expressive mapping families (e.g., low-rank nonlinear, shallow MLP, or lightweight bilinear variants), it is difficult to assess whether the observed gains are specific to cross-attention or simply reflect moving beyond a linear probe.

5. The title (“Multimodal Scaling Laws for Task & Data-Optimized Models of Visual Cortex”) does not clearly convey what is actually done in the paper, creating a gap between expectation and content.

---

> ### Author Rebuttal · Authors · 2026-03-31
>
> We thank the reviewer for the thoughtful critique and constructive feedback. We will open-source the codebase, benchmark suite, and visualization tools. Below we address the main questions.
> > What is the intended impact on NeuroAI?
>
> Our main aim is to shift brain-predictive modeling from a mainly benchmark-driven exercise to a resource-allocation question: where does additional scale still buy gains, and where do returns already diminish? Across eight datasets and four modalities, we find that scaling generic visual pretraining helps but shows clear diminishing returns, whereas neural supervision, more paired neural data, and better mapping design remain higher-leverage directions. Thus, the practical takeaway is that future progress is less likely to come from scaling standard pretraining alone and more likely to come from improving neural supervision and decoding methodology.
> > Why study “model–brain alignment” in this way?
>
> Model–brain alignment is not meant here as a generic benchmark target, but as the standard operational entry point in task-optimized modeling for testing whether models trained on ecologically relevant visual tasks recover brain-predictive representations. This framing follows prior work showing that task-optimized models can serve as useful in-silico hypotheses of sensory cortex when evaluated by their ability to predict neural responses or match neural representational structure (Yamins2016; Schrimpf2018; Conwell2023). In that sense, alignment is valuable not only as a benchmark, but also as a neuroscientifically grounded criterion for comparing alternative model-building choices. It has also supported downstream scientific applications, including studies of visual selectivity, functional topography, and disorder modeling (e.g., Cerdas2024; Margalit2024; Tang2025; Honarmand2025).
>
> Our paper builds on this literature by asking which scaling axes still matter most once this framework is adopted. We separate pretraining, neural fine-tuning, and mapping, and show that standard visual pretraining exhibits diminishing returns, whereas neural supervision and especially mapping/data improvements remain higher-leverage directions. We will make this scientific and ML motivation more explicit in the revision.
> > Why this protocol, and what notion of alignment does it target?
>
> Our protocol follows the dominant task-optimized modeling setting, where frozen model features are paired with simple mappings to test whether the representation itself is brain-predictive (Yamins2016; Schrimpf2020; Conwell2023). In this framing, committed layers align model stages with brain regions, noise normalization accounts for dataset-specific reliability so scores are more comparable across benchmarks, and simple probes preserve the interpretation of alignment as a property of the representation.
>
> At the same time, we agree that probe choice changes the operational meaning of alignment. We therefore separate two goals: in the main pretraining-scaling analysis, we use a regularized linear mapping to emphasize representation-level correspondence; in the mapping section (Secs. 4.8–4.9), we study richer probe families to move toward predictive upper bounds.
> > Alternative Metrics
>
> Appendix E demonstrates how alignment scales under alternative metrics used in NeuroAI. Figures S20–23 display analyses based on RSA and CKA, including voxel-encoding variants, and the main pretraining-scaling trends are qualitatively preserved. In the revision, we will add explained variance as an additional robustness check.
>
> > Comparison to other compact mapping families
>
> We additionally compared our attention-based readout to shallow MLP and low-rank bilinear mappings at matched parameter budgets. Overall, the multi-subject attention variant achieves the strongest average performance (0.527), outperforming both shallow MLP variants (0.455) and the low-rank bilinear mapping (0.428), while remaining highly parameter-efficient. This suggests that the gains are not simply from using a nonlinear readout, but from the specific structure and sharing strategy of the attention-based mapping. Please see our response to Reviewer **tZ5A** under “Additional Readout Results” for the full table.
> > How do we compare across modalities fairly?
>
> We do not pool raw neural signals across modalities. Each dataset is evaluated with modality-appropriate preprocessing and dataset-specific noise normalization, and most results are reported per benchmark. When we report averages, they are over normalized benchmark-level scores, so modalities are not implicitly weighted by voxel/channel count. See our response to Reviewer **vH7z** (3. and 4.) for details.
>
> Thanks again for your review. We believe these additions firmly solidify the generalizability of our claims: the main trends persist across modalities, alternative metrics, and broader mapping families. We ask the reviewer to kindly consider raising their score when taking these new empirical results into account.

---

> > ### Author Rebuttal · Reviewer_twXG · 2026-04-01
> >
> > Thank you for the detailed rebuttal. The clarifications and added results address my main concerns.
> >
> > In particular, the paper’s intended impact on NeuroAI is now clearer, and the distinction between representation-level alignment and predictive upper bounds is better motivated. The added discussion of alternative metrics, broader readout families beyond the linear probe, and cross-modal aggregation also makes me more confident that the main conclusions are not tied to one specific protocol choice.
> >
> > I still think the framing could be sharpened a bit in revision, especially the title. But at this point these are presentation issues rather than core concerns.
> >
> > Overall, my questions have been adequately addressed, and the rebuttal increases my confidence in the paper’s broader claim. I am therefore raising my score from **4 (Weak accept)** to **5 (Accept)**.

---

### Official Review · Reviewer_tZ5A · 2026-03-06

**Soundness:** 3
**Presentation:** 3
**Significance:** 4
**Originality:** 3
**Overall Recommendation:** 5
**Confidence:** 3

**Summary:**

This paper systematically investigates the model-to-brain alignments of diverse visual models across multiple neural recording modalities. The discovered scaling trends reveal saturated alignment with respect to pretraining data across model architectures, as well as improved alignment after neural finetuning. I believe it can serve as an important step towards understanding scaling in the field of neuro-foundational models.

**Compliance With Llm Reviewing Policy:**

Affirmed.

**Final Justification:**

The paper concerns with an important topic on model-to-brain alignments, and extensive experiments and analysis have been conducted. I believe it is an important work for the field, and I maintain my score as 5.

**Key Questions For Authors:**

1. Can authors comment on how much brain alignment does the model need? Given many SOTA models are not even trained in a bio-plausible way, the aligning potential is fundamentally limited. Plus as it is observed in your experiments, pushing towards alignment is at the cost of decreased accuracy.
2. Why freeze the ROI matching during finetuning? Can the ROI matching be further trained after neural finetuning?
3. Most scaling law plots are with respect to alignment only,and I think jointly alignment and accuracy could reveal more architecture specific trends.
4. It would be interesting to explore representation similarity across modalities, if there are overlapping visual stimuli across datasets.
5. What are found committed layer index with respect to different brain regions, I suppose earlier layers would map to earlier regions.

**Limitations:**

Yes.

**Strengths And Weaknesses:**

Strengths:
- Paper is in general easy to follow
- Comprehensive analysis on alignment across a large number of models, modalities and sizes.
- Investigate cross-attention probes as cost-efficient alternatives to linear probes.

Weaknesses:
- Reliance on existing vision models essentially limits the study to vision stimuli tasks, while the idea should be generalized to other tasks (like audio, speech and motor).
- Fail to acknowledge many recent studies on scaling laws in the field of neuroAI.

Just to name a few:

Willeke et. al. OmniMouse: Scaling properties of multi-modal, multi-task Brain Models on 150B Neural Tokens, ICLR 2026.

Ye et. al. A Generalist Intracortical Motor Decoder. NeurIPS 2025.

---

> ### Author Rebuttal · Authors · 2026-03-31
>
> Thank you for the encouraging review and for the constructive suggestions. We respond to each of your comments below.
>
> > Scope of the study and relation to recent neuro-foundation-model work
>
> We agree that recent large-scale neural foundation models are highly relevant to the broader NeuroAI landscape, and we will discuss them more explicitly in the revision. At the same time, our paper addresses a somewhat different question: we study stimulus-evoked model-to-brain alignment starting from task-optimized visual backbones, rather than neural-to-neural foundation models trained within a single recording modality. We focus on vision because it currently provides the broadest set of large-scale stimulus-response datasets spanning Ephys, fMRI, EEG, and MEG under a common sensory domain. While extending this framework to other modalities is an exciting next step, we restricted our scope to primate visual cortex to ensure rigorous cross-dataset comparability.
>
> > How much alignment is needed?
>
> We agree that perfect alignment is unlikely under current task-trained models, and that alignment should not be pursued without regard to task competence. In our view, the right question is not “maximize alignment at any cost,” but rather “what bottlenecks currently limit the use of these models as in-silico brain models?” Our results suggest that the main bottlenecks are no longer only generic pretraining scale, but also neural supervision and especially the mapping/data regime (Sections 4.3-4.9). We also explicitly document the alignment–accuracy trade-off after neural fine-tuning, and will emphasize this balance more clearly in the discussion.
>
> > Why freeze the ROI-specific readouts during fine-tuning?
>
> We freeze them to keep the fine-tuning stage focused on changing the backbone representation rather than absorbing gains into a large subject/ROI-specific decoder. This makes cross-dataset transfer easier to interpret and avoids the readout dominating the parameter budget, which is especially important because simple linear decoders can be substantially larger than the backbone-side adaptation being studied due to the large number of voxels/timepoints (Figure S28). Furthermore, we empirically observed much better transfer when the subject- and ROI-specific readout heads are held constant.
>
> > Task accuracy plots
>
> Thank you for the suggestion. In the updated manuscript, we will add a section that jointly plots pretraining scaling for both alignment and task accuracy across datasets, which should make it easier to inspect architecture-specific trends and how they evolve with scale.
>
> > Committed layers and hierarchy.
>
> We also agree that the layer-to-region relation is informative. Appendix D includes layerwise progression plots across datasets and ROIs using different metrics (Figs S12-S19), and these recover the expected trend that later brain regions tend to prefer deeper model layers.
>
> > Cross-modal representation similarity
>
> We agree this is an exciting direction, especially within the THINGS ecosystem. In the current paper we focus on scaling laws for model-to-brain predictivity rather than direct brain-to-brain representational comparisons across modalities, but we will highlight this as a promising direction for future work.
>
> Thank you again for your review. We appreciate the reviewer’s supportive assessment and hope these clarifications further strengthen the paper.
>
>
> \
> \
> **Additional Readout Results**
>
> For the benefit of all reviewers, we are also including the results of our new baseline readout experiments here, which further demonstrate the efficiency of our attention-based mapping:
> | Benchmark | Linear (SS) | Attention (SS) | Attention (MS) | Shallow MLP (SS) | Shallow MLP (MS) | Low-rank (SS) |
> | --- | --- | --- | --- | --- | --- | --- |
> | TVSD-EP | 0.801 | 0.844 | **0.846** | 0.665 | 0.673 | 0.656 |
> | NSD-fMRI | 0.693 | 0.677 | **0.723** | 0.625 | 0.658 | 0.623 |
> | T-fMRI | 0.458 | 0.461 | **0.487** | 0.390 | 0.403 | 0.370 |
> | T-EEG1 | 0.245 | 0.234 | **0.257** | 0.220 | 0.217 | 0.216 |
> | T-EEG2 | **0.447** | 0.390 | 0.434 | 0.367 | 0.395 | 0.365 |
> | T-MEG | **0.453** | 0.410 | 0.414 | 0.374 | 0.386 | 0.339 |
> | Average | 0.516 | 0.503 | **0.527** | 0.440 | 0.455 | 0.428 |
> | Avg. #Parameters | $9.50\times 10^{8}$ | $8.16\times 10^{7}$ | **$7.79\times 10^{7}$** | $8.16\times 10^{7}$ | $7.79\times 10^{7}$ | $8.01\times 10^{7}$ |

---

> > ### Author Rebuttal · Reviewer_tZ5A · 2026-04-01
> >
> > I thank the authors for the detailed rebuttal. I fully agree with the authors that we shouldn't push for alignment at all cost, and I am particularly interested in the discussion on the alignment-accuracy tradeoff of various models, since it could suggest fundamental discrepancy between how AI and human understand the world.
> >
> > I already suggest for acceptance in my original review, and I will keep it as it is.

---

### Official Review · Reviewer_vH7z · 2026-03-12

**Soundness:** 2
**Presentation:** 3
**Significance:** 3
**Originality:** 2
**Overall Recommendation:** 4
**Confidence:** 2

**Summary:**

This paper studies how model–brain alignment scales with 3 key factors: pretraining scale, neural fine-tuning, and the amount of neural data used to fit model-to-brain mappings—across 8 datasets spanning ephys, fMRI, EEG, and MEG. It shows that larger pretraining improves alignment but eventually saturates, while additional neural supervision and larger mapping datasets continue to provide improvements. The paper also introduces a multi-subject attention-based mapping that improves alignment with fewer parameters.

**Compliance With Llm Reviewing Policy:**

Affirmed.

**Final Justification:**

The authors have mostly addressed my concerns. Overall, I think the proposed model is reasonable and could be useful, and therefore raised my score to weak accept.

**Key Questions For Authors:**

1. The paper evaluates model–brain alignment using Pearson correlation, which capture linear relationships. Could the authors comment on whether the reported scaling trends hold under alternative metrics (e.g., variance explained or likelihood-based measures)?
2. Could the authors clarify how sensitive the results are to the layer-selection procedure?
3. What support is there for training across such heterogeneous modalities (e.g., ephys, EEG, fMRI, etc.)? For example, some modalities provide whole-brain coverage (e.g., fMRI), while others only record from limited brain regions (e.g., ephys). It would be helpful if the authors could clarify how these differences in spatial coverage and measurement characteristics are handled across modalities, and whether they affect the reported scaling trends.

**Limitations:**

Yes

**Strengths And Weaknesses:**

Soundness: The paper presents an empirical study of model–brain alignment across multiple datasets and modalities, with experiments conducted on a large collection of pretrained vision models. The method appears technically sound and the empirical evidence generally supports the paper’s claims.

Presentation: The paper is well organized. The paper also provides a clear description of the evaluation pipeline and situates the work within prior literature on neural alignment and scaling laws.

Significance: Understanding how scaling affects brain–model alignment is an important question in NeuroAI, and the study provides useful empirical insights into how pretraining, neural supervision, and mapping data contribute to performance.

Originality: While the individual components (e.g., scaling analyses, neural fine-tuning, and encoding models) build on existing techniques, the paper’s novelty lies in the unified multimodal analysis across datasets and the systematic investigation of scaling trends.

---

> ### Author Rebuttal · Authors · 2026-03-31
>
> We thank the reviewer for the careful summary and for highlighting the paper’s scope and empirical value. We address your questions and concerns below.
>
> > Main takeaway and why it matters.
>
> The central contribution of the paper is not only to report scaling curves, but to identify which axes of scaling remain most useful for improving brain-predictive models. Across eight datasets and four recording modalities, we find that scaling generic visual pretraining improves alignment but exhibits clear diminishing returns, whereas additional neural supervision, more paired neural data for fitting mappings, and better mapping design remain higher-leverage directions. This matters for NeuroAI because it turns model–brain alignment from a mainly benchmark-driven comparison into a resource-allocation question: where should future effort go to obtain the next meaningful gains? Our results suggest that future progress is less likely to come from scaling standard pretraining alone and more likely to come from improving neural supervision and decoding methodology.
>
> > Alternative metrics.
>
> We agree that robustness beyond Pearson correlation is important. In Appendix E, we demonstrate how alignment scales using other metrics widely adopted in the NeuroAI field. Figs. S20--S23 show complementary analyses using RSA and CKA, including voxel-encoding variants, and the main pretraining-scaling conclusions remain qualitatively unchanged. We also find that explained-variance plots show a similar saturating scaling behavior; we will add a dedicated figure for this in the revision. In the revision, we will make these results more visible so that the paper does not rely on a single metric family.
>
> > Sensitivity to layer selection.
>
> We take this concern seriously. The committed layer is chosen using nested cross-validation on the training split only, with the test split held out throughout, so the layer assignment is not test-tuned. In Appendix D (Figures S12–S19), we report on cross-dataset ROI alignment and how alignment progresses through the model hierarchy for several metrics. Across datasets and model layers, the hierarchical progression is remarkably smooth without sudden spikes, recovering the well-known phenomenon that higher brain regions map to later model stages. Because this replicates prior findings, we placed it in the appendix. Additionally, we tested the robustness of the random projection setup across models and datasets (Appendix C, Figures S9 & S10), showing that our results are highly robust to changes in the projection size.
>
> > How do we handle heterogeneous modalities fairly?
>
> We do not directly concatenate or pool raw responses across modalities. Instead, each dataset is preprocessed with a modality-appropriate pipeline, evaluated with its own subject/ROI/channel structure, and normalized by dataset-specific noise ceilings before comparison (Appendix A). Most main scaling results are shown per benchmark rather than after aggressive aggregation (Figs 2,5). When we report an average summary, it is over normalized benchmark-level scores, so modalities with more voxels/channels do not dominate simply because they have higher dimensionality. To handle differences in spatial coverage across modalities for fine-tuning (Sections 4.3–4.6), we targeted a set of ROIs that impact alignment the most per dataset, while keeping the relevant functional spaces as analogous as possible. For example, we targeted the Inferior Temporal (IT) cortex in T-fMRI, TVSD-EP, the ventral stream in NSD-fMRI, and the occipital-parietal regions for EEG/MEG, ensuring that the regions broadly cover the same visual processing hierarchies. We ran additional checks on how different ROIs affect alignment (Figure 4b), revealing that across datasets, targeting higher ROIs consistently improves alignment more. We will make this aggregation choice and its motivation clearer in the revision.
>
> > Why train across such heterogeneous datasets?
>
> The motivation is not to claim that EEG, MEG, fMRI, and Ephys are interchangeable, but to ask whether the same broad scaling trends persist despite their very different measurement properties. We view that as a strength of the study: pretraining saturation, fine-tuning gains, and mapping-data scaling all recur across these heterogeneous settings, while modality-specific differences remain visible in the absolute ceilings and rate of improvement. We will sharpen this framing in the revision.
>
> > Clarification on Soundness Score
>
> We noticed that you rated the Soundness of our paper as a 2 ("fair"), despite graciously noting in your summary that the method appears technically sound and the empirical evidence supports the claims. Could you kindly clarify if there are specific soundness concerns we missed?
>
> Thank you again for your review. We hope our responses clarify your questions, and we kindly ask you to consider raising your score when taking these explanations and the newly planned manuscript additions into account.

---

> > ### Author Rebuttal · Reviewer_vH7z · 2026-04-02
> >
> > The authors have addressed most of my questions, and I would like to raise my score to 4 (weak accept).

---

> > > ### Author Response · Authors · 2026-04-02
> > >
> > > Thank you for taking the time to read our rebuttal and for your positive feedback. We are very glad to hear that our responses and planned revisions have adequately addressed your concerns.
> > >
> > > We deeply appreciate your willingness to raise your overall recommendation to **4 (Weak Accept)**. As the system currently still reflects your initial rating, we would be very grateful if you could update your score on OpenReview to reflect your current assessment. This helps ensure that the final decision accurately captures your current view of the paper.
> > >
> > > Thank you again for your time and for helping us improve our work.

---

### Official Review · Reviewer_2tA5 · 2026-03-12

**Soundness:** 2
**Presentation:** 3
**Significance:** 3
**Originality:** 3
**Overall Recommendation:** 4
**Confidence:** 4

**Summary:**

This paper introduces scaling laws for computational models of the visual cortex. It evaluates these models based on brain-model alignment, which is measured by how accurately the models' learned embeddings can predict neural responses. Overall, the study provides valuable insights into the capacity of current vision models to capture and predict biological neural activity.

**Compliance With Llm Reviewing Policy:**

Affirmed.

**Final Justification:**

The paper is well written and provides an interesting overview of the scaling behavior of models, as well as an examination of how model representations align with brain data. The authors have addressed my questions regarding the key takeaways in the rebuttal. For the final revision, I encourage the authors to expand the discussion on the potential impact on future research, particularly in relation to newer mapping methods.

**Key Questions For Authors:**

1. While the paper provides extensive empirical analysis of brain–model alignment across model scales, it is unclear what actionable insight follows from these findings. For example, should these results influence how models are pretrained, or suggest new directions for visual neuroscience research?
2. The alignment metric is evaluated using a linear regression mapping between model embeddings and neural responses. It is unclear whether the observed results reflect limitations of the model representations or the linear encoding model used. The authors should justify why linear mapping is appropriate or evaluate whether the conclusions hold with more expressive mapping architectures.

**Limitations:**

yes

**Strengths And Weaknesses:**

Strengths
1. The paper provides an interesting overview of the scaling behavior of models and examines how model representations align with brain data.
2. The paper is well written, and the authors conduct extensive empirical analyses to support their observations.

Weaknesses
1. The authors should better justify the brain–model alignment metric used in the paper. It is unclear why this metric is important beyond applications in multimodal neural response prediction (e.g., fMRI or EEG). Since the evaluated models are not trained for neural prediction tasks, it is unclear what practical insight this metric provides.
2. It is unclear what new insight the paper provides. While the authors present extensive analyses, the main conclusion appears to be that the alignment metric plateaus as model size increases. However, the implications of this observation remain unclear—specifically, how it should guide future research or model design.
3. The alignment metric is evaluated using a linear regression encoding model, which raises concerns about whether the observed effects are due to limitations of the model representations or the mapping function used to predict neural responses. The paper would benefit from a deeper discussion of this issue. In addition, more recent work has explored richer mapping functions for studying brain–model alignment, which are not discussed in the paper[1][2].

[1] Meta-Learning an In-Context Transformer Model of Human Higher Visual Cortex, https://arxiv.org/abs/2505.15813, NeurIPS 2025

[2] Beyond Grid-Locked Voxels: Neural Response Functions for Continuous Brain Encoding https://arxiv.org/pdf/2510.07342, ICLR 2026

---

> ### Author Rebuttal · Authors · 2026-03-31
>
> We thank the reviewer for their feedback and for recognizing the value of our extensive empirical analysis. We respond to your questions and concerns below.
>
> > Why is this useful beyond benchmarking?
>
> The main actionable takeaway of the paper is a resource-allocation result. We studied how alignment scales with three ingredients: task pretraining, neural finetuning, and fitting of neural mappings. Across eight datasets and four recording modalities, generic pretraining scale improves alignment but exhibits clear diminishing returns, whereas additional paired neural data and better mapping strategies continue to yield substantial gains. In other words, our results suggest that the next increment of progress is less likely to come from simply scaling standard visual pretraining further, and more likely to come from (i) collecting/using more neural supervision and (ii) improving the model-to-brain mapping. We will revise the discussion to make this implication more concrete for both NeuroAI practitioners and neuroscientists.
>
> > Why use noise-normalized Pearson with a simple linear encoding model?
>
> Our goal in the main scaling analysis is to isolate the quality of the learned representation rather than the capacity of a highly expressive decoder. Accordingly, we follow the standard NeuroAI protocol of frozen features plus a regularized linear readout, evaluated on held-out data and noise-normalized by dataset-specific ceilings (Yamins2016; Schrimpf2020). In this setting, the metric answers a specific question: how much task-trained model information is already linearly accessible to the recorded neural representation? We agree that richer mappings are also important, but they answer a different question, closer to predictive upper bounds than representation-level correspondence. Accordingly, we first study how representation quality changes as we scale pretraining under a standard protocol in the field (Secs. 4.1–4.2), and then analyze how alignment can be improved further through neural fine-tuning or better mappings (Secs. 4.3–4.9). Pearson’s r is also a widely used, interpretable NeuroAI metric that applies across many experimental settings; both Chen2026 and Yu2025 (suggested by the reviewer) use it and compare against linear baselines. We will clarify this distinction more explicitly in the revision.
>
> > Is the mapping the real bottleneck?
>
> Linear mapping is a strong and widely used baseline in model–brain alignment work. Numerous works have used linear mappings to achieve competitive predictivity (Schrimpf2018, Adeli2025) and to enable scientific discovery based on the brain-predictive models (Cerdas2024, Tang2025). Nevertheless, we agree this is an important concern. In fact, one of the paper’s central conclusions is precisely that mapping design matters substantially. The current submission already goes beyond linear probes by introducing attention-based mappings and showing that subject-shared cross-attention can match or exceed per-subject linear decoding while using far fewer parameters (Table 2, Figs. 6, S27, and S28). We will also add additional comparisons to other compact nonlinear mappings (shallow MLP and low-rank bilinear readouts) to show that the main conclusion is not “linear is sufficient,” but rather that representational scaling alone is not the full story and that mapping design is a major remaining bottleneck. For the expanded mapping comparison, please see our response to Reviewers **tZ5A** and **twXG**.
>
> > Robustness beyond one metric
>
> The main pretraining-scaling trends are reproduced in Appendix E using RSA and CKA (Figs. S20-S23), including voxel-encoding variants, and the qualitative conclusions remain the same. In the revision, we will surface these results more clearly in the main text and add explained-variance results as an additional robustness check.
>
> > Relation to newer readout work
>
> We agree the related-work discussion should better cover recent richer mapping approaches. We thank the reviewer for pointing us to Chen2026 and Yu2025; we will discuss both in the revision. These works reinforce the importance of improved readout design for brain prediction, and Yu2025 is especially aligned with one of our main messages: gains can come not only from better backbone representations, but also from stronger decoding strategies. Both papers also use Pearson r, and compare against linear baselines. Our contribution is complementary: rather than proposing only a new readout, we systematically show across datasets and modalities that mapping design and neural sample size remain major bottlenecks.
>
> Thank you again for your review. We hope these clarifications address the reviewer’s main concerns, and we respectfully ask the reviewer to consider raising their score.

---

> > ### Author Rebuttal · Reviewer_2tA5 · 2026-04-03
> >
> > Thanks for the detailed rebuttal. Most of my questions are addressed. I'll riase my score to 4.

---

### Decision · Program_Chairs · 2026-04-30

**Decision:**

Accept (regular)

**Comment:**

This work is focused on assessing scaling laws in models of the visual system. Specifically, it uses three main factors to assess the "model-to-brain" alignment: 1) pretraining scale (model size/resources), 2) neural fine-tuning, and 3) how much data is used to fit the model-to-brain mappings. The authors apply these assessments across eight different datasets of different neural recording modalities, showing that 1) more pretraining initially improves alignment but has diminishing returns, and 2) increasing neural supervision and the size of mapping datasets improve alignment without the eventual saturation.

 There were a number of noted concerns by the reviewers, including the utility of models across overly diverse datasets, missing references, and clarification on the scope of the work. While there are some points that are still weaknesses, they are not unique to this work but rather orthogonal limitations of the field as it stands (e.g., using linear brain-model mappings and specific tasks). The remainder of the clarifications and concerns seem to be addressed adequately by the reviewers so I will recommend to accept this work.